# Beyond Embeddings: On the Geometry
# of Probabilistic Information Retrieval via Diffusion

## Abstract

Recent theoretical analyses establish a sign-rank barrier that fundamentally limits the expressive capacity of fixed-dimensional embeddings for ranking as corpus size increases. In this work, we isolate geometric saturation effects through a parameter- and training-free diagnostic framework, contrasting the sensitivity of single-vector embeddings with probabilistic diffusion-based retrieval via the local Lipschitz constant of the optimal retrieval mapping. We show formally that token-level conditioning in diffusion circumvents the sign-rank barrier, yielding logarithmic rather than exponential scaling in geometric separation requirements. Evaluated across over 100 diverse information retrieval (IR) datasets, our framework confirms these divergent scaling behaviors in practice: embeddings approach geometric saturation, while diffusion preserves robust separability. Notably, the observed magnitude of embedding degradation remains small, providing a principled explanation for the continued empirical success of embedding-based methods despite theoretical sub-optimality. Our diagnostics offer a low-cost tool for anticipating when probabilistic paradigms may yield structural advantages in future compositional retrieval tasks.

## 1. Introduction

Information retrieval (IR) serves as the backbone of modern AI, facilitating knowledge access for large-scale search engines and the retrieval-augmented generation (RAG) components of large language models. The evolution of these systems has led to a demand for increasingly complex retrieval capabilities, such as resolving multi-document queries, capturing semantic nuances beyond lexical overlap, and main-

taining precision across massive document corpora. The dominant production paradigm remains embedding-based dense retrieval, valued for its simplicity and inference efficiency. However, a growing body of theoretical work suggests that mapping high-entropy document semantics into a fixed-dimensional vector space is subject to fundamental geometric constraints, such as the exponential sign-rank barrier (Weller et al., 2025). While these constraints imply that embeddings should struggle with complex or large-scale tasks, they continue to perform remarkably well in practice. To bridge this gap, the IR community has largely relied on auxiliary systems, such as multi-stage re-ranking (Nogueira & Cho, 2019) pipelines or query rewriting (Yu et al., 2020), to circumvent perceived embedding failures. Yet, there remains a lack of principled diagnostic frameworks to quantify the actual geometric saturation of these models or to explain their continued efficacy despite theoretical sub-optimality. While retrieval accuracy metrics like NDCG or Recall remain paramount for measuring downstream performance of retrieval systems, they are *lagging indicators* that are only interpretable with respect to a single static dataset and unreliable under post-hoc changes (e.g. adding an arbitrary number of random false negatives can alter performance metrics).

In this work, we investigate the geometric sensitivity of retrieval models by contrasting the standard embedding paradigm with a diffusion-based generative retrieval approach. We view diffusion not merely as a candidate method, but as a mathematically grounded baseline that utilizes token-level conditioning to bypass the sign-rank limitations inherent in single-vector mappings. Our analysis provides an unbiased explanation for the prevalence of embeddings while establishing a theoretical environment for the next generation of compositional retrieval systems. Our contributions are as follows:

- **A Parameter-Free Diagnostic Framework:** We derive a training- and parameter-free method that stays consistent under dynamic document corpora by systematically analyzing the geometry of document spaces. By measuring both the geometric distortion and local sensitivity of the optimal retrieval mapping, this framework quantifies the trade-off between mathematical

[1]Anonymous Institution, Anonymous City, Anonymous Region, Anonymous Country. Correspondence to: Anonymous Author <anon.email@domain.com>.

Preliminary work. Under review by the International Conference on Machine Learning (ICML). Do not distribute.

retrieval capacity and task complexity, while providing an empirical estimate of the sign-rank barrier's impact.

- **Theoretical Characterization of Scaling Laws:** We provide a formal proof that token-level conditioning in diffusion-based retrieval models enables a logarithmic dependency on corpus size for maintaining separability. This is contrasted with the exponential dependency required by fixed-dimensional embeddings, demonstrating how iterative denoising natively circumvents the sign-rank barrier in multi-document tasks.

- **Empirical Validation and Geometric Explanation:** We apply our framework to over 100 diverse IR datasets and 3 embedding-models of different capacity to validate these scaling trajectories. Our results reveal that, while embeddings do follow the predicted path toward geometric saturation and information loss, the observed magnitudes of degradation remain sufficiently small for current task complexities. This provides a principled explanation for the empirical success of embeddings while identifying the structural thresholds where probabilistic paradigms become necessary.

The full source code to reproduce the displayed results is available as a ZIP in the supplementary material.[1]

## 2. Related Work

**Scaling laws in information retrieval** Inspired by neural scaling laws in large language models (Kaplan et al., 2020), recent studies characterize retrieval performance as power-law functions of model parameters and data size. Early analyses identified the *embedding bottleneck*, where compressing varying-length semantics into fixed-size vectors incurs inevitable information loss (Luan et al., 2021). (Fang et al., 2024) systematically analyze how model size and training data affects generative retrieval (GR), finding favorable logarithmic capacity growth when encoding corpus information in parameters. This contrasts with dense retrieval, which (Muennighoff et al., 2023) show exhibits sublinear scaling returns due to the embedding saturation described above. Similarly, (Pradeep et al., 2023) argue that generative models effectively compress the corpus into weights, avoiding linear memory growth of dense indices. Most notably, theoretical results on the *sign-rank* of neural networks establish that the dimension required to linearly separate arbitrary preference relations via dot products scales exponentially with the complexity of the target function (Weller et al., 2025). This theoretically precludes fixed-dimensional embeddings from resolving highly combinatorial relevance patterns in large corpora.

---

[1]This is done to preserve anonymity during the peer-review process. The full code will be accessible in a public repository for the camera-ready version.

Our work accepts these hard limits as a theoretical baseline but diverges in aim. While Weller et al. (2025) prove that embeddings *eventually* fail, they do not explain the *unreasonable effectiveness* of embeddings in the pre-saturation regime. We bridge this gap by introducing a local Lipschitz-based diagnostic that quantifies exactly how close a system is to this barrier for a given task complexity, rather than bounding the worst-case scenario.

**Generative and diffusion-based retrieval** To avoid the bottleneck of single-vector encodings, the field has adopted increasingly granular representations. Late interaction paradigms, most notably ColBERT (Khattab & Zaharia, 2020), break the single-embedding constraint by retaining token-level matrices, effectively mitigating geometric saturation through multi-vector matching. However, these methods remain bound to discriminative scoring functions (e.g., MaxSim) and do not model the underlying probability distribution of relevant documents.

Parallel research has moved toward generative paradigms. Methods like Differentiable Search Index (DSI) (Tay et al., 2022) and Search Engines with Autoregressive Language Models (SEAL) (Bevilacqua et al., 2022) replace the geometric index with an autoregressive model that generates document identifiers directly. More recently, diffusion-based methods (e.g., DiffuRetrieval (Zhang et al., 2024), DiffuGR (Zhao et al., 2025)) have utilized iterative denoising to model complex query-document distributions.

While these methods show empirical promise, their theoretical justification has largely been heuristic. We provide the missing formal link: we show that the token-level conditioning (a structural advantage shared with late interaction) enables the diffusion reverse process to mathematically circumvent the sign-rank barrier. This transforms the exponential dependency on corpus size into a logarithmic one, establishing diffusion as a theoretically grounded probabilistic generalization of high-capacity retrieval.

## 3. Framework for Analyzing Retrieval Geometry

We analyze retrieval systems through a geometry-first lens. A retrieval problem is specified by a finite corpus $\mathcal{D}$ and a query space $\mathcal{Q}$. For each query $q \in \mathcal{Q}$ define the (binary or graded) relevance function $r(q, d)$ and the ground-truth relevance set

$$\mathcal{S}_q := \{d \in \mathcal{D} : r(q, d) = 1\}.$$

We suppose that each document $d \in \mathcal{D}$ admits a fixed representation $f(d) \in \mathcal{F} \subset \mathbb{R}^{d_f}$ (the *document space*), and a retrieval method is specified by a triple $(E_\phi, g_\psi, \text{sim})$ where

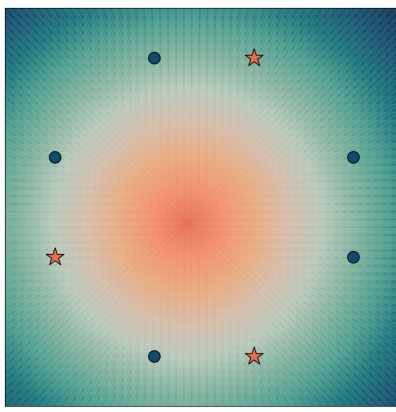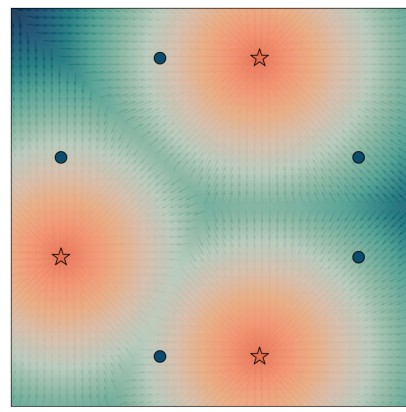

*Figure 1.* Visualization of geometric failure modes of embedding-based models (left), diffusion with embedding-based conditioning (middle) and diffusion with token-level conditioning (right) in 2d space. Systems with low capacity (query-embedding) and low expressivity (centroid) accumulate error. Systems with low capacity (query-embedding) and high expressivity (idealized score-field) are unable to translate expressivity into practice. Systems with high capacity (token-level-embeddings) and high expressivity (score-field) can resolve complex, multi-hop retrieval tasks.

- $E_\phi : \mathcal{Q} \to \mathcal{C} \subset \mathbb{R}^d$ is a (possibly learned) query encoder,

- $g_\psi : \mathcal{C} \to \mathcal{F}$ is a mapping that aligns encoded queries to the document space (for canonical embedding retrieval $g_\psi = \mathrm{id}$),

- $\mathrm{sim} : \mathcal{F} \times \mathcal{F} \to \mathbb{R}$ is a metric evaluating relevance of queries and retrieval targets.

The method induces a *predictive* distribution over documents for a query $q$ by applying a tempered softmax over similarity scores:

$$p_q^{(\phi,\psi)}(d) = \frac{\exp\big(\mathrm{sim}\big(g_\psi(E_\phi(q)), f(d)\big)/\tau\big)}{\sum_{d' \in \mathcal{D}} \exp\big(\mathrm{sim}\big(g_\psi(E_\phi(q)), f(d')\big)/\tau\big)}. \quad (1)$$

### 3.1. Geometric Failure Modes

We operationalize two distinct geometric failure modes. Concepts are visualized in Figure 1.

**(i) Geometric distortion.** Define the *ideal target distribution* $\pi_q$ which places mass uniformly (or graded by relevance like in Appendix B.1) on $\mathcal{S}_q$ and zero elsewhere:

$$\pi_q(d) = \begin{cases} \frac{1}{|\mathcal{S}_q|} & d \in \mathcal{S}_q, \\ 0 & \text{otherwise.} \end{cases}$$

We quantify geometric distortion by the expected divergence between the induced predictive distribution and the ideal:

$$\varepsilon_{\text{target}}(\phi, \psi) = \mathbb{E}_{q \sim \mathcal{Q}}\big[\mathrm{D}\big(\pi_q \,\|\, p_q^{(\phi,\psi)}\big)\big], \quad (2)$$

where $\mathrm{D}(\cdot\|\cdot)$ is an arbitrary divergence, for instance, KL. Note that while this divergence is not disconnected from

the query-embedding in practice, this failure mode assumes perfect information capacity given idealized output of $g_\psi$.

**(ii) Geometric sensitivity.** Let $h_{\phi,\psi} \coloneqq g_\psi \circ E_\phi : \mathcal{Q} \to \mathcal{F}$ denote the end-to-end mapping from queries to points in document space. For any $q \in \mathcal{Q}$ define the local geometric sensitivity (local Lipschitz constant)

$$K_h(q; r_q) = \sup_{\substack{q' \in \mathcal{Q} \\ 0 < \mathrm{dist}_{\mathcal{Q}}(q,q') \leq r_q}} \frac{\mathrm{dist}_{\mathcal{F}}\big(h_{\phi,\psi}(q), h_{\phi,\psi}(q')\big)}{\mathrm{dist}_{\mathcal{Q}}(q, q')},$$
$$(3)$$

where $\mathrm{dist}_{\mathcal{Q}}$ and $\mathrm{dist}_{\mathcal{F}}$ are chosen distances on query and document spaces, and $r_q > 0$ is a neighborhood radius (we take $r_q$ equal to the distance to the nearest relevance-distinct query). Intuitively, $K_h(q)$ measures how sharply the representation changes with small perturbations to the query; large $K_h(q)$ implies a stiff geometry that a bounded-capacity model must resolve with high precision.

This failure mode aims at measuring the strain on the mapping function $g_\psi$, by evaluating systematic mismatches in $\mathcal{Q}$ and $\mathcal{F}$. For instance, if $E_\phi$ produces low-capacity embeddings for $\varepsilon(\phi, \psi) = 0$ with zero-distortion $\varepsilon_{\text{target}}(\phi, \psi^*) = 0$ for idealized $\psi^*$, this relocates resolution pressure to $g_{\psi^*}$.

### 3.2. Estimating Sign-Rank Proxies

Because the exact sign-rank of a binary decision matrix is NP-hard to compute (Bhangale & Kopparty, 2015), it remains intractable for large corpora. To address this, we adopt empirically computable geometric proxies that reflect the same underlying structural pressure. In particular, we argue that the *number of relevant documents per query*, denoted $n_q \coloneqq |\mathcal{S}_q|$ for query $q$, serves as a practical and informative surrogate for the geometric complexity associ-

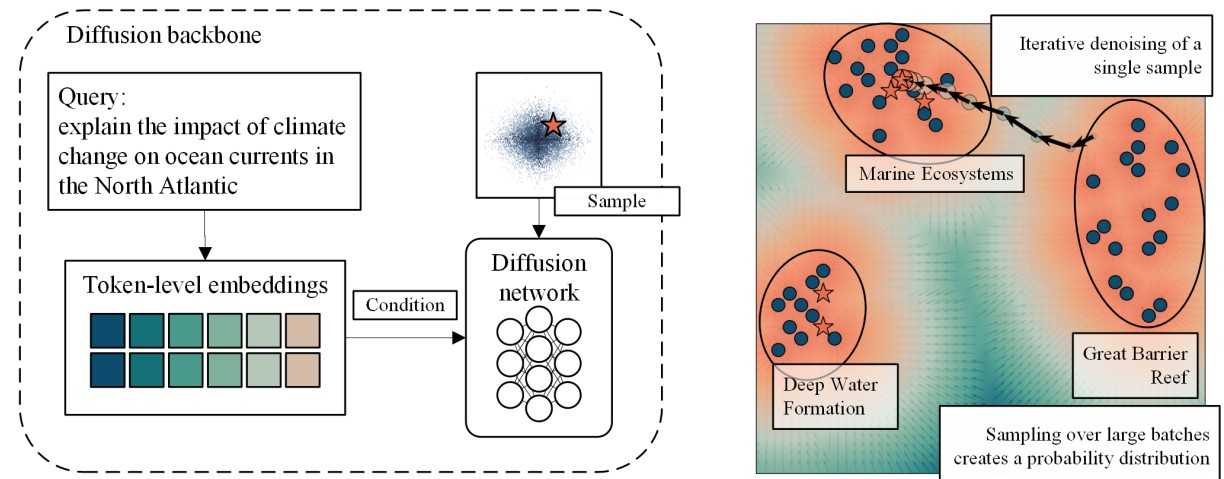

*Figure 2.* Visualization of information retrieval via diffusion with token-level conditioning. The query encoder converts the query into token-level embeddings. This input is used as the condition for our diffusion model (mapping function), iteratively denoising a random sample (black arrows for a single sample). Once repeated over many random samples, a probability distribution over document space where relevant documents are found emerges.

ated with high sign-rank. This choice is motivated by two observations. First, any binary decision matrix with high sign-rank must include queries whose positive sets contain multiple documents; such combinatorial variation across rows (queries) cannot occur when all relevance sets are singletons. Second, the converse does not necessarily hold: a large $n_q$ does not guarantee high sign-rank, since redundancy or clusterable documents may yield geometrically simple structures in the embedding space $\mathcal{F}$. Hence, $n_q$ is a *necessary but not sufficient* indicator of sign-rank pressure.

Operationally, we treat $n_q$ as a control parameter governing geometric stress within the encoder. As $n_q$ increases, two distinct effects may emerge depending on the spatial distribution of relevant documents in $\mathcal{F}$. When these documents are tightly clustered, larger $n_q$ primarily adds redundancy, leaving geometric distortion and sensitivity largely unchanged. In contrast, when the documents are geometrically diverse, increasing $n_q$ raises the combinatorial complexity of the task and thereby intensifies the effective sign-rank pressure on any finite-dimensional representation. Consequently, $n_q$ provides a measurable and interpretable axis along which to organize empirical results (see Sec. 5.1).

### 3.3. Summarizing the Framework

Weller et al. (2025) show that the sign-rank barrier inhibits models in representing arbitrary queries. However, since embedding-models are trained to align query- and document-space, we expect this limitation to be mitigated in practice.

Therefore, our framework estimates the impact of sign-rank pressure on a given document space $\mathcal{F}$ of an arbitrary (pre-trained) encoder. In 3.1, we argue that geometric pressure can occur geometrically in two distinct modes, distortion and sensitivity. Further, we reason that the chance of encountering high sign-rank is enabled (but not guaranteed) by the number of retrieval targets per query $n_q$. Therefore, we compare our evaluation metrics for increasing $n_q$, to empirically measure how well $\mathcal{F}$ remains organized under increasing combinatorial stress.

## 4. Probabilistic Information Retrieval via Diffusion with Token-Level Conditioning

We now formalize conditional diffusion retrieval and show how its conditioning mode interacts with the geometric failure modes defined above.

### 4.1. A Conditional Diffusion View of Retrieval

We represent a retrieval target as a *soft selection vector* over the corpus,

$$x \in \Delta^{N_D} \subset \mathbb{R}^{N_D}, \qquad x_d \geq 0, \sum_d x_d = 1,$$

where ideally $x = \pi_q$, the relevance distribution for query $q$ (see Sec. 3). A conditional diffusion model defines a forward noising process on $x$ and learns a conditional score field

$$s_\theta(x, t; c) = \nabla_x \log p_\theta(x, t \mid c),$$

with conditioning information $c$ derived from the query (e.g., $c = E_\phi(q)$ for embedding conditioning or $c = q$ for token-level conditioning). The model is trained by standard denoising score matching over progressively noised samples $x_t$, encouraging $s_\theta$ to approximate the true score $s^*(x_t, t; c)$

*Table 1.* Scaling behavior of empirical Lipschitz constants ($K$) and geometric sensitivity ($\varepsilon$). We observe distinct scaling laws for the two metrics: $K$ is best modeled by an exponential fit ($K_b = C_K e^{\lambda_b n_q}$), while $\varepsilon$ follows a power-law relationship ($\varepsilon_b = C_\varepsilon n_q^\alpha$). $C$, $\lambda$, and $\alpha$ denote the base coefficient, exponential rate, and scaling exponent respectively. Values are evaluated across all datasets and encoders analyzed. Note that [–] indicates no approximated scaling regime, as for our zero-distortion diffusion-based methods.

| Mode ($b$) | Median $\Delta K_b$ | | | Median $\Delta \varepsilon_b$ | | |
|---|---|---|---|---|---|---|
| | $C_{K_b}$ | $\lambda_{K_b}$ | $R^2_{K_b}$ | $C_{\varepsilon_b}$ | $\alpha_{\varepsilon_b}$ | $R^2_{\varepsilon_b}$ |
| | *Global aggregate* | | | | | |
| $b_1$ (Embedding) | 0.9830 | -0.0003 | 0.9558 | 0.5480 | 0.0781 | 0.8687 |
| $b_2$ (Diffusion + Embedding Cond.) | 0.9894 | 0.0001 | 0.8756 | [–] | [–] | [–] |
| $b_3$ (Diffusion + Token Cond.) | 1.0106 | 0.0000 | 0.0420 | [–] | [–] | [–] |

across diffusion timesteps. At inference, the learned reverse process generates a clean estimate $\hat{x}_0$, from which the top-$k$ documents are retrieved by selecting the highest-mass components of $\hat{x}_0$ (shown in Figure 2).

### 4.2. Embedding Conditioning vs. Token conditioning

**Embedding conditioning ($c = E_\phi(q)$).** When the conditioning is an encoded, fixed-dimensional vector $c \in \mathbb{R}^d$, the diffusion head is strictly limited by the resolution of the conditioning channel. The volumetric pigeonhole argument (Theorem A.1) implies that as $|\mathcal{D}|$ grows, the available volume for distinguishing relevance sets shrinks exponentially. This forces the conditional score function $s^*(\cdot, \cdot, c)$ to learn increasingly sharp gradients to distinguish between queries with infinitesimally close embeddings ($E_\phi(q) \approx E_\phi(q')$). In practice, this manifests as diverging local Lipschitz constants ($L_{\text{loc}} \to \infty$), leading to training instability and mode collapse unless the embedding dimension $d$ scales logarithmically with the corpus size.

**The illusion of $\delta$-separation.** While contrastive learning objectives are theoretically shown to encourage uniform distribution and alignment (Wang & Isola, 2020), they are bound by the physical limits of the hypersphere. Theorem A.2 suggests that retrieval complexity is independent of corpus size provided a minimum separation $\delta$ is maintained. However, this is an idealized upper bound that fails in large-scale settings. Packing bounds dictate that maintaining a fixed $\delta$ requires the dimension to grow as $d \approx \Omega(\log |\mathcal{D}|)$. In fixed dimensions, as $|\mathcal{D}|$ increases, the maximum achievable separation $\delta$ vanishes. Thus, the corpus-independent bound collapses, as the term $1/\delta^2$ in the sample complexity grows to dominate the retrieval cost.

**Token-level conditioning ($c = q$ or $c = \textbf{tokens}(q)$).** Conditioning directly on the discrete token sequence bypasses this continuous geometric bottleneck. Similar to the input preservation mechanism in ColBERT (Khattab & Zaharia, 2020), this approach leverages the combinatorial capacity of the vocabulary, which scales as $|V|^L$. However, unlike Late Interaction methods that aggregate tokens via fixed,

bounded similarity operations (e.g., MaxSim), diffusion utilizes this high-capacity conditioning to drive a highly non-linear generative process. Because distinct relevance concepts map to distinct token sequences (which are by definition separated by discrete edit distances rather than continuous $\epsilon$-balls), the model effectively conditions on signals whose distinguishability scales logarithmically with corpus size (Theorem A.3), avoiding the vanishing margin problem entirely.

## 5. The Framework in Practice

### 5.1. Methodology

We evaluate our geometric framework on a large and diverse collection of information retrieval datasets and pre-trained document spaces, quantifying how geometric error and sensitivity vary with increasing $n_q$.

**Defining document space.** Since we aim at identifying the practical implications of sign-rank barrier for retrieval tasks, we use popular pre-trained embedding models (all-MiniLM-L6-v2, all-mpnet-base-v2, and all-roberta-large-v1) of different sizes and dimensionality to define the document space $\mathcal{D}$. Table 2 outlines all encoders analyzed.

**Datasets and sampling.** Experiments are conducted on over one hundred publicly available retrieval datasets from the ir_datasets library, spanning a wide range of domains and corpus sizes. To better balance statistics across $n_q$, we sample at most $N_{\max} = 50$ queries per $n_q$ bin per dataset, drawn uniformly without replacement. All metrics are computed independently within each dataset and subsequently aggregated across datasets to yield stable, dataset-independent summaries. Queries are grouped into logarithmic bins based on $\log_2 n_q$, which provides an interpretable measure of geometric multiplicity while maintaining more balanced occupancy across bins. We share the total number of query-to-query comparisons per dataset for a given bin in Table 3.

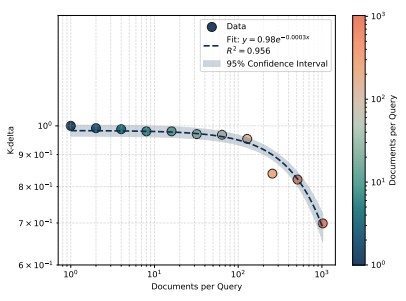 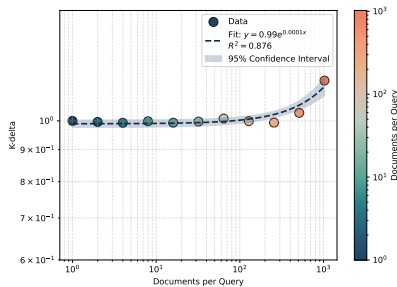 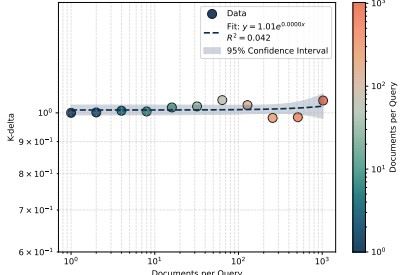

*Figure 3.* Visualization of median sensitivity change scaling in embedding-based systems (left), diffusion with query-embedding conditioning (middle), and diffusion with token-level conditioning (right). Embedding-based systems show decreasing sensitivity with rising number of documents per query, diffusion with embedding-based conditioning a slight increase, while diffusion with token-level conditioning remains largely stable.

**Metrics per query-pair.** For every query pair $(q, q')$ we compute the local geometric sensitivity $\widehat{K}_h(q, q')$ and geometric distortion $\widehat{\varepsilon}_{\text{target}}(q), \widehat{\varepsilon}_{\text{target}}(q')$. For analysis, we compare three different modes. Here, $b_1$ denotes our regime for traditional embedding-based methods, $b_2$ for diffusion conditioned on query embeddings, and $b_3$ for diffusion with token-level conditioning. Note that we define geometric distortion only for $b = 1$, since we assume the idealized score field at timestep $T = 0$ can greedily navigate to the closest document in space, resolving multi-hop dependencies via randomness. The exact calculations for each $K_b$ and $\varepsilon_b$ are outlined in the Appendix B.

**Differential analysis.** To isolate how geometry evolves with increasing document multiplicity, we compute bin-wise deltas relative to the smallest non-empty bin. Let $(K_{b_0}, \varepsilon_{b_0})$ denote the medians of the first bin. For each subsequent bin $b$ we define

$$\Delta K_{b_i} = K_{b_i} - K_{b_0}, \qquad \Delta \varepsilon_{b_i} = \varepsilon_{b_i} - \varepsilon_{b_0}.$$

These differences capture the average growth of geometric sensitivity (complexity) and target-collapse error as the number of relevant documents increases. Note that we have reported all absolute values in Tables 4, 5 and 6 in the Appendix. We argue that the rate per bin is more insightful, as absolute magnitudes can easily be adjusted through the addition of constant values in either the numerator or denominator of $\Delta K_b$ and $\Delta \varepsilon_b$.

To calculate the trends displayed in the Figure 3, we use standard regression in the respective regime (exponential and power-law). The calculated slopes are supported by corresponding $R^2$.

### 5.2. Results

We report empirical scaling behavior of geometric sensitivity and geometric distortion as functions of the number of relevant documents per query. Across all experiments, results are aggregated within logarithmic bins of $\log_2 n_q$

and summarized by change in median values as described in Sec. 5.1.

**Global scaling trends.** Figure 3 and Table 1 summarize the power-law relationships observed between local Lipschitz constants and query multiplicity. All three retrieval paradigms exhibit approximately exponential growth of median geometric sensitivity and power-law relations for error:

$$K_b = C_{K_b} \cdot e^{\lambda_{K_b} n_q}, \ \varepsilon_b = C_{\varepsilon_b} \cdot n_q^{\alpha_{\varepsilon_b}}$$

Base sensitivity $(C)$ and exponents $(\lambda, \alpha)$ are parameters for each paradigm, which are shown in Table 1.

**Relative geometric sensitivity.** The three retrieval paradigms, denoted $K_1$ (embedding-based), $K_2$ (diffusion with embedding conditioning), and $K_3$ (diffusion with token-level conditioning), display distinct yet smooth scaling profiles:

For $K_1$, median geometric sensitivity relaxes by a scaling exponent of around $\lambda_{K_b} \approx -0.0003$. However, this is accompanied by a larger rise in geometric distortion ($\alpha_{\varepsilon_b} \approx 0.0781$), indicating that the embedding centroid is collapsing away from the true relevance geometry.

For $K_2$, median geometric sensitivity increases for larger $n_q$ by a scaling exponent $\lambda_{K_b} \approx 0.0001$.

For $K_3$, median geometric sensitivity increases for larger $n_q$, but with a lower scaling exponent $\lambda_{K_b} \approx 0.0000$, resembling an almost constant regime.

**Relative geometric distortion.** As argued in the methodology, we can assume a zero-distortion regime for score-field based objectives. Nonetheless, for embedding-based systems, we can see a clear trend towards steadily increasing rate of geometric distortion for rising $n_q$.

**Encoder ablations.** Grouping results by encoder (Table 2) reveals comparable scaling exponents across all encoders,

*Table 2.* Scaling behavior of empirical Lipschitz constants ($K$) and geometric sensitivity ($\varepsilon$). We observe distinct scaling laws for the two metrics: $K$ is best modeled by an exponential fit ($K_b = C_K e^{\lambda_b n_q}$), while $\varepsilon$ follows a power-law relationship ($\varepsilon_b = C_\varepsilon n_q^\alpha$). $C$, $\lambda$, and $\alpha$ denote the base coefficient, exponential rate, and scaling exponent respectively. Values are evaluated across all datasets analyzed and grouped by encoder. Additional information on model size, dimensions, and performance (evaluated across 14 diverse tasks from different domains) sourced from the repository associated with (Reimers & Gurevych, 2019). Note that [–] indicates no approximated scaling regime, as for our zero-distortion diffusion-based methods.

| | Median $\Delta K_b$ | | | Median $\Delta \varepsilon_b$ | | |
|---|---|---|---|---|---|---|
| Mode ($b$) | $C_{K_b}$ | $\lambda_{K_b}$ | $R^2_{K_b}$ | $C_{\varepsilon_b}$ | $\alpha_{\varepsilon_b}$ | $R^2_{\varepsilon_b}$ |
| `all-MiniLM-L6-v2`: *Size: 80MB, dim=384, Performance: 0.6806* | | | | | | |
| $b_1$ (Embedding) | 0.9835 | -0.0001 | 0.5377 | 0.8293 | 0.0909 | 0.8333 |
| $b_2$ (Diffusion + Embedding Cond.) | 1.0641 | 0.0003 | 0.5535 | [–] | [–] | [–] |
| $b_3$ (Diffusion + Token Cond.) | 1.0156 | 0.0000 | 0.0259 | [–] | [–] | [–] |
| `all-mpnet-base-v2`: *Size: 420MB, dim=768, Performance: 0.6957* | | | | | | |
| $b_1$ (Embedding) | 0.9835 | -0.0001 | 0.6359 | 0.8435 | 0.0860 | 0.7477 |
| $b_2$ (Diffusion + Embedding Cond.) | 1.0053 | 0.0000 | 0.7574 | [–] | [–] | [–] |
| $b_3$ (Diffusion + Token Cond.) | 1.0143 | 0.0000 | 0.3764 | [–] | [–] | [–] |
| `all-roberta-large-v1`: *Size: 1360M, dim=1024, Performance: 0.7023* | | | | | | |
| $b_1$ (Embedding) | 0.9704 | -0.0002 | 0.5212 | 0.8432 | 0.0869 | 0.7091 |
| $b_2$ (Diffusion + Embedding Cond.) | 1.0038 | 0.0001 | 0.7223 | [–] | [–] | [–] |
| $b_3$ (Diffusion + Token Cond.) | 1.0133 | 0.0000 | 0.2091 | [–] | [–] | [–] |

with a slight bias towards less extreme scaling regimes for larger number of parameters and dimensions. For instance, the magnitude of sensitivity slope $\lambda_{K_2}$, our mismatched regime, is lower for our two largest encoders. Conversely, this pattern does not seem to repeat for geometric distortion.

### 5.3. Discussion

The empirical results presented in Sec. 5.2 support the central hypothesis that geometric constraints related to sign-rank manifest as systematic, though small, scaling effects in retrieval geometry. Across retrieval paradigms, both geometric sensitivity and target-collapse error follow approximate power-law relations with the number of relevant documents per query, consistent with the theoretical predictions derived from our framework.

**Magnitude of observed effects.** While the presence of power-law scaling confirms that sign-rank pressure is measurable in practice, the absolute magnitudes of change remain small, as indicated by the $\lambda_{K_1} = -0.0003$ and $\alpha_{\varepsilon_1} = 0.0781$ for embedding-based regimes. This observation suggests that, although the theoretical limitations associated with finite-dimensional embeddings (i.e., exponential sensitivity under fixed $d$) are real, they operate at magnitudes too small to meaningfully impair current retrieval performance within the explored data regimes.

**Relation to $\delta$-separation and encoder training.** The bounded slopes observed for all architectures are consistent with the $\delta$-separation condition formalized in Theorem A.2. Encoders appear to preserve a minimum inter-query separation that prevents the geometric sensitivity constant from diverging even as query multiplicity increases. This behavior provides a plausible reconciliation between the mathematical impossibility results, derived under worst-case assumptions, and the strong empirical performance of modern retrieval systems: encoders do not entirely escape the sign-rank barrier but operate in a regime where its quantitative impact is negligible. By analyzing three encoders of different parameter count, dimensionality and average downstream retrieval performance (Reimers & Gurevych, 2019), we establish an early cautious link with geometric sensitivity that should be investigated further in the future.

**Implications for large-scale retrieval.** Despite the small magnitudes, the monotonic trends indicate that geometric stiffness and sensitivity continue to grow with task complexity. Extrapolating the fitted exponential relations suggests that, at substantially larger scales (e.g., orders of magnitude more larger $n_q$ with disjoint relevance), these effects could become non-trivial. It is not difficult to imagine such regimes: for instance, critical applications using Agentic RAG and multi-hop reasoning can push retrieval requirements into the high-$n_q$ regime by including some post-hoc fact-checking algorithms. In such regimes, the geometric separation provided by current embedding models may no longer suffice to maintain low sensitivity without increasing dimensionality or auxiliary mechanisms.

**Diffusion-based retrieval.** Diffusion retrieval exhibits the theoretically expected behavior: token-level conditioning maintains near-constant geometric sensitivity across bins, while embedding-level conditioning inherits the same

growth trend as conventional encoders. This confirms that diffusion architectures can, in principle, decouple retrieval capacity from fixed-dimensional embedding geometry. However, the present framework intentionally abstracts away the substantial training and computational complexities intrinsic to diffusion models, such as instability of score-field optimization, high memory cost, and inference-time stochasticity. Consequently, while diffusion methods demonstrate favorable theoretical scaling, further work is required to determine whether these properties can be realized in practical retrieval settings.

**Summarizing the Results.** In summary, sign-rank–related geometric limitations are empirically detectable but remain of modest magnitude within current-scale retrieval systems. Existing encoders appear to satisfy approximate $\delta$-separation, mitigating exponential sensitivity effects. Nonetheless, as retrieval tasks continue to scale in size and multiplicity, the same geometric principles predict a gradual but unavoidable increase in sensitivity: an effect that diffusion-based retrieval may be uniquely positioned to address once its practical training challenges are resolved.

### 5.4. Limitations and Future Work

**Geometric analysis and retrieval failures.** Our framework focuses exclusively on the geometry of retrieval tasks and does not directly incorporate downstream retrieval performance metrics. Consequently, while the observed trends in geometric sensitivity and target-collapse error provide insight into sign-rank–related limitations, we refrain from making definitive claims that these directly cause retrieval failures. From the limited sample size of 3 encoders, we can see a tendency of high-performing models in terms of retrieval exhibit favorable geometric properties. Nonetheless, establishing a rigorous connection would require careful correlation analysis, and potentially causal inference, to link geometric measurements to end-to-end retrieval outcomes. Our initial results should serve as motivation for investigations but lie beyond the scope of the present work.

**Diffusion-based retrieval.** The theoretical analysis of diffusion retrieval in this work relies on a zero-error, final-timestep approximation of the score field (Sec. 4). As such, it does not account for practical considerations such as the iterative denoising dynamics prior to the final timestep, stochasticity in the reverse process and its effect on ranking, or model optimization stability and convergence issues. These factors may substantially influence empirical performance, and their interaction with the idealized geometric framework warrants detailed future investigation.

**Parameter scaling and architectural considerations.** Our diffusion-based retrievers are analyzed as a conceptual high-capacity counterpart to embedding-based retrieval augmented *with* auxiliary mechanisms such as query rewriting or re-ranking. While diffusion-based retrieval systems will likely be larger in parameter counts than pure embedding-based systems, further experiments will show whether they scale more gracefully in comparison to full pipelines. Future studies could explore scaling laws, memory usage, and runtime behavior of diffusion models in realistic retrieval pipelines, building on prior efforts (e.g., DiffuRetrieval (Zhang et al., 2024)) to deploy diffusion architectures in large-scale retrieval.

**Summarizing future directions.** Overall, the present study establishes a rigorous, geometry-focused framework for analyzing retrieval systems, identifies empirical scaling patterns, and proposes high-capacity alternatives. Future work should aim to empirically link geometric measurements to end-to-end retrieval performance and robustness, investigate the training and inference dynamics of diffusion-based retrieval in realistic deployments, or assess the practical feasibility of token-level diffusion at scale, including memory, computation, and model size considerations. These directions will help bridge the gap between theoretical capacity analyses and operational retrieval systems.

## 6. Conclusion

We present a rigorous framework for analyzing the geometric properties of retrieval systems, focusing on two key quantities: *geometric sensitivity*, which captures the local Lipschitz continuity of the query-to-document mapping, and *geometric distortion*, which measures the representational fidelity of queries to their relevant documents. By systematically evaluating a range of retrieval paradigms, we identify consistent scaling laws relating these geometric metrics to the number of relevant documents per query. While theoretical limitations imposed by sign-rank are evident in the scaling of geometric sensitivity and distortion, the absolute magnitudes remain small in the regimes studied, suggesting that modern encoders effectively mitigate potential failures under typical workloads.

In sum, this study provides both theoretical and empirical evidence that geometric limitations in retrieval systems are measurable but, under current encoding strategies, operate at modest magnitudes, highlighting opportunities for high-capacity models such as diffusion to extend retrieval capabilities in large-scale or high-multiplicity tasks.

## Impact Statement

"This paper presents work whose goal is to advance the field of Machine Learning. There are many potential societal

consequences of our work, none which we feel must be specifically highlighted here."

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

## A. Proofs

### A.1. Exponential Lower and Upper Bound on Conditioning Complexity

#### A.1.1. EXPONENTIAL LOWER BOUND ON CONDITIONING COMPLEXITY

**Theorem A.1.** *There exists a family of retrieval tasks with corpus size $|\mathcal{D}| = 2m$ and $\Theta(m^2)$ distinct queries such that for any query encoder $E_\phi : \mathcal{Q} \to \mathcal{C} \subset \mathbb{R}^d$ of bounded diameter $diam(E_\phi) \leq D$, there exist two queries $q, q'$ with disjoint relevance sets ($\mathcal{S}_q \cap \mathcal{S}_{q'} = \emptyset$) satisfying*

$$\|E_\phi(q) - E_\phi(q')\| \leq O\left(D\left(\frac{1}{m}\right)^{1/(2d)}\right). \tag{4}$$

*Consequently, for any fixed diffusion noise level $\sigma > 0$, the conditional score field $c \mapsto s^*(\cdot, c) = \nabla_x \log p_\sigma^*(x|c)$ has local Lipschitz constant obeying*

$$L_{loc} \geq \Omega\left(m^{2/d}\right) = 2^{\Omega(\log |\mathcal{D}|/d)}, \tag{5}$$

*i.e., exponential in corpus size unless the conditioning dimension $d = \Omega(\log |\mathcal{D}|)$.*

**Interpretation**   The exponential lower bound arises from geometric crowding in the conditioning space: with a bounded-diameter encoder, exponentially many relevance-distinct queries must map to exponentially close embeddings. The diffusion model must therefore learn a score field whose sensitivity to $c$ grows as $2^{\Omega((\log |\mathcal{D}|)/d)}$, implying that generalization requires either exponentially large model capacity or linearly growing conditioning dimension.

**Proof**   We construct an explicit hard family of retrieval tasks and apply a volumetric pigeonhole argument.

1. **Retrieval task construction**: For an integer $m \geq 1$, let the corpus be $\mathcal{D} = \{d_i^+, d_i^- : i \in [m]\}$ with $|\mathcal{D}| = 2m$. Each query corresponds to an unordered pair of indices $\{i, j\}$ with $1 \leq i < j \leq m$:

$$q_{i,j} \longleftrightarrow \mathcal{S}_{q_{i,j}} = \{d_i^+, d_i^-, d_j^+, d_j^-\}. \tag{6}$$

We treat each index $i \in [m]$ as a document pair $(d_i^+, d_i^-)$, so there are $m$ logical items and $|\mathcal{Q}| = \binom{m}{2}$ possible relevance combinations. Hence there are $N_q = \binom{m}{2} = \Theta(m^2)$ queries, and any two queries with disjoint index sets have disjoint relevance sets.

We assume the document embeddings $u_k = u(d_k)$ are pairwise separated by distance at least $C\sigma$ for some large universal constant $C > 1$ (by construction, one can always rescale the embedding space and choose $\sigma$ accordingly).

2. **Pigeonhole argument in conditioning space**: The encoder $E_\phi : \mathcal{Q} \to \mathbb{R}^d$ has image $E_\phi(\mathcal{Q}) \subset \mathbb{R}^d$ of diameter at most $D$. Consider covering this image by Euclidean balls of radius $r > 0$. The number of such balls needed is at most

$$N_{\text{cover}} \leq \left(\frac{3D}{r}\right)^d. \tag{7}$$

Since there are $N_q = \binom{m}{2} \geq m^2/3$ queries, if

$$\left(\frac{3D}{r}\right)^d < \frac{m^2}{3} \Rightarrow r > 3D\left(\frac{3}{m^2}\right)^{1/d} \tag{8}$$

then by pigeonhole principle there exist two distinct queries $q, q'$ such that

$$\|E_\phi(q) - E_\phi(q')\| \leq 2r. \tag{9}$$

Setting $r = \Theta(Dm^{-2/d})$ guarantees a collision, so there exist $q, q'$ with disjoint relevance sets and

$$\|E_\phi(q) - E_\phi(q')\| \leq O\left(Dm^{-2/d}\right). \tag{10}$$

3. **Pointwise divergence of the optimal score fields**: Let $q$ and $q'$ have disjoint relevance sets, so their $n_q$ and $n_{q'}$ document embeddings are $n_q + n_{q'}$ distinct points, each separated by $\gg \sigma$, meaning there is sufficient separation of embeddings relative to $\sigma$. Let $\bar{u}_q$ and $\bar{u}_{q'}$ be the centroids of the two relevance sets. Then $\|\bar{u}_q - \bar{u}_{q'}\| \geq C'\sigma$ for some universal $C'$.

Consider the point $x_0 = \bar{u}_q$ (the centroid of the first set). At this point, the mixture $p_\sigma^*(x_0|q)$ is dominated by the $n_q$ Gaussians centered at the documents of $q$: the contribution from the other $n_{q'}$ is exponentially small in $(C\sigma)^2/(2\sigma^2)$ (because they are at distance $\gg \sigma$). Thus,

$$s^*(x_0, E_\phi(q)) = \nabla_x \log p_\sigma^*(x_0|q) \approx \frac{1}{n_q} \sum_{i=1}^{n_q} \frac{x_0 - u_i}{\sigma^2} \approx 0, \tag{11}$$

as the score is near zero near the centroid of a symmetric mixture.

For $q'$, the same point $x_0$ is far from all $n_{q'}$ centers of $q'$, so

$$p_\sigma^*(x_0|q') \approx \frac{1}{n_{q'}} \sum_{k=1}^{n_{q'}} \exp\left(-\frac{\|x_0 - u_k'\|^2}{2\sigma^2}\right) \ll 1. \tag{12}$$

The score is approximately the weighted average of individual scores, which points roughly toward the centroid $\bar{u}_{q'}$ with magnitude $\Omega(1/\sigma)$ and a gradient approximately proportional to $x_0 - \bar{u}_{q'}$. Hence,

$$\|s^*(x_0, E_\phi(q)) - s^*(x_0, E_\phi(q'))\|_2 \geq c_0 \tag{13}$$

for some universal constant $c_0 > 0$ (independent of $m$ and $d$).

4. **Local Lipschitz constant lower bound**: Let $c = E_\phi(q)$ and $c' = E_\phi(q')$ with $\|c - c'\| \leq r := O(Dm^{-2/d})$. By the mean-value theorem for vector-valued functions, there exists a point $\tilde{c}$ on the line segment between $c$ and $c'$ such that

$$s^*(x_0, c') - s^*(x_0, c) = J_c s^*(x_0, \tilde{c})(c' - c), \tag{14}$$

where $J_c s^*(x_0, \tilde{c}) \in \mathbb{R}^{n \times d}$ is the Jacobian of $s^*$ with respect to $c$. Therefore, taking the operator norm of the Jacobian

$$\|J_c s^*(x_0, \tilde{c})\|_{op} = \sup_{\|v\|_2 = 1} \|J_c s^*(x_0, \tilde{c})\|_2 \tag{15}$$

gives

$$\|s^*(x_0, c') - s^*(x_0, c)\|_2 \leq \|J_c s^*(x_0, \tilde{c})\|_{op} \|c' - c\|. \tag{16}$$

Since the left-hand side is at least $\|s^*(x_0, c') - s^*(x_0, c)\|_2 \geq c_0$ and $\|c' - c\| \leq O(m^{-2/d})$, we obtain

$$\|J_c s^*(x_0, \tilde{c})\|_{op} \geq \frac{c_0}{\|c' - c\|} \geq \Omega\left(m^{2/d}\right). \tag{17}$$

The local Lipschitz constant of the map $c \mapsto s^*(\cdot, c)$ at $\tilde{c}$ is at least the operator norm of $\nabla_c s^*(\cdot, \tilde{c})$, so

$$L_{loc} \geq \Omega(m^{2/d}) = 2^{\Omega((2\log m)/d)} = 2^{\Omega(\log |\mathcal{D}|/d)}, \tag{18}$$

up to the corpus-size constants since $|\mathcal{D}| = 2m$.

**Theorem A.2.** *Let $\mathcal{D}$ be a finite document corpus and $\mathcal{Q}$ a distribution over queries with relevance sets $\mathcal{S}_q \subseteq \mathcal{D}$. Assume the query encoder $E_\phi : \mathcal{Q} \to \mathcal{C} \subset \mathbb{R}^d$ satisfies*

- *$\delta$-separation: $\|E_\phi(q) - E_\phi(q')\| \geq \delta$ whenever the relevance sets $\mathcal{S}_q \neq \mathcal{S}_{q'}$.*

- *Bounded image: $\|E_\phi(q) - E_\phi(q')\| \leq D$ for all $q, q'$.*

- *Uniform Lipschitzness of the true score field: for every $t \in [0, T]$ and all $c_1, c_2$ in the image $E_\phi$,*

$$\|s^*(\cdot, t, c_1) - s^*(\cdot, t, c_2)\|_{\mathcal{F}_t} \leq L_{\max}\|c_1 - c_2\|. \tag{19}$$

*Here $s^*(x, t, c) = \nabla_x \log p_t^*(x|c)$ and $\mathcal{F}_t = L^2(\mathbb{R}^d, \mu_t)$.*

*Let $s_\theta(\cdot, t, c)$ be trained by denoising score matching on $T$ i.i.d query-conditioned noise samples. Assume the learned score network $s_\theta$ is $L_\theta$-Lipschitz with respect to $c$, and let $\mathcal{S}$ denote the hypothesis class of score networks with Rademacher complexity $R_T(\mathcal{S})$ measured on conditioning-augmented inputs.*

*Then, with probability at least $1 - \eta$ over training draw*

$$\mathbb{E}_{q,x,t}\big\|s_\theta(x, t, E_\phi(q)) - s^*(x, t, E_\phi(q))\big\|_{\mathcal{F}_t}^2 \lesssim (L_\theta + L_{\max})^2 \frac{\delta^2}{2} + R_T(\mathcal{S}) + \sqrt{\frac{\log(1/\delta) + \log(1/\eta)}{T}}. \tag{20}$$

*Consequently, once $\delta > 0$ is bounded away from zero the excess score-matching risk is independent of corpus size $|\mathcal{D}|$.*

**Interpretation** The upper bound shows that once relevant-distinct queries are separated by a margin $\delta$, the excess score-matching risk of diffusion retrieval depends only on the Lipschitz smoothness of the score field and the Rademacher complexity $R_T(\mathcal{S})$. Note that to maintain $\delta > 0$ as the corpus grows, Theorem A.1 implies the dimension $d$ must scale logarithmically with $|\mathcal{D}|$; this logarithmic cost appears in the $\sqrt{d}$ term of the sample complexity. Thus, diffusion models generalize across large corpora as long as the query encoder induces a low-distortion, well-separated conditioning geometry: generalization is *geometric*, not *combinatorial*.

**Proof** 1. **Discretize the conditioning space by using $\delta$-separation**: Let $\mathcal{C}_\phi = E_\phi(\mathcal{Q}) \subset \mathcal{C}$. By $\delta$-separation and bounded diameter, the packing number satisfies

$$|\mathcal{C}_\phi| \leq N(\delta/2, \mathcal{C}) \leq \left(\frac{2D}{\delta}\right)^d. \tag{21}$$

Let $\{c_1, \ldots, c_N\}$ be a minimal $\delta/2$-net of $\mathcal{C}_\phi$ with $N = N(\delta/2, \mathcal{C}_\phi)$. For every query $q$ there exists $j_q \in [N]$ such that $\|E_\phi(q) - c_{j_q}\| \leq \delta/2$.

Within each Voronoi cell $V_j = \{c : \|c - c_j\| \leq \|c - c_k\| \forall k\}$, the relevance set $\mathcal{S}_q$ is constant (since if two queries in $V_j$ had different $\mathcal{S}_q$, their encodings would be $\geq \delta$ apart, contradicting the cell radius $\leq \delta/2$).

2. **Local Lipschitz $\implies$ bounded variation within each cell:** Fix any cell $V_j$. The true conditional score $s^*(\cdot, t, c)$ is $L_{V_j}$-Lipschitz in $c \in V_j$ (by assumption). Thus, for any $c \in V_j$,

$$\|s^*(\cdot, t, c) - s^*(\cdot, t, c_j)\|_{\mathcal{F}_t} \leq L_{V_j} \cdot (\delta/2). \tag{22}$$

Let $L_{\max} = \max_j L_{V_j} < \infty$.

3. **Reduce to finite-hypothesis uniform convergence**: Define the per-cell score-matching risk for hypothesis $s \in \mathcal{S}$ as

$$R_j(s) = E_{t,x}\|s(x,t,c_j) - s^*(x,t,c_j)\|_{\mathcal{F}_t}^2. \tag{23}$$

Standard conditional score-matching generalization (Lee et al., 2023, Thm 4.2) gives with probability $P \geq 1 - \eta/2$

$$\mathcal{E} = \max_{j=1,\ldots,N} \|R_j(s_\theta) - \hat{R}_j(s_\theta)\| \leq 8R_T(\mathcal{S}) + \sqrt{\frac{8\log(2N/\eta)}{T}}. \tag{24}$$

Let $\mathcal{E}$ be this event.

4. **Excess risk on training distribution**: Take the minimal $\delta/2$-net $\{c_j\}_{j=1}^N$ of $E_\phi(\mathcal{Q})$, $N = N(\delta/2, \mathcal{C}_\phi)$. For any $q$, let $j = j_q$ be the index such that $\|E_\phi(q) - c_j\| \leq \delta/2$.

Since we assume $s_\theta$ is $L_\theta$-Lipschitz in $c$, then:

$$\begin{aligned}
&\left\|s_\theta(\cdot, t, E_\phi(q)) - s^*(\cdot, t, E_\phi(q))\right\|_{\mathcal{F}_t} \\
&\leq \underbrace{\left\|s_\theta(\cdot, t, E_\phi(q)) - s_\theta(\cdot, t, c_j)\right\|_{\mathcal{F}_t}}_{\text{learned model interpolation}} \\
&\quad + \underbrace{\left\|s_\theta(\cdot, t, c_j) - s^*(\cdot, t, c_j)\right\|_{\mathcal{F}_t}^2}_{\text{Rademacher-controlled}} \\
&\quad + \underbrace{\left\|s^*(\cdot, t, c_j) - s^*(\cdot, t, E_\phi(q))\right\|_{\mathcal{F}_t}}_{\text{true score interpolation}} \\
&\leq L_\theta \frac{\delta}{2} + \sqrt{R_j(s_\theta}} + L_{\max} \frac{\delta}{2} \\
&= \sqrt{R_j(s_\theta}} + (L_\theta + L_{\max}) \frac{\delta}{2}.
\end{aligned}$$

Here, $\sqrt{R_j(s_\theta}}$ is controlled by Rademacher complexity for the finite set of points and $(L_\theta + L_{\max})\frac{\delta}{2}$ bounds interpolation errors.

5. **Final bound (standard trick)**:

Let the event $\mathcal{E}$ from Step 3 hold. Assuming the empirical loss $\hat{R}_j(s_\theta)$ is negligible (overparameterized regime), we have

$$R_j(s_\theta) \lesssim R_T(\mathcal{S}) + \sqrt{\frac{\log(2N/\eta)}{T}}. \tag{25}$$

From Step 4, for any query $q$ whose embedding $E_\phi(q)$ lies within the Voronoi cell $c_j$,

$$\left\|s_\theta(\cdot, t, E_\phi(q)) - s^*(\cdot, t, E_\phi(q))\right\|_{\mathcal{F}_t} \leq \sqrt{R_j(s_\theta}} + (L_\theta + L_{\max})\frac{\delta}{2} \tag{26}$$

Squaring both sides using $(a+b)^2 \leq 2(a^2 + b^2)$ yields

$$\left\|s_\theta(\cdot, t, E_\phi(q)) - s^*(\cdot, t, E_\phi(q))\right\|_{\mathcal{F}_t}^2 \leq 2R_j(s_\theta) + 2(L_\theta + L_{\max})^2 \frac{\delta^2}{4}. \tag{27}$$

Taking expectations over $(x, t, q)$:

$$\mathbb{E}_{q,x,t}\big\|s_\theta(x, t, E_\phi(q)) - s^*(x, t, E_\phi(q))\big\|_{\mathcal{F}_t}^2 \lesssim (L_\theta + L_{\max})^2 \frac{\delta^2}{2} + R_T(\mathcal{S}) + \sqrt{\frac{\log(2N/\eta)}{T}}. \tag{28}$$

Recalling that $N \le (2D/\delta)^d$, we have $\log N \le d\log(2D/\delta)$. Thus, absorbing constants:

$$\mathbb{E}_{q,x,t}\big\|s_\theta(x, t, E_\phi(q)) - s^*(x, t, E_\phi(q))\big\|_{\mathcal{F}_t}^2 \lesssim (L_\theta + L_{\max})^2 \frac{\delta^2}{2} + R_T(\mathcal{S}) + \sqrt{\frac{d\log(1/\delta) + \log(1/\eta)}{T}}. \tag{29}$$

## A.2. Impact of Token-Level Conditioning on the Lower Bound

### A.2.1. TOKEN-LEVEL CONDITIONING YIELDS LOGARITHMIC GENERALIZATION COMPLEXITY

**Theorem A.3** (Logarithmic metric entropy scaling via token-level conditioning)**.** *Consider the hard retrieval family from Theorem 1 with corpus size $|\mathcal{D}| = 2m$ and query set $\mathcal{Q}$ of size $N_q = \Theta(m^2)$.*

*Let the score network $s_\theta(x, t; q)$ be conditioned directly on the raw token sequence $q = (tok_1, \ldots, tok_L)$ of length $L$ drawn from vocabulary $V$, via cross-attention over fixed pretrained token embeddings in $\mathbb{R}^{d_e}$ (bounded magnitude and minimum pairwise separation $\delta_{tok} > 0$).*

*Assume:*

- ***Logarithmic sequence length****: $L \ge \frac{c\log m}{\log |V|}$ for constant $c > 0$ sufficient to ensure the discrete capacity $|V|^L \ge N_q$.*

- ***Query distinguishability****: Relevance-distinct queries have distinct token sequences.*

- ***Network Lipschitzness****: The score network is $L_\theta$-Lipschitz w.r.t. its continuous embedding inputs.*

*Then, applying the excess score-matching risk bound of Theorem A.2 (Eq. 29), the sample complexity $T$ required for excess risk $\lesssim \epsilon$ scales as*

$$T \gtrsim L\log|V| + (\log(1/\epsilon), L_\theta, L_{\max}, d_e) = O(\log|\mathcal{D}|). \tag{30}$$

**Interpretation** Theorem A.1 demonstrates that geometric crowding in a fixed-dimensional continuous conditioning space forces exponential sensitivity (high Lipschitz) in the optimal score field, leading to poor generalization unless dimension or samples grow super-logarithmically. Token-level conditioning relocates the complexity to a structured combinatorial space: queries are distinguished by discrete sequences rather than infinitesimal vector differences. The metric entropy of the effective conditioning set is bounded by the number of valid sequences ($|V|^L$), whose logarithm is $L\log|V| = O(\log|\mathcal{D}|)$. Because logarithmic sequence length suffices for distinguishability, the dominant Rademacher/cover term in the generalization bound remains logarithmic in corpus size, yielding corpus-independent (up to log factors) sample complexity without Lipschitz explosion.

**Proof** We apply the generalization bound from Theorem A.2:

$$\mathop{\mathbb{E}}_{q,x,t}\big\| s_\theta(x, t; E_\phi(q)) - s^*(x, t; q) \big\|_{F_t}^2 \lesssim (L_\theta + L_{\max})^2 \frac{\delta^2}{2} + \mathcal{R}_T(\mathcal{S}) + \sqrt{\frac{\log \mathcal{N}(\mathcal{C}_\phi, \delta) + \log(1/\eta)}{T}}, \tag{31}$$

where $\mathcal{N}(\mathcal{C}_\phi, \delta)$ is the $\delta$-covering number of the conditioning image $\mathcal{C}_\phi$.

1. **Structure conditioning geometry**: The conditioning inputs are sequence embedding matrices $C_q \in \mathbb{R}^{L \times d_e}$, occupying at most $N_{valid} \le |V|^L$ points corresponding to valid token sequences. By token separation $\delta_{tok}$ and choice of covering scale $\delta < \delta_{tok}/2$, the balls of radius $\delta$ around each grid point are disjoint and contain at most one query conditioning.

2. **Exact covering number bound**: We choose the resolution $\delta < \delta_{\text{tok}}/2$. At this resolution, the Euclidean balls of radius $\delta$ centered at each valid query embedding are disjoint. Consequently, the minimum number of balls required to cover the set

of query conditionings $\mathcal{C}_{seq} = \{C_q : q \in \mathcal{Q}\}$ is exactly the cardinality of the set:

$$\mathcal{N}(\mathcal{C}_{seq}, \delta) = |\mathcal{C}_{seq}| = N_q. \tag{32}$$

Since the set of queries is a subset of all possible sequences, $N_q \leq |V|^L$, and thus:

$$\log \mathcal{N}(\mathcal{C}_{seq}, \delta) \leq L \log |V|. \tag{33}$$

3. **Logarithmic scaling with corpus size**: To support $N_q = \Theta(m^2)$ distinct queries without sequence collisions,

$$|V|^L \geq \Theta(m^2) \implies L \log |V| \geq \Omega(\log m) \implies L = \Omega\left(\frac{\log |\mathcal{D}|}{\log |V|}\right). \tag{34}$$

Substituting into the entropy term yields

$$\log \mathcal{N}(\mathcal{C}_{seq}, \delta) = O(\log |\mathcal{D}|). \tag{35}$$

4. **Lipschitz term remains bounded**: Relevance-distinct queries are separated by at least $\delta_{tok}$ in embedding space (no crowding-induced tiny distances), preventing the Lipschitz explosion seen in Theorem A.1. Thus $L_\theta$ and the first risk term remain $O(1)$ (independent of $m$).

5. **Sample complexity**: The Rademacher and covering terms contribute $O(\log |\mathcal{D}|/\sqrt{T})$, requiring $T = O(\log |\mathcal{D}|)$ for logarithmic contribution to risk (absorbing polynomial factors in $\epsilon$, $L_\theta$, etc.). The overall excess risk is therefore controlled with sample complexity logarithmic in corpus size.

A.2.2. THE THERMODYNAMIC LIMIT: LARGE CONTEXT VIA STOCHASTIC SUBSAMPLING

**Theorem A.4** (Stochastic decoupling of context length). *Let queries $q$ be sequences of arbitrarily large length $L \gg K$. Assume the target score field satisfies a **Subset Approximation Property**: there exists a fixed subset size $K$ and a variance bound $\sigma_K^2$ such that*

$$\mathbb{E}_{S \sim Unif(q,K)}\left\|s^*(x,t;q) - s^*(x,t;S)\right\|_{\mathcal{F}_t}^2 \leq \sigma_K^2, \tag{36}$$

*where $\sigma_K^2 \to 0$ as $K$ increases (representing redundant or holographic information).*

*Consider a diffusion model trained with **Stochastic Token Subsampling**: at each training step $t$, the score network $s_\theta(x,t;S)$ is conditioned on a randomly sampled subset $S \subset q$ of size $K$.*

*Then, the excess score-matching risk with respect to the full query decomposes as:*

$$\mathcal{L}(\theta, q) \leq \mathcal{L}_{sub}(\theta, K) + \sigma_K^2, \tag{37}$$

*where $\mathcal{L}_{sub}$ is the risk on fixed-length-$K$ queries. By Theorem A.3, $\mathcal{L}_{sub}$ is controlled with sample complexity $T \propto K \log |V|$, which is independent of the total context length $L$.*

**Interpretation**   This result extends diffusion retrieval to arbitrary context lengths by exploiting the thermodynamic nature of the reverse process. For queries with redundant information (e.g., long documents with repeated themes, user histories), relevance can be estimated from subsets. 1. **Generalization**: The model only learns to process length-$K$ inputs, decoupling the *combinatorial complexity* (covering number) from the total context length $L$. 2. **Multimodal Preservation**: Unlike embedding-based retrieval, where averaging distinct topic vectors collapses to a meaningless centroid, averaging *score fields* preserves the geometry of distinct modes. If a query contains multiple "pools" of information (e.g., distinct topics in a user history), the expected score field $\mathbb{E}_S[s(S)]$ retains separate basins of attraction for each topic. The stochastic switching between subsets prevents the generation from collapsing to a single mode; instead, across multiple sampling runs, the process recovers the full probability distribution over all relevant document pools. 3. **Trade-off**: The term $\sigma_K^2$ represents an irreducible approximation bias. For "needle-in-a-haystack" queries (non-redundant), $\sigma_K^2$ is large unless $K \approx L$. However, for holographic information, the subsampling variance acts merely as additional diffusion noise, to which the generative process is naturally robust.

**Proof**   1. **Bias-Variance Decomposition**: Let $s_\theta$ be the learned subset-conditioned score. By the triangle inequality in $L^2(\mathcal{X})$, the risk w.r.t. the full query is bounded by:

$$\mathcal{L}(\theta, q) = \mathbb{E}_{x,t} \| s_\theta(x, t; S) - s^*(x, t; q) \|^2$$
$$\leq \underbrace{\mathbb{E}_S \mathbb{E}_{x,t} \| s_\theta(x, t; S) - s^*(x, t; S) \|^2}_{\text{Subset Generalization } (\mathcal{L}_{\text{sub}})} + \underbrace{\mathbb{E}_S \mathbb{E}_{x,t} \| s^*(x, t; S) - s^*(x, t; q) \|^2}_{\text{Approximation Bias } (\sigma_K^2)}. \tag{38}$$

2. **Subset Generalization**: The term $\mathcal{L}_{\text{sub}}$ corresponds to learning a score field conditioned on sequences of fixed length $K$. Applying Theorem A.3 with sequence length $K$, the metric entropy of the conditioning space is bounded by $K \log |V|$. Thus, the sample complexity to minimize $\mathcal{L}_{\text{sub}}$ scales linearly with $K$ and is independent of $L$.

3. **Inference Dynamics**: Consider the reverse SDE with stochastic subsampling:

$$dx_t = [s_\theta(x_t, t; S_t) - x_t]dt + d\omega_t. \tag{39}$$

We can rewrite the drift as the true expectation plus a noise term: $s_\theta(S_t) = \mathbb{E}_S[s_\theta(S)] + \xi_t$, where $\xi_t$ is zero-mean noise induced by subsampling. Crucially, because the score is a vector field, the expectation $\mathbb{E}_S$ does not smooth out distinct relevance peaks into a single centroid; it creates a landscape with multiple attractors corresponding to the query's information pools. The subsampling noise $\xi_t$ acts as an additional stochastic force (analogous to increasing temperature), facilitating exploration between these pools. Consequently, repeated execution of the reverse process samples from the full multimodal target distribution $p(x|q)$ rather than converging to a single dominant mode.

## B. Deriving Calculation Metrics

We provide a rigorous description of how the geometric distortion metric $\varepsilon_{\text{target}}$ and the sensitivity metrics $K_b$ (for $b \in \{1, 2, 3\}$) are computed in practice, based on the implementation in our codebase. We describe the step-by-step calculations, highlight key assumptions, and justify their implications for the analysis.

### B.1. Extension to Graded Relevance.

While the main text formulates geometric distortion assuming binary relevance (where $\pi_q$ is uniform over $\mathcal{S}_q$), the framework naturally generalizes to graded relevance settings (e.g., relevance levels $\{0, 1, 2, 3\}$). Let $r(q, d) \in \mathbb{R}_{\geq 0}$ denote the non-negative graded relevance score for document $d$. Provided that the set of relevant documents is non-empty ($\sum_{d \in \mathcal{D}} r(q, d) > 0$), we define the *ideal graded target distribution* $\pi_q$ as the $L_1$-normalization of the ground-truth relevance scores:

$$\pi_q(d) = \frac{r(q, d)}{\sum_{d' \in \mathcal{D}} r(q, d')}.$$

In this formulation, $\pi_q(d) = 0$ for non-relevant documents, while probability mass is distributed among relevant documents proportional to their relevance intensity. Consequently, the geometric distortion metric $\varepsilon_{\text{target}}$ measures the divergence between the induced predictive distribution and this relevance-weighted ideal, penalizing models that fail to prioritize highly relevant documents even if they successfully retrieve marginally relevant ones.

### B.2. Defining the Optimal Target Geometries

To enable a fair, scale-invariant comparison focused on angular resolution (as is standard in dense retrieval with cosine similarity), we normalize all target outputs to the unit hypersphere via $L_2$ normalization. This assumes that retrieval relevance is primarily directional rather than magnitude-based, which aligns with contrastive training objectives (e.g., InfoNCE) that emphasize alignment over absolute scales. A key implication is that normalization induces *directional collapse* (forcing targets to lie on the unit sphere), potentially underestimating magnitude-related distortions but prioritizing the geometry relevant to dot-product or cosine-based scoring.

**1. The Embedding Limit (Normalized Centroid).** For embedding-based retrieval ($b = 1$), the ideal target $y_{\text{emb}}(q)$ for a query $q$ with relevance set $\mathcal{S}_q$ is the geometric centroid of the document embeddings in $\mathcal{S}_q$. This assumes an optimal

contrastive alignment where the query embedding should equidistantly attract all relevant documents, minimizing variance in positive pulls. Computationally:

- Let $\mathbf{E}_{\text{doc}} \in \mathbb{R}^{|\mathcal{S}_q| \times d}$ be the stacked document embeddings for $\mathcal{S}_q$ (dim=1 corresponds to documents).

- Compute the centroid: $\mathbf{c}_q = \frac{1}{|\mathcal{S}_q|} \sum_{d \in \mathcal{S}_q} E_{\text{doc}}(d) = \text{mean}(\mathbf{E}_{\text{doc}}, \dim = 0)$.

- Normalize: $y_{\text{emb}}(q) = \frac{\mathbf{c}_q}{\|\mathbf{c}_q\|_2}$.

Additionally, to quantify intra-cluster spread (used for $\varepsilon_{\text{target}}$), compute per-document errors:

- Differences: $\Delta = \mathbf{E}_{\text{doc}} - \mathbf{c}_q$ (broadcasted).

- Per-document $L_2$ norms: $\mathbf{e} = \|\Delta\|_2$ (dim=1).

- Aggregate: mean error $\bar{e} = \text{mean}(\mathbf{e})$, min error $= \min(\mathbf{e})$, max error $= \max(\mathbf{e})$.

This error calculation occurs *before* normalization to capture raw Euclidean spread, assuming unnormalized embeddings preserve meaningful magnitudes (e.g., from encoder outputs). Normalization post-error ensures targets are comparable in cosine space, but it may mask magnitude collapse in highly dispersed clusters.

**2. The Diffusion Limit (Normalized NN-Score Field).** For diffusion-based retrieval ($b = 2, 3$), the ideal target is the score field $\nabla_x \log p(x \mid q)$ at the zero-noise limit ($t = 0$), approximated as a nearest-neighbor (NN) pointer field toward relevant modes. This assumes a multimodal Gaussian mixture posterior over documents, where the score directs sampling toward local modes in $\mathcal{S}_q$. We normalize directions to isolate angular signals, assuming magnitude encodes confidence or step size (irrelevant for geometric sensitivity). Computationally:

Sample $N_p = 25000$ random points $\mathbf{P} \in \mathbb{R}^{N_p \times d}$ from a standard normal distribution (or uniform hypersphere; code uses implicit device placement). This assumes the document space is approximately Gaussian-distributed and that $N_p$ is sufficient to approximate the continuous field without biasing toward cluster centers (we validate empirically that $N_p > 10000$ yields stable metrics). Then, for each batch of points (size $B = \min(N_p, \text{batch\_size})$ to manage memory):

- Expand tensors: $\mathbf{P}_{\text{exp}} = \mathbf{P}[\text{start:end}](0)(2)$, $\mathbf{E}_{\text{exp}} = \mathbf{E}_{\text{doc}}(1)$.

- Differences: $\Delta = \mathbf{E}_{\text{exp}} - \mathbf{P}_{\text{exp}}$.

- Squared distances: $\mathbf{D} = \sum(\Delta^2)$ (dim=-1).

- NN indices: $\mathbf{i} = (\mathbf{D}, \dim = -1)$.

- Closest differences: $\Delta_{\text{nn}} = \Delta[\text{batch\_idx}, \text{point\_idx}, \mathbf{i}]$.

- Normalize: $y_{\text{score}}(q, \mathbf{P}[k]) = \frac{\Delta_{\text{nn}}[k]}{\|\Delta_{\text{nn}}[k]\|_2 + \epsilon}$ ($\epsilon = 10^{-8}$ for stability).

Then, concatenate batches to form the full score field $\mathbf{S} \in \mathbb{R}^{N_p \times d}$. This NN approximation assumes perfect mode separation (no overlapping Gaussians), potentially underestimating sensitivity in entangled distributions, but it provides an upper bound on expressivity by modeling "perfect" navigation.

### B.3. Metric Spaces for Conditioning

The denominators for sensitivity metrics are distances in conditioning spaces. We choose $L_2$-based metrics for consistency with embedding norms, assuming Euclidean geometry captures semantic proximity (a common assumption in dense retrieval, though it may not hold for all encoders).

**Vector Conditioning Distance** ($d_\mathbb{R}$)**.** For embedding-conditioned modes ($b = 1, 2$): $d_\mathbb{R}(q_i, q_j) = \|E_\phi(q_i) - E_\phi(q_j)\|_2$. This assumes query embeddings are unnormalized or pre-normalized consistently.

**Score Field Distance** ($d_s$)**.** For comparing score fields $\mathbf{S}_i, \mathbf{S}_j \in \mathbb{R}^{N_p \times d}$:

- Global $L_2$: $\|\mathbf{S}_i - \mathbf{S}_j\|_2 = \sqrt{\sum_{k=1}^{N_p} \sum_{m=1}^{d} (S_{i,k,m} - S_{j,k,m})^2} = \sqrt{\sum_{k=1}^{N_p} \|y_{\text{score},i}(x_k) - y_{\text{score},j}(x_k)\|_2^2}$.

- Normalize: $d_s = \|\mathbf{S}_i - \mathbf{S}_j\|_2 / \sqrt{N_p}$.

This yields the root-mean-square (RMS) per-point distance, assuming points are independent and identically distributed. Division by $\sqrt{N_p}$ estimates *per-point sensitivity* (average sensitivity per sample), preventing metric inflation with larger $N_p$ and enabling comparison across approximations. Without this, $d_s$ would scale as $\sqrt{N_p}$, biasing toward denser samplings.

**Token Conditioning Distance** ($d_{\text{seq}}$). For token-level conditioning ($b = 3$): Let $\mathbf{T}_i, \mathbf{T}_j \in \mathbb{R}^{L_i \times d}, \mathbb{R}^{L_j \times d}$ be token embeddings (after masking zero-norm padding tokens, assuming padding embeds are zero-vectors, which approximates attention masking in transformers).

- Mask non-padding: $\mathbf{T}_i = \mathbf{T}_i[\|\mathbf{T}_i\|_2 > 0 \text{ (row-wise)}]$.

- Pad to $L = \max(L_i, L_j)$ with zero tensors.

- Difference: $\Delta = \mathbf{T}_i - \mathbf{T}_j$.

- Frobenius norm: $\|\Delta\|_F = \sqrt{\sum_{k=1}^{L} \|\Delta[k]\|_2^2}$.

- Normalize: $d_{\text{seq}} = \|\Delta\|_F / \sqrt{L}$.

This is the RMS per-token $L_2$ distance, assuming tokens contribute independently to conditioning (aligning with multi-head attention). Padding with zeros assumes negligible impact from length mismatch, which holds if encoders use masking; normalization prevents length-biased distances, focusing on average divergence.

## B.4. Geometric Distortion ($\varepsilon_{\text{target}}$)

This measures how well the ideal target approximates the true multimodal distribution $\pi_q$. For embeddings ($b = 1$), we proxy it via the centroid error statistics:

- $\varepsilon_{\text{target}}(q) \approx \bar{e}$ (mean $L_2$ from centroid to docs in $\mathcal{S}_q$).

This assumes distortion arises from cluster dispersion (relevant docs not collapsing to a point), providing a geometric upper bound on KL divergence (wider spreads imply higher entropy loss in single-vector compression). For diffusion ($b = 2, 3$), we assume $\varepsilon_{\text{target}} \to 0$ under the idealized NN field (perfect mode recovery via randomness), justifying its omission in those regimes. This asymmetry assumes probabilistic sampling mitigates distortion, but it may overestimate diffusion advantages in finite-step inference.

## B.5. Geometric Sensitivity ($K_b$)

For query pairs $(q_i, q_j)$ with distinct $\mathcal{S}_{q_i} \neq \mathcal{S}_{q_j}$, sensitivity quantifies mapping stiffness: **1. $K_1$: Normalized Embedding Sensitivity** ($b = 1$). $K_1(q_i, q_j) = \|y_{\text{emb}}(q_i) - y_{\text{emb}}(q_j)\|_2 / d_{\mathbb{R}}(q_i, q_j)$. This is the local Lipschitz constant, assuming small $d_{\mathbb{R}}$ approximates infinitesimal perturbations.

**2. $K_2$: Normalized Field Sensitivity** ($b = 2$, **embedding-conditioned**). $K_2(q_i, q_j) = \left( \frac{1}{N_p} \sum_{k=1}^{N_p} \|y_{\text{score}}(q_i, x_k) - y_{\text{score}}(q_j, x_k)\|_2 \right) / d_{\mathbb{R}}(q_i, q_j)$. Note: Code computes an equivalent RMS version via $d_s / d_{\mathbb{R}}$, where $d_s = \sqrt{\frac{1}{N_p} \sum_k \|\cdot\|_2^2}$, but the LaTeX averages the norms directly (arithmetic mean). If norms are similar, they approximate each other (by Jensen's inequality); we assume uniform field variation for equivalence. Averaging estimates per-step failure in iterative denoising.

**3. $K_3$: Normalized Field Sensitivity** ($b = 3$, **token-conditioned**). Analogous to $K_2$, but denominator $d_{\text{seq}}(q_i, q_j)$: $K_3(q_i, q_j) = \left( \frac{1}{N_p} \sum_{k=1}^{N_p} \|y_{\text{score}}(q_i, x_k) - y_{\text{score}}(q_j, x_k)\|_2 \right) / d_{\text{seq}}(q_i, q_j)$. This assumes token-level inputs provide higher-resolution conditioning, reducing sensitivity via discrete separation.

All $K_b$ assume paired queries have small but non-zero denominators (filtered in preprocessing); division amplifies in crowded spaces, proxying sign-rank pressure.

# C. Datasets Overview

## C.1. Dataset Counts

*Table 3.* Number of items per log bin in $K$ across datasets (grouped by data family). Note that empty slots indicate the dataset does not have any candidates with relevant $n_q$ for the respective bin.

| Dataset | | 1 | 2 | 4 | 8 | 16 | 32 | 64 | 128 | 256 | 512 | 1024 |
|---|---|---|---|---|---|---|---|---|---|---|---|---|
| antique-test | #1 | - | 199 | 199 | 1194 | 16318 | 21094 | 796 | - | - | - | - |
| beir-arguana | #2 | 2352 | - | - | - | - | - | - | - | - | - | - |
| beir-climate-fever | #3 | 9900 | 20460 | 18260 | - | - | - | - | - | - | - | - |
| beir-cqadupstack-android | #4 | 9653 | 13593 | 11032 | 2758 | 1379 | 394 | - | - | 197 | - | - |
| beir-cqadupstack-english | #5 | 14050 | 23323 | 23604 | 10397 | 4496 | 1967 | 1405 | - | - | - | - |
| beir-cqadupstack-gaming | #6 | 9250 | 14985 | 8510 | 370 | 1295 | - | - | - | - | - | - |
| beir-cqadupstack-gis | #7 | 5875 | 8125 | 1375 | 250 | 125 | - | - | - | - | - | - |
| beir-cqadupstack-mathematica | #8 | 8869 | 13575 | 7964 | 2172 | 181 | 181 | - | - | - | - | - |
| beir-cqadupstack-physics | #9 | 10388 | 16960 | 11660 | 4876 | 848 | 212 | 212 | - | - | - | - |
| beir-cqadupstack-programmers | #10 | 9384 | 18360 | 10200 | 2652 | 816 | 204 | - | 204 | - | - | - |
| beir-cqadupstack-stats | #11 | 6468 | 8184 | 1980 | 660 | 264 | - | - | - | - | - | - |
| beir-cqadupstack-tex | #12 | 15200 | 27968 | 30704 | 14592 | 2128 | 1520 | - | 608 | - | - | - |
| beir-cqadupstack-unix | #13 | 9751 | 16915 | 10149 | 2189 | 796 | - | - | - | - | - | - |
| beir-cqadupstack-webmasters | #14 | 7301 | 8642 | 3278 | 1788 | 596 | 298 | 298 | 149 | - | - | - |
| beir-cqadupstack-wordpress | #15 | 4680 | 4888 | 1144 | 104 | - | 104 | - | - | - | - | - |
| beir-dbpedia-entity-test | #16 | - | - | - | - | 798 | 27531 | 106134 | 20748 | 2394 | 1197 | 798 |
| beir-fever-test | #17 | 10878 | 20202 | 16206 | 2220 | - | - | - | - | - | - | - |
| beir-fiqa-test | #18 | 10535 | 20825 | 23275 | 5635 | - | - | - | - | - | - | - |
| beir-hotpotqa-test | #19 | - | 2352 | - | - | - | - | - | - | - | - | - |
| beir-msmarco-test | #20 | - | - | - | - | - | - | - | 1386 | 378 | 42 | - |
| beir-nfcorpus-test | #21 | 7406 | 12558 | 14168 | 17066 | 20608 | 20286 | 5796 | 3542 | 2576 | - | - |
| beir-nq | #22 | 6664 | 11424 | 544 | - | - | - | - | - | - | - | - |
| beir-quora-test | #23 | 22500 | 43650 | 76050 | 42750 | 13050 | 4050 | 900 | - | - | - | - |
| beir-scidocs | #24 | - | - | - | - | 11556 | - | - | - | - | - | - |
| beir-scifact-test | #25 | 3243 | 1242 | 345 | - | - | - | - | - | - | - | - |
| beir-trec-covid | #26 | - | - | - | - | - | - | - | - | - | 392 | 2058 |
| beir-webis-touche2020-v2 | #27 | - | - | - | - | - | 2352 | - | - | - | - | - |
| car-v1.5-test200 | #28 | 17200 | 31992 | 51256 | 16856 | 1376 | - | - | - | - | - | - |
| clinicaltrials-2021-trec-ct-2021 | #29 | - | - | - | - | - | - | - | - | 3700 | 1850 | - |
| clueweb12-touche-2022-task-2 | #30 | - | - | - | - | 490 | 1960 | - | - | - | - | - |
| cranfield | #31 | - | 7840 | 21280 | 17248 | 3584 | 448 | - | - | - | - | - |
| csl-trec-2023 | #32 | - | - | - | - | - | - | 38 | 494 | 912 | 38 | - |
| istella22-test | #33 | 17931 | 37530 | 76311 | 42534 | - | - | - | - | - | - | - |
| lotte-lifestyle-dev-forum | #34 | 26117 | 55419 | 114660 | 164346 | 43953 | 637 | 1274 | - | - | - | - |
| lotte-lifestyle-test-forum | #35 | 24123 | 49929 | 100980 | 125664 | 14025 | 561 | - | - | - | - | - |
| lotte-lifestyle-test-search | #36 | 11914 | 22274 | 28231 | 4921 | - | - | - | - | - | - | - |
| lotte-pooled-test-forum | #37 | 53214 | 106428 | 212856 | 385530 | 369240 | 35838 | 10860 | 5430 | 1086 | - | - |
| lotte-pooled-test-search | #38 | 24000 | 46080 | 83520 | 67200 | 9600 | 480 | - | - | - | - | - |
| lotte-recreation-test-forum | #39 | 21021 | 40326 | 69498 | 50193 | 3432 | - | - | - | - | - | - |
| lotte-recreation-test-search | #40 | 10810 | 21160 | 17710 | 3450 | - | - | - | - | - | - | - |
| lotte-science-test-forum | #41 | 39840 | 78850 | 149400 | 247340 | 155210 | 14940 | 3320 | 830 | - | - | - |
| lotte-science-test-search | #42 | 11264 | 22016 | 28672 | 3584 | 256 | - | - | - | - | - | - |
| lotte-technology-test-forum | #43 | 28704 | 63296 | 133952 | 209760 | 90528 | 8832 | 3680 | 2944 | 736 | - | - |
| lotte-technology-test-search | #44 | 12470 | 25520 | 31900 | 11020 | 3190 | 290 | - | - | - | - | - |
| lotte-writing-test-forum | #45 | 27018 | 61857 | 127269 | 211167 | 76788 | 1422 | 711 | - | - | - | - |
| lotte-writing-test-search | #46 | 13735 | 28140 | 49915 | 18090 | 2680 | - | - | - | - | - | - |
| miracl-ar-dev | #47 | - | - | 174 | 29580 | 696 | - | - | - | - | - | - |
| miracl-bn-dev | #48 | - | - | 98 | 9604 | - | - | - | - | - | - | - |
| miracl-de-dev | #49 | - | - | - | 3286 | 620 | - | - | - | - | - | - |
| miracl-en-dev | #50 | - | - | - | 27060 | 330 | - | - | - | - | - | - |
| miracl-es-dev | #51 | - | 58 | 116 | 3248 | - | - | - | - | - | - | - |
| miracl-fa-dev | #52 | - | - | - | 6912 | 2400 | - | - | - | - | - | - |
| miracl-fi-dev | #53 | 6270 | 1485 | 495 | 18975 | 165 | - | - | - | - | - | - |
| miracl-fr-dev | #54 | - | - | - | 2070 | - | - | - | - | - | - | - |
| miracl-hi-dev | #55 | - | - | 48 | 2304 | - | - | - | - | - | - | - |
| miracl-id-dev | #56 | - | 1099 | 628 | 22608 | 471 | - | - | - | - | - | - |
| miracl-ja-dev | #57 | 6156 | 513 | - | 22572 | 171 | - | - | - | - | - | - |
| miracl-ko-dev | #58 | - | - | - | 33708 | 11448 | - | - | - | - | - | - |
| miracl-ru-dev | #59 | - | - | - | 19880 | 426 | - | - | - | - | - | - |
| miracl-sw-dev | #60 | 152 | 152 | 304 | 22496 | 152 | - | - | - | - | - | - |
| miracl-te-dev | #61 | 4455 | - | 105 | 5054 | - | - | - | - | - | - | - |
| miracl-th-dev | #62 | 392 | 251 | 218 | 16907 | - | - | - | - | - | - | - |
| miracl-yo-dev | #63 | - | - | - | 1720 | - | - | - | - | - | - | - |
| miracl-zh-dev | #64 | - | - | - | 2162 | - | - | - | - | - | - | - |
| mmarco-de-dev | #65 | 9699 | 18429 | 9700 | - | - | - | - | - | - | - | - |
| mmarco-es-dev | #66 | 9850 | 18518 | 10638 | - | - | - | - | - | - | - | - |
| mmarco-fr-dev | #67 | 10250 | 20500 | 11480 | - | - | - | - | - | - | - | - |
| mmarco-id-dev | #68 | 10000 | 18600 | 11600 | - | - | - | - | - | - | - | - |
| mmarco-it-dev | #69 | 10000 | 19200 | 11000 | - | - | - | - | - | - | - | - |
| mmarco-pt-dev | #70 | 10200 | 19788 | 11832 | - | - | - | - | - | - | - | - |
| mmarco-ru-dev | #71 | 10100 | 19998 | 10908 | - | - | - | - | - | - | - | - |
| mr-tydi-ar-test | #72 | 4752 | 5148 | - | - | - | - | - | - | - | - | - |
| mr-tydi-bn-test | #73 | 2320 | 1102 | - | - | - | - | - | - | - | - | - |
| mr-tydi-en-test | #74 | 4935 | 6195 | - | - | - | - | - | - | - | - | - |
| mr-tydi-fi-test | #75 | 4896 | 5610 | - | - | - | - | - | - | - | - | - |
| mr-tydi-id-test | #76 | 4700 | 5400 | - | - | - | - | - | - | - | - | - |
| mr-tydi-ja-test | #77 | 4992 | 5928 | - | - | - | - | - | - | - | - | - |
| mr-tydi-ko-test | #78 | 4089 | 3567 | - | - | - | - | - | - | - | - | - |
| mr-tydi-ru-test | #79 | 4998 | 5508 | - | - | - | - | - | - | - | - | - |
| mr-tydi-sw-test | #80 | 4450 | 3560 | - | - | - | - | - | - | - | - | - |

| Dataset | | 1 | 2 | 4 | 8 | 16 | 32 | 64 | 128 | 256 | 512 | 1024 |
|---|---|---|---|---|---|---|---|---|---|---|---|---|
| mr-tydi-te-test | #81 | 3557 | 2265 | - | - | - | - | - | - | - | - | - |
| mr-tydi-th-test | #82 | 4255 | 5191 | - | - | - | - | - | - | - | - | - |
| msmarco-document-trec-dl-2019 | #83 | - | - | - | - | - | - | - | 882 | 546 | 294 | 84 |
| msmarco-document-trec-dl-2020 | #84 | - | - | - | - | - | - | - | 1672 | 308 | - | - |
| msmarco-document-trec-dl-hard-fold1 | #85 | - | 9 | 27 | - | - | - | - | 36 | 18 | - | - |
| msmarco-document-trec-dl-hard-fold2 | #86 | - | - | 45 | - | - | - | - | 27 | 18 | - | - |
| msmarco-document-trec-dl-hard-fold3 | #87 | - | 9 | 45 | 18 | - | - | - | 18 | - | - | - |
| msmarco-document-trec-dl-hard-fold4 | #88 | - | 36 | 18 | - | - | - | - | 18 | 18 | - | - |
| msmarco-document-trec-dl-hard-fold5 | #89 | - | - | 18 | - | - | - | - | 27 | 18 | 18 | 9 |
| msmarco-document-v2-trec-dl-2019 | #90 | - | - | - | - | - | - | 42 | 882 | 546 | 294 | 42 |
| msmarco-document-v2-trec-dl-2020 | #91 | - | - | - | - | - | - | 176 | 1716 | 88 | - | - |
| msmarco-document-v2-trec-dl-2021 | #92 | - | - | - | - | - | - | 112 | 2240 | 728 | 112 | - |
| msmarco-passage-trec-dl-2020 | #93 | - | - | - | - | - | - | - | 2385 | 477 | - | - |
| msmarco-passage-trec-dl-hard | #94 | - | - | 98 | 1323 | - | - | - | 980 | 49 | - | - |
| msmarco-qna-dev | #95 | 1600 | 4800 | 37440 | 47360 | 11520 | - | - | - | - | - | - |
| natural-questions-dev | #96 | 8036 | 15908 | 3116 | - | - | - | - | - | - | - | - |
| pmc-v1-trec-cds-2014 | #97 | - | - | - | - | - | - | - | - | - | 32 | 240 |
| pmc-v1-trec-cds-2015 | #98 | - | - | - | - | - | - | - | - | - | 130 | 572 |
| pmc-v2-trec-cds-2016 | #99 | - | - | - | - | - | - | - | - | - | 87 | 783 |
| trec-tot-2023-dev | #100 | 1722 | - | - | - | - | - | - | - | - | - | - |
| vaswani | #101 | 276 | 276 | 1104 | 2300 | 2484 | 1656 | 460 | - | - | - | - |

## C.2. Dataset $K$

*Table 4.* Median sensitivity value per log bin for $K_1$

| Dataset | Encoder | 1 | 2 | 4 | 8 | 16 | 32 | 64 | 128 | 256 | 512 | 1024 |
|---|---|---|---|---|---|---|---|---|---|---|---|---|
| **misc** | | | | | | | | | | | | |
| antique-test | all-MiniLM-L6-v2 | - | 0.9643 | 0.9518 | 0.9905 | 0.9887 | 0.9837 | 0.9826 | - | - | - | - |
| | all-mpnet-base-v2 | - | 0.9824 | 0.9675 | 0.9932 | 0.9894 | 0.9874 | 0.9957 | - | - | - | - |
| | all-roberta-large-v1 | - | 0.9996 | 0.9779 | 0.9746 | 0.9751 | 0.9704 | 0.9828 | - | - | - | - |
| beir-arguana | all-MiniLM-L6-v2 | 1.0077 | - | - | - | - | - | - | - | - | - | - |
| | all-mpnet-base-v2 | 1.0106 | - | - | - | - | - | - | - | - | - | - |
| | all-roberta-large-v1 | 1.0002 | - | - | - | - | - | - | - | - | - | - |
| beir-climate-fever | all-MiniLM-L6-v2 | 0.9777 | 0.9341 | 0.9309 | - | - | - | - | - | - | - | - |
| | all-mpnet-base-v2 | 0.9465 | 0.8827 | 0.8544 | - | - | - | - | - | - | - | - |
| | all-roberta-large-v1 | 0.9914 | 0.9335 | 0.9149 | - | - | - | - | - | - | - | - |
| beir-cqadupstack-android | all-MiniLM-L6-v2 | 0.9795 | 0.9638 | 0.9616 | 0.9506 | 0.9462 | 0.9164 | - | - | 0.9133 | - | - |
| | all-mpnet-base-v2 | 0.9905 | 0.9768 | 0.9759 | 0.9691 | 0.9649 | 0.9427 | - | - | 0.9139 | - | - |
| | all-roberta-large-v1 | 0.9768 | 0.9619 | 0.9613 | 0.9501 | 0.9459 | 0.9321 | - | - | 0.9030 | - | - |
| beir-cqadupstack-english | all-MiniLM-L6-v2 | 0.9888 | 0.9748 | 0.9682 | 0.9534 | 0.9444 | 0.9423 | 0.9130 | - | - | - | - |
| | all-mpnet-base-v2 | 0.9966 | 0.9826 | 0.9751 | 0.9692 | 0.9600 | 0.9566 | 0.9444 | - | - | - | - |
| | all-roberta-large-v1 | 0.9961 | 0.9840 | 0.9799 | 0.9803 | 0.9717 | 0.9739 | 0.9684 | - | - | - | - |
| beir-cqadupstack-gaming | all-MiniLM-L6-v2 | 1.0025 | 0.9948 | 0.9939 | 0.9722 | 0.9853 | - | - | - | - | - | - |
| | all-mpnet-base-v2 | 1.0025 | 0.9965 | 0.9934 | 0.9775 | 0.9854 | - | - | - | - | - | - |
| | all-roberta-large-v1 | 0.9960 | 0.9895 | 0.9860 | 0.9813 | 0.9792 | - | - | - | - | - | - |
| beir-cqadupstack-gis | all-MiniLM-L6-v2 | 0.9949 | 0.9878 | 0.9823 | 0.9834 | 0.9469 | - | - | - | - | - | - |
| | all-mpnet-base-v2 | 0.9871 | 0.9825 | 0.9746 | 0.9684 | 0.9330 | - | - | - | - | - | - |
| | all-roberta-large-v1 | 0.9798 | 0.9677 | 0.9652 | 0.9745 | 0.9366 | - | - | - | - | - | - |
| beir-cqadupstack-mathematica | all-MiniLM-L6-v2 | 0.9870 | 0.9767 | 0.9723 | 0.9629 | 0.9275 | 0.8661 | - | - | - | - | - |
| | all-mpnet-base-v2 | 0.9858 | 0.9819 | 0.9778 | 0.9652 | 0.9356 | 0.8903 | - | - | - | - | - |
| | all-roberta-large-v1 | 0.9747 | 0.9702 | 0.9677 | 0.9542 | 0.9083 | 0.8701 | - | - | - | - | - |
| beir-cqadupstack-physics | all-MiniLM-L6-v2 | 1.0019 | 0.9961 | 0.9858 | 0.9820 | 0.9819 | 0.9739 | 0.9807 | - | - | - | - |
| | all-mpnet-base-v2 | 0.9989 | 0.9903 | 0.9811 | 0.9769 | 0.9674 | 0.9676 | 0.9448 | - | - | - | - |
| | all-roberta-large-v1 | 0.9963 | 0.9875 | 0.9836 | 0.9785 | 0.9697 | 0.9912 | 0.9445 | - | - | - | - |
| beir-cqadupstack-programmers | all-MiniLM-L6-v2 | 0.9881 | 0.9724 | 0.9607 | 0.9583 | 0.9254 | 0.9526 | - | 0.8684 | - | - | - |
| | all-mpnet-base-v2 | 0.9864 | 0.9762 | 0.9699 | 0.9650 | 0.9350 | 0.9432 | - | 0.8911 | - | - | - |
| | all-roberta-large-v1 | 0.9825 | 0.9728 | 0.9648 | 0.9651 | 0.9404 | 0.9286 | - | 0.9069 | - | - | - |
| beir-cqadupstack-stats | all-MiniLM-L6-v2 | 1.0073 | 0.9984 | 1.0015 | 0.9757 | 0.9904 | - | - | - | - | - | - |
| | all-mpnet-base-v2 | 1.0071 | 0.9949 | 0.9963 | 0.9843 | 0.9748 | - | - | - | - | - | - |
| | all-roberta-large-v1 | 0.9931 | 0.9814 | 0.9824 | 0.9707 | 0.9515 | - | - | - | - | - | - |
| beir-cqadupstack-tex | all-MiniLM-L6-v2 | 0.9938 | 0.9816 | 0.9780 | 0.9726 | 0.9649 | 0.9769 | - | 0.9556 | - | - | - |
| | all-mpnet-base-v2 | 0.9902 | 0.9872 | 0.9823 | 0.9757 | 0.9712 | 0.9801 | - | 0.9650 | - | - | - |
| | all-roberta-large-v1 | 0.9709 | 0.9646 | 0.9567 | 0.9517 | 0.9426 | 0.9501 | - | 0.9518 | - | - | - |
| beir-cqadupstack-unix | all-MiniLM-L6-v2 | 0.9852 | 0.9791 | 0.9711 | 0.9729 | 0.9663 | - | - | - | - | - | - |
| | all-mpnet-base-v2 | 0.9948 | 0.9878 | 0.9821 | 0.9812 | 0.9803 | - | - | - | - | - | - |
| | all-roberta-large-v1 | 0.9942 | 0.9879 | 0.9847 | 0.9833 | 0.9787 | - | - | - | - | - | - |
| beir-cqadupstack-webmasters | all-MiniLM-L6-v2 | 0.9836 | 0.9749 | 0.9555 | 0.9640 | 0.9606 | 0.9010 | 0.9346 | 0.9598 | - | - | - |
| | all-mpnet-base-v2 | 0.9920 | 0.9800 | 0.9637 | 0.9683 | 0.9673 | 0.9337 | 0.9547 | 0.9700 | - | - | - |
| | all-roberta-large-v1 | 0.9795 | 0.9722 | 0.9528 | 0.9601 | 0.9549 | 0.9298 | 0.9386 | 0.9714 | - | - | - |
| beir-cqadupstack-wordpress | all-MiniLM-L6-v2 | 0.9986 | 0.9862 | 0.9799 | 0.9766 | - | 0.9727 | - | - | - | - | - |
| | all-mpnet-base-v2 | 0.9968 | 0.9900 | 0.9952 | 0.9951 | - | 0.9876 | - | - | - | - | - |
| | all-roberta-large-v1 | 0.9734 | 0.9690 | 0.9628 | 0.9712 | - | 0.9500 | - | - | - | - | - |
| beir-dbpedia-entity-test | all-MiniLM-L6-v2 | - | - | - | - | 0.9944 | 1.0003 | 0.9895 | 0.9822 | 0.9763 | 1.0038 | 1.0004 |
| | all-mpnet-base-v2 | - | - | - | - | 0.9687 | 0.9740 | 0.9625 | 0.9530 | 0.9540 | 0.9836 | 0.9773 |
| | all-roberta-large-v1 | - | - | - | - | 0.9645 | 0.9738 | 0.9620 | 0.9548 | 0.9490 | 0.9918 | 0.9802 |
| beir-fever-test | all-MiniLM-L6-v2 | 1.0064 | 1.0098 | 1.0060 | 1.0106 | - | - | - | - | - | - | - |
| | all-mpnet-base-v2 | 1.0090 | 1.0056 | 1.0035 | 1.0081 | - | - | - | - | - | - | - |
| | all-roberta-large-v1 | 1.0275 | 1.0243 | 1.0207 | 1.0304 | - | - | - | - | - | - | - |
| beir-fiqa-test | all-MiniLM-L6-v2 | 0.9967 | 0.9762 | 0.9632 | 0.9516 | - | - | - | - | - | - | - |
| | all-mpnet-base-v2 | 1.0023 | 0.9944 | 0.9903 | 0.9888 | - | - | - | - | - | - | - |
| | all-roberta-large-v1 | 0.9549 | 0.9423 | 0.9366 | 0.9337 | - | - | - | - | - | - | - |
| beir-hotpotqa-test | all-MiniLM-L6-v2 | - | 1.0202 | - | - | - | - | - | - | - | - | - |

| Dataset | Encoder | 1 | 2 | 4 | 8 | 16 | 32 | 64 | 128 | 256 | 512 | 1024 |
|---|---|---|---|---|---|---|---|---|---|---|---|---|
| | all-mpnet-base-v2 | - | 1.0242 | - | - | - | - | - | - | - | - | - |
| | all-roberta-large-v1 | - | 1.0028 | - | - | - | - | - | - | - | - | - |
| | all-MiniLM-L6-v2 | - | - | - | - | - | - | - | 0.9911 | 0.9997 | 0.9782 | - |
| beir-msmarco-test | all-mpnet-base-v2 | - | - | - | - | - | - | - | 0.9724 | 0.9839 | 0.9692 | - |
| | all-roberta-large-v1 | - | - | - | - | - | - | - | 0.9888 | 0.9928 | 0.9937 | - |
| | all-MiniLM-L6-v2 | 0.8409 | 0.8088 | 0.7641 | 0.7406 | 0.7094 | 0.6628 | 0.6604 | 0.6166 | 0.5958 | - | - |
| beir-nfcorpus-test | all-mpnet-base-v2 | 0.8099 | 0.7905 | 0.7359 | 0.7211 | 0.6869 | 0.6494 | 0.6431 | 0.6013 | 0.5697 | - | - |
| | all-roberta-large-v1 | 0.8824 | 0.8585 | 0.8138 | 0.7888 | 0.7583 | 0.7149 | 0.7109 | 0.6658 | 0.6335 | - | - |
| | all-MiniLM-L6-v2 | 0.9984 | 0.9986 | 0.9999 | - | - | - | - | - | - | - | - |
| beir-nq | all-mpnet-base-v2 | 0.9967 | 0.9943 | 0.9921 | - | - | - | - | - | - | - | - |
| | all-roberta-large-v1 | 0.9996 | 0.9978 | 0.9963 | - | - | - | - | - | - | - | - |
| | all-MiniLM-L6-v2 | 0.9978 | 0.9968 | 0.9966 | 0.9959 | 0.9962 | 0.9961 | 0.9896 | - | - | - | - |
| beir-quora-test | all-mpnet-base-v2 | 0.9981 | 0.9967 | 0.9976 | 0.9965 | 0.9971 | 0.9951 | 0.9903 | - | - | - | - |
| | all-roberta-large-v1 | 0.9987 | 0.9958 | 0.9962 | 0.9948 | 0.9961 | 0.9962 | 0.9894 | - | - | - | - |
| | all-MiniLM-L6-v2 | - | - | - | - | 0.6214 | - | - | - | - | - | - |
| beir-scidocs | all-mpnet-base-v2 | - | - | - | - | 0.5877 | - | - | - | - | - | - |
| | all-roberta-large-v1 | - | - | - | - | 0.6764 | - | - | - | - | - | - |
| | all-MiniLM-L6-v2 | 0.9795 | 0.9727 | 0.9702 | - | - | - | - | - | - | - | - |
| beir-scifact-test | all-mpnet-base-v2 | 0.9912 | 0.9893 | 0.9760 | - | - | - | - | - | - | - | - |
| | all-roberta-large-v1 | 1.0093 | 1.0020 | 0.9983 | - | - | - | - | - | - | - | - |
| | all-MiniLM-L6-v2 | - | - | - | - | - | - | - | - | - | 0.5774 | 0.5092 |
| beir-trec-covid | all-mpnet-base-v2 | - | - | - | - | - | - | - | - | - | 0.5585 | 0.4939 |
| | all-roberta-large-v1 | - | - | - | - | - | - | - | - | - | 0.5816 | 0.5223 |
| | all-MiniLM-L6-v2 | - | - | - | - | - | 0.8817 | - | - | - | - | - |
| beir-webis-touche2020-v2 | all-mpnet-base-v2 | - | - | - | - | - | 0.8755 | - | - | - | - | - |
| | all-roberta-large-v1 | - | - | - | - | - | 0.8572 | - | - | - | - | - |
| | all-MiniLM-L6-v2 | 1.0056 | 1.0048 | 1.0022 | 1.0008 | 0.9983 | - | - | - | - | - | - |
| car-v1.5-test200 | all-mpnet-base-v2 | 1.0064 | 1.0019 | 0.9983 | 0.9966 | 0.9854 | - | - | - | - | - | - |
| | all-roberta-large-v1 | 1.0095 | 1.0108 | 1.0063 | 1.0043 | 0.9939 | - | - | - | - | - | - |
| | all-MiniLM-L6-v2 | - | - | - | - | - | - | - | - | 0.6919 | 0.6915 | - |
| clinicaltrials-2021-trec-ct-2021 | all-mpnet-base-v2 | - | - | - | - | - | - | - | - | 0.7599 | 0.7744 | - |
| | all-roberta-large-v1 | - | - | - | - | - | - | - | - | 0.5492 | 0.5592 | - |
| | all-MiniLM-L6-v2 | - | - | - | - | 1.0107 | 1.0091 | - | - | - | - | - |
| clueweb12-touche-2022-task-2 | all-mpnet-base-v2 | - | - | - | - | 1.0399 | 1.0448 | - | - | - | - | - |
| | all-roberta-large-v1 | - | - | - | - | 1.0376 | 1.0402 | - | - | - | - | - |
| | all-MiniLM-L6-v2 | - | 0.8741 | 0.8509 | 0.8336 | 0.7800 | 0.7518 | - | - | - | - | - |
| cranfield | all-mpnet-base-v2 | - | 0.8594 | 0.8341 | 0.8152 | 0.7691 | 0.7469 | - | - | - | - | - |
| | all-roberta-large-v1 | - | 0.7313 | 0.6982 | 0.6747 | 0.6499 | 0.6050 | - | - | - | - | - |
| | all-MiniLM-L6-v2 | - | - | - | - | - | - | 0.4366 | 0.2979 | 0.2892 | 0.2283 | - |
| csl-trec-2023 | all-mpnet-base-v2 | - | - | - | - | - | - | 0.4513 | 0.3280 | 0.3147 | 0.2160 | - |
| | all-roberta-large-v1 | - | - | - | - | - | - | 0.1846 | 0.2320 | 0.2233 | 0.1679 | - |
| | all-MiniLM-L6-v2 | 0.8508 | 0.7974 | 0.7617 | 0.7595 | - | - | - | - | - | - | - |
| istella22-test | all-mpnet-base-v2 | 0.9136 | 0.8462 | 0.8373 | 0.8376 | - | - | - | - | - | - | - |
| | all-roberta-large-v1 | 0.9732 | 0.9055 | 0.8956 | 0.8900 | - | - | - | - | - | - | - |
| | all-MiniLM-L6-v2 | 1.0045 | 1.0018 | 0.9981 | 0.9914 | 0.9870 | 0.9646 | 0.9896 | - | - | - | - |
| lotte-lifestyle-dev-forum | all-mpnet-base-v2 | 1.0025 | 1.0013 | 0.9980 | 0.9936 | 0.9923 | 0.9753 | 0.9809 | - | - | - | - |
| | all-roberta-large-v1 | 0.9861 | 0.9771 | 0.9748 | 0.9730 | 0.9757 | 0.9563 | 0.9437 | - | - | - | - |
| | all-MiniLM-L6-v2 | 1.0085 | 1.0004 | 0.9962 | 0.9925 | 0.9908 | 0.9960 | - | - | - | - | - |
| lotte-lifestyle-test-forum | all-mpnet-base-v2 | 1.0072 | 1.0027 | 1.0002 | 0.9968 | 0.9937 | 0.9900 | - | - | - | - | - |
| | all-roberta-large-v1 | 0.9945 | 0.9878 | 0.9846 | 0.9818 | 0.9821 | 0.9805 | - | - | - | - | - |
| | all-MiniLM-L6-v2 | 1.0187 | 1.0136 | 1.0100 | 1.0023 | - | - | - | - | - | - | - |
| lotte-lifestyle-test-search | all-mpnet-base-v2 | 1.0283 | 1.0208 | 1.0250 | 1.0080 | - | - | - | - | - | - | - |
| | all-roberta-large-v1 | 0.9903 | 0.9841 | 0.9813 | 0.9767 | - | - | - | - | - | - | - |
| | all-MiniLM-L6-v2 | 1.0053 | 1.0018 | 0.9999 | 0.9972 | 0.9943 | 0.9961 | 0.9962 | 0.9912 | 1.0011 | - | - |
| lotte-pooled-test-forum | all-mpnet-base-v2 | 1.0027 | 1.0019 | 0.9998 | 0.9967 | 0.9931 | 0.9911 | 0.9933 | 0.9908 | 0.9882 | - | - |
| | all-roberta-large-v1 | 0.9975 | 0.9947 | 0.9915 | 0.9898 | 0.9869 | 0.9862 | 0.9863 | 0.9911 | 0.9805 | - | - |
| | all-MiniLM-L6-v2 | 1.0167 | 1.0153 | 1.0141 | 1.0112 | 1.0095 | 1.0122 | - | - | - | - | - |
| lotte-pooled-test-search | all-mpnet-base-v2 | 1.0159 | 1.0164 | 1.0165 | 1.0139 | 1.0150 | 1.0247 | - | - | - | - | - |
| | all-roberta-large-v1 | 0.9999 | 0.9974 | 0.9951 | 0.9941 | 0.9978 | 0.9918 | - | - | - | - | - |
| | all-MiniLM-L6-v2 | 1.0016 | 1.0005 | 0.9948 | 0.9911 | 0.9826 | - | - | - | - | - | - |
| lotte-recreation-test-forum | all-mpnet-base-v2 | 1.0061 | 1.0043 | 1.0018 | 0.9996 | 0.9928 | - | - | - | - | - | - |
| | all-roberta-large-v1 | 0.9832 | 0.9826 | 0.9814 | 0.9781 | 0.9555 | - | - | - | - | - | - |
| | all-MiniLM-L6-v2 | 1.0253 | 1.0229 | 1.0213 | 1.0157 | - | - | - | - | - | - | - |
| lotte-recreation-test-search | all-mpnet-base-v2 | 1.0244 | 1.0231 | 1.0254 | 1.0162 | - | - | - | - | - | - | - |
| | all-roberta-large-v1 | 0.9982 | 0.9944 | 0.9953 | 0.9936 | - | - | - | - | - | - | - |
| | all-MiniLM-L6-v2 | 0.9932 | 0.9919 | 0.9879 | 0.9805 | 0.9761 | 0.9664 | 0.9531 | 0.9434 | - | - | - |
| lotte-science-test-forum | all-mpnet-base-v2 | 0.9955 | 0.9931 | 0.9898 | 0.9797 | 0.9747 | 0.9609 | 0.9495 | 0.9457 | - | - | - |
| | all-roberta-large-v1 | 0.9882 | 0.9784 | 0.9770 | 0.9656 | 0.9577 | 0.9433 | 0.9250 | 0.9259 | - | - | - |
| | all-MiniLM-L6-v2 | 1.0184 | 1.0189 | 1.0090 | 1.0057 | 0.9942 | - | - | - | - | - | - |
| lotte-science-test-search | all-mpnet-base-v2 | 1.0213 | 1.0177 | 1.0052 | 1.0078 | 0.9934 | - | - | - | - | - | - |
| | all-roberta-large-v1 | 1.0086 | 1.0030 | 0.9934 | 0.9990 | 0.9954 | - | - | - | - | - | - |
| | all-MiniLM-L6-v2 | 0.9973 | 0.9974 | 0.9946 | 0.9918 | 0.9903 | 0.9986 | 0.9887 | 0.9898 | 0.9973 | - | - |
| lotte-technology-test-forum | all-mpnet-base-v2 | 0.9998 | 1.0007 | 0.9994 | 0.9971 | 0.9944 | 0.9991 | 0.9970 | 0.9966 | 0.9960 | - | - |
| | all-roberta-large-v1 | 0.9811 | 0.9731 | 0.9722 | 0.9717 | 0.9736 | 0.9756 | 0.9639 | 0.9666 | 0.9708 | - | - |
| | all-MiniLM-L6-v2 | 1.0086 | 1.0097 | 1.0087 | 1.0083 | 1.0080 | 1.0113 | - | - | - | - | - |
| lotte-technology-test-search | all-mpnet-base-v2 | 1.0101 | 1.0112 | 1.0120 | 1.0101 | 1.0087 | 1.0140 | - | - | - | - | - |
| | all-roberta-large-v1 | 0.9980 | 0.9979 | 0.9964 | 0.9943 | 0.9994 | 0.9984 | - | - | - | - | - |
| | all-MiniLM-L6-v2 | 1.0014 | 0.9940 | 0.9887 | 0.9775 | 0.9672 | 0.9728 | 0.9676 | - | - | - | - |
| lotte-writing-test-forum | all-mpnet-base-v2 | 1.0077 | 1.0069 | 1.0051 | 0.9948 | 0.9861 | 0.9894 | 0.9633 | - | - | - | - |
| | all-roberta-large-v1 | 0.9658 | 0.9552 | 0.9520 | 0.9372 | 0.9254 | 0.9365 | 0.9356 | - | - | - | - |
| | all-MiniLM-L6-v2 | 1.0254 | 1.0194 | 1.0110 | 1.0061 | 1.0072 | - | - | - | - | - | - |
| lotte-writing-test-search | all-mpnet-base-v2 | 1.0333 | 1.0308 | 1.0271 | 1.0306 | 1.0279 | - | - | - | - | - | - |
| | all-roberta-large-v1 | 0.9775 | 0.9714 | 0.9682 | 0.9662 | 0.9645 | - | - | - | - | - | - |
| | all-MiniLM-L6-v2 | - | - | 0.4845 | 0.3620 | 0.3425 | - | - | - | - | - | - |
| miracl-ar-dev | all-mpnet-base-v2 | - | - | 0.9382 | 0.9664 | 0.8842 | - | - | - | - | - | - |
| | all-roberta-large-v1 | - | - | 0.8367 | 0.6870 | 0.6403 | - | - | - | - | - | - |
| | all-MiniLM-L6-v2 | - | - | 0.4838 | 0.4230 | - | - | - | - | - | - | - |
| miracl-bn-dev | all-mpnet-base-v2 | - | - | 0.7289 | 0.4533 | - | - | - | - | - | - | - |
| | all-roberta-large-v1 | - | - | 0.8390 | 0.4817 | - | - | - | - | - | - | - |
| | all-MiniLM-L6-v2 | - | - | - | 0.9181 | 0.8639 | - | - | - | - | - | - |
| miracl-de-dev | | | | | | | | | | | | |

| Dataset | Encoder | 1 | 2 | 4 | 8 | 16 | 32 | 64 | 128 | 256 | 512 | 1024 |
|---|---|---|---|---|---|---|---|---|---|---|---|---|
| | all-mpnet-base-v2 | - | - | - | 0.9413 | 0.8958 | - | - | - | - | - | - |
| | all-roberta-large-v1 | - | - | - | 0.9449 | 0.8969 | - | - | - | - | - | - |
| | all-MiniLM-L6-v2 | - | - | - | 1.0472 | 1.0360 | - | - | - | - | - | - |
| miracl-en-dev | all-mpnet-base-v2 | - | - | - | 1.0616 | 1.0420 | - | - | - | - | - | - |
| | all-roberta-large-v1 | - | - | - | 1.0653 | 1.0250 | - | - | - | - | - | - |
| | all-MiniLM-L6-v2 | - | 0.9837 | 0.9487 | 0.9524 | - | - | - | - | - | - | - |
| miracl-es-dev | all-mpnet-base-v2 | - | 1.0251 | 1.0115 | 1.0062 | - | - | - | - | - | - | - |
| | all-roberta-large-v1 | - | 0.9635 | 0.9450 | 0.9243 | - | - | - | - | - | - | - |
| | all-MiniLM-L6-v2 | - | - | - | 0.4932 | 0.4441 | - | - | - | - | - | - |
| miracl-fa-dev | all-mpnet-base-v2 | - | - | - | 0.6628 | 0.6065 | - | - | - | - | - | - |
| | all-roberta-large-v1 | - | - | - | 0.4666 | 0.4086 | - | - | - | - | - | - |
| | all-MiniLM-L6-v2 | 0.9534 | 0.9033 | 1.0176 | 0.7965 | 0.7561 | - | - | - | - | - | - |
| miracl-fi-dev | all-mpnet-base-v2 | 0.9892 | 0.9658 | 1.0534 | 0.8844 | 0.8488 | - | - | - | - | - | - |
| | all-roberta-large-v1 | 0.9928 | 1.0082 | 0.9775 | 0.8756 | 0.8807 | - | - | - | - | - | - |
| | all-MiniLM-L6-v2 | - | - | - | 0.9688 | - | - | - | - | - | - | - |
| miracl-fr-dev | all-mpnet-base-v2 | - | - | - | 1.0126 | - | - | - | - | - | - | - |
| | all-roberta-large-v1 | - | - | - | 1.0171 | - | - | - | - | - | - | - |
| | all-MiniLM-L6-v2 | - | - | 0.4534 | 0.4510 | - | - | - | - | - | - | - |
| miracl-hi-dev | all-mpnet-base-v2 | - | - | 0.4747 | 0.5099 | - | - | - | - | - | - | - |
| | all-roberta-large-v1 | - | - | 0.6235 | 0.4776 | - | - | - | - | - | - | - |
| | all-MiniLM-L6-v2 | - | 0.8873 | 0.7569 | 0.7597 | 0.6826 | - | - | - | - | - | - |
| miracl-id-dev | all-mpnet-base-v2 | - | 0.9714 | 0.8731 | 0.8600 | 0.7450 | - | - | - | - | - | - |
| | all-roberta-large-v1 | - | 1.0510 | 0.9975 | 0.9302 | 0.7791 | - | - | - | - | - | - |
| | all-MiniLM-L6-v2 | 1.1188 | 1.1127 | - | 0.8726 | 0.7930 | - | - | - | - | - | - |
| miracl-ja-dev | all-mpnet-base-v2 | 1.0291 | 0.9877 | - | 0.8603 | 0.8065 | - | - | - | - | - | - |
| | all-roberta-large-v1 | 0.9662 | 0.9114 | - | 0.7059 | 0.6950 | - | - | - | - | - | - |
| | all-MiniLM-L6-v2 | - | - | - | 0.5574 | 0.5257 | - | - | - | - | - | - |
| miracl-ko-dev | all-mpnet-base-v2 | - | - | - | 0.8181 | 0.7710 | - | - | - | - | - | - |
| | all-roberta-large-v1 | - | - | - | 0.9777 | 0.9206 | - | - | - | - | - | - |
| | all-MiniLM-L6-v2 | - | - | - | 0.4261 | 0.4024 | - | - | - | - | - | - |
| miracl-ru-dev | all-mpnet-base-v2 | - | - | - | 0.5509 | 0.5104 | - | - | - | - | - | - |
| | all-roberta-large-v1 | - | - | - | 0.5722 | 0.5479 | - | - | - | - | - | - |
| | all-MiniLM-L6-v2 | 0.9093 | 0.6715 | 0.6869 | 0.6241 | 0.5643 | - | - | - | - | - | - |
| miracl-sw-dev | all-mpnet-base-v2 | 0.9375 | 0.8202 | 0.7448 | 0.7092 | 0.6317 | - | - | - | - | - | - |
| | all-roberta-large-v1 | 0.9476 | 0.7875 | 0.7026 | 0.6965 | 0.6732 | - | - | - | - | - | - |
| | all-MiniLM-L6-v2 | 2.4890 | - | 0.7665 | 1.8069 | - | - | - | - | - | - | - |
| miracl-te-dev | all-mpnet-base-v2 | 3.3636 | - | 0.9654 | 2.6062 | - | - | - | - | - | - | - |
| | all-roberta-large-v1 | 1.6474 | - | 0.9428 | 1.0728 | - | - | - | - | - | - | - |
| | all-MiniLM-L6-v2 | 1.1004 | 1.0153 | 0.7033 | 0.8087 | - | - | - | - | - | - | - |
| miracl-th-dev | all-mpnet-base-v2 | 1.0579 | 1.0153 | 0.7551 | 0.7880 | - | - | - | - | - | - | - |
| | all-roberta-large-v1 | 1.1054 | 0.7193 | 0.5881 | 0.6339 | - | - | - | - | - | - | - |
| | all-MiniLM-L6-v2 | - | - | - | 0.5947 | - | - | - | - | - | - | - |
| miracl-yo-dev | all-mpnet-base-v2 | - | - | - | 0.6432 | - | - | - | - | - | - | - |
| | all-roberta-large-v1 | - | - | - | 0.6481 | - | - | - | - | - | - | - |
| | all-MiniLM-L6-v2 | - | - | - | 0.7882 | - | - | - | - | - | - | - |
| miracl-zh-dev | all-mpnet-base-v2 | - | - | - | 0.7767 | - | - | - | - | - | - | - |
| | all-roberta-large-v1 | - | - | - | 0.8158 | - | - | - | - | - | - | - |
| | all-MiniLM-L6-v2 | 1.0430 | 1.0061 | 0.9934 | - | - | - | - | - | - | - | - |
| mmarco-de-dev | all-mpnet-base-v2 | 1.0469 | 1.0134 | 0.9968 | - | - | - | - | - | - | - | - |
| | all-roberta-large-v1 | 1.0605 | 1.0198 | 1.0037 | - | - | - | - | - | - | - | - |
| | all-MiniLM-L6-v2 | 1.0895 | 1.0362 | 1.0151 | - | - | - | - | - | - | - | - |
| mmarco-es-dev | all-mpnet-base-v2 | 1.1151 | 1.0707 | 1.0490 | - | - | - | - | - | - | - | - |
| | all-roberta-large-v1 | 1.0657 | 1.0371 | 1.0125 | - | - | - | - | - | - | - | - |
| | all-MiniLM-L6-v2 | 1.0662 | 1.0353 | 1.0235 | - | - | - | - | - | - | - | - |
| mmarco-fr-dev | all-mpnet-base-v2 | 1.1043 | 1.0787 | 1.0683 | - | - | - | - | - | - | - | - |
| | all-roberta-large-v1 | 1.0710 | 1.0478 | 1.0400 | - | - | - | - | - | - | - | - |
| | all-MiniLM-L6-v2 | 0.9636 | 0.9294 | 0.8806 | - | - | - | - | - | - | - | - |
| mmarco-id-dev | all-mpnet-base-v2 | 0.9613 | 0.9278 | 0.9057 | - | - | - | - | - | - | - | - |
| | all-roberta-large-v1 | 1.0106 | 0.9966 | 0.9757 | - | - | - | - | - | - | - | - |
| | all-MiniLM-L6-v2 | 1.0142 | 0.9604 | 0.9246 | - | - | - | - | - | - | - | - |
| mmarco-it-dev | all-mpnet-base-v2 | 1.0929 | 1.0477 | 1.0190 | - | - | - | - | - | - | - | - |
| | all-roberta-large-v1 | 1.0819 | 1.0401 | 1.0217 | - | - | - | - | - | - | - | - |
| | all-MiniLM-L6-v2 | 1.0547 | 0.9933 | 0.9843 | - | - | - | - | - | - | - | - |
| mmarco-pt-dev | all-mpnet-base-v2 | 1.0699 | 1.0087 | 1.0015 | - | - | - | - | - | - | - | - |
| | all-roberta-large-v1 | 1.1048 | 1.0596 | 1.0565 | - | - | - | - | - | - | - | - |
| | all-MiniLM-L6-v2 | 0.8366 | 0.7431 | 0.6619 | - | - | - | - | - | - | - | - |
| mmarco-ru-dev | all-mpnet-base-v2 | 0.9299 | 0.8138 | 0.7399 | - | - | - | - | - | - | - | - |
| | all-roberta-large-v1 | 1.0269 | 0.8507 | 0.7673 | - | - | - | - | - | - | - | - |
| | all-MiniLM-L6-v2 | 0.8436 | 0.7491 | - | - | - | - | - | - | - | - | - |
| mr-tydi-ar-test | all-mpnet-base-v2 | 1.5930 | 1.4724 | - | - | - | - | - | - | - | - | - |
| | all-roberta-large-v1 | 1.3149 | 1.1772 | - | - | - | - | - | - | - | - | - |
| | all-MiniLM-L6-v2 | 0.7608 | 0.7566 | - | - | - | - | - | - | - | - | - |
| mr-tydi-bn-test | all-mpnet-base-v2 | 0.8373 | 0.7149 | - | - | - | - | - | - | - | - | - |
| | all-roberta-large-v1 | 1.1507 | 1.0248 | - | - | - | - | - | - | - | - | - |
| | all-MiniLM-L6-v2 | 1.0570 | 1.0489 | - | - | - | - | - | - | - | - | - |
| mr-tydi-en-test | all-mpnet-base-v2 | 1.0833 | 1.0749 | - | - | - | - | - | - | - | - | - |
| | all-roberta-large-v1 | 1.0818 | 1.0772 | - | - | - | - | - | - | - | - | - |
| | all-MiniLM-L6-v2 | 1.0114 | 0.9412 | - | - | - | - | - | - | - | - | - |
| mr-tydi-fi-test | all-mpnet-base-v2 | 1.0621 | 0.9964 | - | - | - | - | - | - | - | - | - |
| | all-roberta-large-v1 | 1.0961 | 1.0303 | - | - | - | - | - | - | - | - | - |
| | all-MiniLM-L6-v2 | 1.0252 | 0.9791 | - | - | - | - | - | - | - | - | - |
| mr-tydi-id-test | all-mpnet-base-v2 | 1.0860 | 1.0595 | - | - | - | - | - | - | - | - | - |
| | all-roberta-large-v1 | 1.1161 | 1.1031 | - | - | - | - | - | - | - | - | - |
| | all-MiniLM-L6-v2 | 1.2202 | 1.1451 | - | - | - | - | - | - | - | - | - |
| mr-tydi-ja-test | all-mpnet-base-v2 | 1.0842 | 1.0174 | - | - | - | - | - | - | - | - | - |
| | all-roberta-large-v1 | 1.0716 | 0.9628 | - | - | - | - | - | - | - | - | - |
| | all-MiniLM-L6-v2 | 0.9719 | 0.9485 | - | - | - | - | - | - | - | - | - |
| mr-tydi-ko-test | all-mpnet-base-v2 | 1.3114 | 1.2399 | - | - | - | - | - | - | - | - | - |
| | all-roberta-large-v1 | 1.9154 | 1.7743 | - | - | - | - | - | - | - | - | - |
| | all-MiniLM-L6-v2 | 0.8008 | 0.7448 | - | - | - | - | - | - | - | - | - |
| mr-tydi-ru-test | | | | | | | | | | | | |

| Dataset | Encoder | 1 | 2 | 4 | 8 | 16 | 32 | 64 | 128 | 256 | 512 | 1024 |
|---|---|---|---|---|---|---|---|---|---|---|---|---|
| | all-mpnet-base-v2 | 0.9144 | 0.8205 | - | - | - | - | - | - | - | - | - |
| | all-roberta-large-v1 | 0.9946 | 0.8944 | - | - | - | - | - | - | - | - | - |
| | all-MiniLM-L6-v2 | 1.0325 | 0.9230 | - | - | - | - | - | - | - | - | - |
| mr-tydi-sw-test | all-mpnet-base-v2 | 1.1153 | 1.0085 | - | - | - | - | - | - | - | - | - |
| | all-roberta-large-v1 | 1.0871 | 0.9998 | - | - | - | - | - | - | - | - | - |
| | all-MiniLM-L6-v2 | 2.0689 | 2.2970 | - | - | - | - | - | - | - | - | - |
| mr-tydi-te-test | all-mpnet-base-v2 | 2.8740 | 3.1982 | - | - | - | - | - | - | - | - | - |
| | all-roberta-large-v1 | 1.6067 | 1.4740 | - | - | - | - | - | - | - | - | - |
| | all-MiniLM-L6-v2 | 1.7386 | 1.2216 | - | - | - | - | - | - | - | - | - |
| mr-tydi-th-test | all-mpnet-base-v2 | 1.5355 | 1.1623 | - | - | - | - | - | - | - | - | - |
| | all-roberta-large-v1 | 1.5392 | 1.3155 | - | - | - | - | - | - | - | - | - |
| | all-MiniLM-L6-v2 | - | - | - | - | - | - | - | 0.8831 | 0.8993 | 0.8955 | 0.8594 |
| msmarco-document-trec-dl-2019 | all-mpnet-base-v2 | - | - | - | - | - | - | - | 0.8796 | 0.8972 | 0.8859 | 0.8593 |
| | all-roberta-large-v1 | - | - | - | - | - | - | - | 0.8511 | 0.8645 | 0.8521 | 0.8373 |
| | all-MiniLM-L6-v2 | - | - | - | - | - | - | - | 0.8846 | 0.8763 | - | - |
| msmarco-document-trec-dl-2020 | all-mpnet-base-v2 | - | - | - | - | - | - | - | 0.8877 | 0.8834 | - | - |
| | all-roberta-large-v1 | - | - | - | - | - | - | - | 0.8632 | 0.8627 | - | - |
| | all-MiniLM-L6-v2 | - | 0.9460 | 0.9307 | - | - | - | - | 0.9064 | 0.9081 | - | - |
| msmarco-document-trec-dl-hard-fold1 | all-mpnet-base-v2 | - | 0.9261 | 0.9575 | - | - | - | - | 0.9063 | 0.9139 | - | - |
| | all-roberta-large-v1 | - | 0.9256 | 0.9262 | - | - | - | - | 0.8916 | 0.9148 | - | - |
| | all-MiniLM-L6-v2 | - | - | 0.9591 | - | - | - | - | 0.8840 | 0.9144 | - | - |
| msmarco-document-trec-dl-hard-fold2 | all-mpnet-base-v2 | - | - | 0.9596 | - | - | - | - | 0.9163 | 0.9140 | - | - |
| | all-roberta-large-v1 | - | - | 0.9217 | - | - | - | - | 0.8743 | 0.8768 | - | - |
| | all-MiniLM-L6-v2 | - | 0.9374 | 0.9528 | 0.9828 | - | - | - | 0.8860 | - | - | - |
| msmarco-document-trec-dl-hard-fold3 | all-mpnet-base-v2 | - | 0.9331 | 0.9623 | 0.9604 | - | - | - | 0.8800 | - | - | - |
| | all-roberta-large-v1 | - | 0.9090 | 0.9355 | 0.9288 | - | - | - | 0.8642 | - | - | - |
| | all-MiniLM-L6-v2 | - | 0.9645 | 0.9171 | - | - | - | - | 0.9169 | 0.9172 | - | - |
| msmarco-document-trec-dl-hard-fold4 | all-mpnet-base-v2 | - | 0.9707 | 0.9231 | - | - | - | - | 0.9234 | 0.9270 | - | - |
| | all-roberta-large-v1 | - | 0.9288 | 0.8909 | - | - | - | - | 0.8751 | 0.8824 | - | - |
| | all-MiniLM-L6-v2 | - | - | 0.9782 | - | - | - | - | 0.8724 | 0.8774 | 0.9023 | 0.8561 |
| msmarco-document-trec-dl-hard-fold5 | all-mpnet-base-v2 | - | - | 0.9653 | - | - | - | - | 0.8836 | 0.8604 | 0.8847 | 0.8645 |
| | all-roberta-large-v1 | - | - | 0.9028 | - | - | - | - | 0.8383 | 0.8045 | 0.8508 | 0.8053 |
| | all-MiniLM-L6-v2 | - | - | - | - | - | 0.9243 | - | 0.9177 | 0.9260 | 0.9083 | 0.8789 |
| msmarco-document-v2-trec-dl-2019 | all-mpnet-base-v2 | - | - | - | - | - | 0.9286 | - | 0.9088 | 0.9209 | 0.8968 | 0.8649 |
| | all-roberta-large-v1 | - | - | - | - | - | 0.8427 | - | 0.8378 | 0.8424 | 0.8223 | 0.8060 |
| | all-MiniLM-L6-v2 | - | - | - | - | - | 0.9272 | - | 0.9143 | 0.9046 | - | - |
| msmarco-document-v2-trec-dl-2020 | all-mpnet-base-v2 | - | - | - | - | - | 0.9226 | - | 0.9124 | 0.8995 | - | - |
| | all-roberta-large-v1 | - | - | - | - | - | 0.8666 | - | 0.8474 | 0.8406 | - | - |
| | all-MiniLM-L6-v2 | - | - | - | - | - | 0.9576 | - | 0.9492 | 0.9554 | 0.9598 | - |
| msmarco-document-v2-trec-dl-2021 | all-mpnet-base-v2 | - | - | - | - | - | 0.9479 | - | 0.9497 | 0.9527 | 0.9483 | - |
| | all-roberta-large-v1 | - | - | - | - | - | 0.8875 | - | 0.8864 | 0.8885 | 0.9041 | - |
| | all-MiniLM-L6-v2 | - | - | - | - | - | - | - | 0.9936 | 1.0010 | - | - |
| msmarco-passage-trec-dl-2020 | all-mpnet-base-v2 | - | - | - | - | - | - | - | 0.9723 | 0.9809 | - | - |
| | all-roberta-large-v1 | - | - | - | - | - | - | - | 0.9814 | 0.9954 | - | - |
| | all-MiniLM-L6-v2 | - | - | 1.0031 | 0.9983 | - | - | - | 0.9901 | 0.9867 | - | - |
| msmarco-passage-trec-dl-hard | all-mpnet-base-v2 | - | - | 0.9873 | 0.9893 | - | - | - | 0.9798 | 0.9715 | - | - |
| | all-roberta-large-v1 | - | - | 1.0022 | 0.9965 | - | - | - | 0.9903 | 0.9872 | - | - |
| | all-MiniLM-L6-v2 | 0.9992 | 1.0049 | 1.0007 | 1.0018 | 1.0019 | - | - | - | - | - | - |
| msmarco-qna-dev | all-mpnet-base-v2 | 0.9961 | 0.9971 | 0.9932 | 0.9936 | 0.9941 | - | - | - | - | - | - |
| | all-roberta-large-v1 | 0.9891 | 0.9955 | 0.9914 | 0.9910 | 0.9935 | - | - | - | - | - | - |
| | all-MiniLM-L6-v2 | 0.9993 | 0.9960 | 0.9951 | - | - | - | - | - | - | - | - |
| natural-questions-dev | all-mpnet-base-v2 | 0.9930 | 0.9897 | 0.9880 | - | - | - | - | - | - | - | - |
| | all-roberta-large-v1 | 0.9982 | 0.9957 | 0.9930 | - | - | - | - | - | - | - | - |
| | all-MiniLM-L6-v2 | - | - | - | - | - | - | - | - | - | 0.7506 | 0.7304 |
| pmc-v1-trec-cds-2014 | all-mpnet-base-v2 | - | - | - | - | - | - | - | - | - | 0.7692 | 0.7802 |
| | all-roberta-large-v1 | - | - | - | - | - | - | - | - | - | 0.8171 | 0.8007 |
| | all-MiniLM-L6-v2 | - | - | - | - | - | - | - | - | - | 0.8911 | 0.8315 |
| pmc-v1-trec-cds-2015 | all-mpnet-base-v2 | - | - | - | - | - | - | - | - | - | 0.9280 | 0.8950 |
| | all-roberta-large-v1 | - | - | - | - | - | - | - | - | - | 0.9626 | 0.9386 |
| | all-MiniLM-L6-v2 | - | - | - | - | - | - | - | - | - | 0.8121 | 0.7413 |
| pmc-v2-trec-cds-2016 | all-mpnet-base-v2 | - | - | - | - | - | - | - | - | - | 0.9086 | 0.8223 |
| | all-roberta-large-v1 | - | - | - | - | - | - | - | - | - | 0.9069 | 0.8343 |
| | all-MiniLM-L6-v2 | 1.1284 | - | - | - | - | - | - | - | - | - | - |
| trec-tot-2023-dev | all-mpnet-base-v2 | 1.1324 | - | - | - | - | - | - | - | - | - | - |
| | all-roberta-large-v1 | 1.1609 | - | - | - | - | - | - | - | - | - | - |
| | all-MiniLM-L6-v2 | 0.9931 | 0.9366 | 0.9209 | 0.9150 | 0.9220 | 0.9055 | 0.9218 | - | - | - | - |
| vaswani | all-mpnet-base-v2 | 0.9458 | 0.9278 | 0.8854 | 0.8922 | 0.8875 | 0.8706 | 0.8956 | - | - | - | - |
| | all-roberta-large-v1 | 0.8258 | 0.7912 | 0.7356 | 0.7194 | 0.7164 | 0.6911 | 0.7124 | - | - | - | - |

*Table 5.* Median sensitivity value per log bin for $K_2$

| Dataset | Encoder | 1 | 2 | 4 | 8 | 16 | 32 | 64 | 128 | 256 | 512 | 1024 |
|---|---|---|---|---|---|---|---|---|---|---|---|---|
| **misc** | | | | | | | | | | | | |
| | all-MiniLM-L6-v2 | - | 0.4121 | 0.4140 | 0.4215 | 0.4204 | 0.4210 | 0.4189 | - | - | - | - |
| antique-test | all-mpnet-base-v2 | - | 0.3031 | 0.3039 | 0.3070 | 0.3064 | 0.3067 | 0.3072 | - | - | - | - |
| | all-roberta-large-v1 | - | 0.2680 | 0.2661 | 0.2662 | 0.2666 | 0.2664 | 0.2654 | - | - | - | - |
| | all-MiniLM-L6-v2 | 0.4072 | - | - | - | - | - | - | - | - | - | - |
| beir-arguana | all-mpnet-base-v2 | 0.3013 | - | - | - | - | - | - | - | - | - | - |
| | all-roberta-large-v1 | 0.2610 | - | - | - | - | - | - | - | - | - | - |
| | all-MiniLM-L6-v2 | 0.4102 | 0.4165 | 0.4213 | - | - | - | - | - | - | - | - |
| beir-climate-fever | all-mpnet-base-v2 | 0.2974 | 0.2973 | 0.3021 | - | - | - | - | - | - | - | - |
| | all-roberta-large-v1 | 0.2760 | 0.2745 | 0.3215 | - | - | - | - | - | - | - | - |
| | all-MiniLM-L6-v2 | 0.4017 | 0.4010 | 0.4027 | 0.4029 | 0.4059 | 0.4046 | - | - | 0.4112 | - | - |
| beir-cqadupstack-android | all-mpnet-base-v2 | 0.2981 | 0.2971 | 0.2980 | 0.2987 | 0.2996 | 0.2958 | - | - | 0.2971 | - | - |

| Dataset | Encoder | 1 | 2 | 4 | 8 | 16 | 32 | 64 | 128 | 256 | 512 | 1024 |
|---|---|---|---|---|---|---|---|---|---|---|---|---|
|  | all-roberta-large-v1 | 0.2573 | 0.2561 | 0.2570 | 0.2571 | 0.2577 | 0.2556 | - | - | 0.2534 | - | - |
| beir-cqadupstack-english | all-MiniLM-L6-v2 | 0.4069 | 0.4075 | 0.4097 | 0.4108 | 0.4102 | 0.4121 | 0.4079 | - | - | - | - |
|  | all-mpnet-base-v2 | 0.3002 | 0.2995 | 0.3001 | 0.3016 | 0.2996 | 0.2991 | 0.2996 | - | - | - | - |
|  | all-roberta-large-v1 | 0.2619 | 0.2609 | 0.2612 | 0.2625 | 0.2616 | 0.2615 | 0.2612 | - | - | - | - |
| beir-cqadupstack-gaming | all-MiniLM-L6-v2 | 0.4081 | 0.4087 | 0.4101 | 0.4056 | 0.4089 | - | - | - | - | - | - |
|  | all-mpnet-base-v2 | 0.3001 | 0.2998 | 0.2999 | 0.2969 | 0.2997 | - | - | - | - | - | - |
|  | all-roberta-large-v1 | 0.2610 | 0.2606 | 0.2607 | 0.2598 | 0.2609 | - | - | - | - | - | - |
| beir-cqadupstack-gis | all-MiniLM-L6-v2 | 0.4040 | 0.4044 | 0.4047 | 0.4071 | 0.4009 | - | - | - | - | - | - |
|  | all-mpnet-base-v2 | 0.2955 | 0.2965 | 0.2960 | 0.2967 | 0.2876 | - | - | - | - | - | - |
|  | all-roberta-large-v1 | 0.2571 | 0.2565 | 0.2573 | 0.2592 | 0.2551 | - | - | - | - | - | - |
| beir-cqadupstack-mathematica | all-MiniLM-L6-v2 | 0.4036 | 0.4045 | 0.4065 | 0.4081 | 0.4049 | 0.3975 | - | - | - | - | - |
|  | all-mpnet-base-v2 | 0.2965 | 0.2979 | 0.2986 | 0.2983 | 0.2955 | 0.2948 | - | - | - | - | - |
|  | all-roberta-large-v1 | 0.2566 | 0.2577 | 0.2588 | 0.2583 | 0.2521 | 0.2537 | - | - | - | - | - |
| beir-cqadupstack-physics | all-MiniLM-L6-v2 | 0.4089 | 0.4110 | 0.4106 | 0.4121 | 0.4167 | 0.4211 | 0.4152 | - | - | - | - |
|  | all-mpnet-base-v2 | 0.3001 | 0.3002 | 0.2999 | 0.3006 | 0.3018 | 0.3049 | 0.2942 | - | - | - | - |
|  | all-roberta-large-v1 | 0.2617 | 0.2613 | 0.2617 | 0.2618 | 0.2619 | 0.2683 | 0.2570 | - | - | - | - |
| beir-cqadupstack-programmers | all-MiniLM-L6-v2 | 0.4049 | 0.4056 | 0.4063 | 0.4072 | 0.4074 | 0.4168 | - | 0.4045 | - | - | - |
|  | all-mpnet-base-v2 | 0.2975 | 0.2981 | 0.2985 | 0.2987 | 0.2969 | 0.3021 | - | 0.2945 | - | - | - |
|  | all-roberta-large-v1 | 0.2589 | 0.2591 | 0.2590 | 0.2596 | 0.2602 | 0.2604 | - | 0.2583 | - | - | - |
| beir-cqadupstack-stats | all-MiniLM-L6-v2 | 0.4100 | 0.4110 | 0.4140 | 0.4079 | 0.4207 | - | - | - | - | - | - |
|  | all-mpnet-base-v2 | 0.3021 | 0.3012 | 0.3023 | 0.3004 | 0.3051 | - | - | - | - | - | - |
|  | all-roberta-large-v1 | 0.2605 | 0.2601 | 0.2609 | 0.2608 | 0.2609 | - | - | - | - | - | - |
| beir-cqadupstack-tex | all-MiniLM-L6-v2 | 0.4056 | 0.4051 | 0.4065 | 0.4073 | 0.4068 | 0.4108 | - | 0.4109 | - | - | - |
|  | all-mpnet-base-v2 | 0.2975 | 0.2986 | 0.2988 | 0.2984 | 0.2988 | 0.3015 | - | 0.3011 | - | - | - |
|  | all-roberta-large-v1 | 0.2558 | 0.2561 | 0.2558 | 0.2557 | 0.2552 | 0.2566 | - | 0.2582 | - | - | - |
| beir-cqadupstack-unix | all-MiniLM-L6-v2 | 0.4023 | 0.4044 | 0.4039 | 0.4064 | 0.4072 | - | - | - | - | - | - |
|  | all-mpnet-base-v2 | 0.2980 | 0.2982 | 0.2978 | 0.2982 | 0.3005 | - | - | - | - | - | - |
|  | all-roberta-large-v1 | 0.2605 | 0.2601 | 0.2601 | 0.2603 | 0.2600 | - | - | - | - | - | - |
| beir-cqadupstack-webmasters | all-MiniLM-L6-v2 | 0.4034 | 0.4066 | 0.4050 | 0.4104 | 0.4090 | 0.3997 | 0.4089 | 0.4120 | - | - | - |
|  | all-mpnet-base-v2 | 0.2989 | 0.2987 | 0.2977 | 0.3009 | 0.2996 | 0.2984 | 0.3007 | 0.3023 | - | - | - |
|  | all-roberta-large-v1 | 0.2579 | 0.2587 | 0.2563 | 0.2593 | 0.2583 | 0.2567 | 0.2596 | 0.2620 | - | - | - |
| beir-cqadupstack-wordpress | all-MiniLM-L6-v2 | 0.4056 | 0.4045 | 0.4060 | 0.4086 | - | 0.4086 | - | - | - | - | - |
|  | all-mpnet-base-v2 | 0.2981 | 0.2982 | 0.3002 | 0.3036 | - | 0.3033 | - | - | - | - | - |
|  | all-roberta-large-v1 | 0.2553 | 0.2558 | 0.2559 | 0.2593 | - | 0.2555 | - | - | - | - | - |
| beir-dbpedia-entity-test | all-MiniLM-L6-v2 | - | - | - | - | 0.4300 | 0.4279 | 0.4264 | 0.4275 | 0.4300 | 0.4285 | 0.4331 |
|  | all-mpnet-base-v2 | - | - | - | - | 0.3091 | 0.3097 | 0.3089 | 0.3095 | 0.3112 | 0.3085 | 0.3103 |
|  | all-roberta-large-v1 | - | - | - | - | 0.2690 | 0.2688 | 0.2673 | 0.2681 | 0.2693 | 0.2710 | 0.2731 |
| beir-fever-test | all-MiniLM-L6-v2 | 0.4114 | 0.4150 | 0.4168 | 0.4187 | - | - | - | - | - | - | - |
|  | all-mpnet-base-v2 | 0.3034 | 0.3048 | 0.3065 | 0.3092 | - | - | - | - | - | - | - |
|  | all-roberta-large-v1 | 0.2700 | 0.2708 | 0.2715 | 0.2739 | - | - | - | - | - | - | - |
| beir-fiqa-test | all-MiniLM-L6-v2 | 0.4088 | 0.4087 | 0.4093 | 0.4099 | - | - | - | - | - | - | - |
|  | all-mpnet-base-v2 | 0.3010 | 0.3011 | 0.3018 | 0.3028 | - | - | - | - | - | - | - |
|  | all-roberta-large-v1 | 0.2529 | 0.2536 | 0.2544 | 0.2553 | - | - | - | - | - | - | - |
| beir-hotpotqa-test | all-MiniLM-L6-v2 | - | 0.4192 | - | - | - | - | - | - | - | - | - |
|  | all-mpnet-base-v2 | - | 0.3104 | - | - | - | - | - | - | - | - | - |
|  | all-roberta-large-v1 | - | 0.2648 | - | - | - | - | - | - | - | - | - |
| beir-msmarco-test | all-MiniLM-L6-v2 | - | - | - | - | - | - | - | 0.4184 | 0.4186 | 0.4163 | - |
|  | all-mpnet-base-v2 | - | - | - | - | - | - | - | 0.3011 | 0.3004 | 0.3002 | - |
|  | all-roberta-large-v1 | - | - | - | - | - | - | - | 0.2629 | 0.2622 | 0.2627 | - |
| beir-nfcorpus-test | all-MiniLM-L6-v2 | 0.3817 | 0.3854 | 0.3849 | 0.3877 | 0.3889 | 0.3873 | 0.3913 | 0.3958 | 0.4068 | - | - |
|  | all-mpnet-base-v2 | 0.2731 | 0.2766 | 0.2750 | 0.2782 | 0.2779 | 0.2780 | 0.2804 | 0.2857 | 0.2892 | - | - |
|  | all-roberta-large-v1 | 0.2528 | 0.2545 | 0.2538 | 0.2547 | 0.2544 | 0.2534 | 0.2550 | 0.2592 | 0.2619 | - | - |
| beir-nq | all-MiniLM-L6-v2 | 0.4047 | 0.4066 | 0.4072 | - | - | - | - | - | - | - | - |
|  | all-mpnet-base-v2 | 0.2977 | 0.2980 | 0.2972 | - | - | - | - | - | - | - | - |
|  | all-roberta-large-v1 | 0.2612 | 0.2613 | 0.2612 | - | - | - | - | - | - | - | - |
| beir-quora-test | all-MiniLM-L6-v2 | 0.4055 | 0.4064 | 0.4076 | 0.4083 | 0.4094 | 0.4097 | 0.4089 | - | - | - | - |
|  | all-mpnet-base-v2 | 0.2986 | 0.2989 | 0.2997 | 0.2998 | 0.3005 | 0.3001 | 0.2989 | - | - | - | - |
|  | all-roberta-large-v1 | 0.2614 | 0.2612 | 0.2617 | 0.2617 | 0.2621 | 0.2625 | 0.2606 | - | - | - | - |
| beir-scidocs | all-MiniLM-L6-v2 | - | - | - | - | 0.4082 | - | - | - | - | - | - |
|  | all-mpnet-base-v2 | - | - | - | - | 0.2952 | - | - | - | - | - | - |
|  | all-roberta-large-v1 | - | - | - | - | 0.2584 | - | - | - | - | - | - |
| beir-scifact-test | all-MiniLM-L6-v2 | 0.3978 | 0.3998 | 0.3982 | - | - | - | - | - | - | - | - |
|  | all-mpnet-base-v2 | 0.2961 | 0.2978 | 0.2947 | - | - | - | - | - | - | - | - |
|  | all-roberta-large-v1 | 0.2641 | 0.2640 | 0.2638 | - | - | - | - | - | - | - | - |
| beir-trec-covid | all-MiniLM-L6-v2 | - | - | - | - | - | - | - | - | - | 0.4750 | 0.4648 |
|  | all-mpnet-base-v2 | - | - | - | - | - | - | - | - | - | 0.3353 | 0.3262 |
|  | all-roberta-large-v1 | - | - | - | - | - | - | - | - | - | 0.2956 | 0.2906 |
| beir-webis-touche2020-v2 | all-MiniLM-L6-v2 | - | - | - | - | - | 0.4325 | - | - | - | - | - |
|  | all-mpnet-base-v2 | - | - | - | - | - | 0.3204 | - | - | - | - | - |
|  | all-roberta-large-v1 | - | - | - | - | - | 0.2835 | - | - | - | - | - |
| car-v1.5-test200 | all-MiniLM-L6-v2 | 0.4097 | 0.4118 | 0.4129 | 0.4139 | 0.4144 | - | - | - | - | - | - |
|  | all-mpnet-base-v2 | 0.3017 | 0.3022 | 0.3023 | 0.3028 | 0.3014 | - | - | - | - | - | - |
|  | all-roberta-large-v1 | 0.2647 | 0.2660 | 0.2657 | 0.2658 | 0.2655 | - | - | - | - | - | - |
| clinicaltrials-2021-trec-ct-2021 | all-MiniLM-L6-v2 | - | - | - | - | - | - | - | - | 0.2838 | 0.2889 | - |
|  | all-mpnet-base-v2 | - | - | - | - | - | - | - | - | 0.3355 | 0.3449 | - |
|  | all-roberta-large-v1 | - | - | - | - | - | - | - | - | 0.3018 | 0.3078 | - |
| clueweb12-touche-2022-task-2 | all-MiniLM-L6-v2 | - | - | - | - | 0.2720 | 0.2776 | - | - | - | - | - |
|  | all-mpnet-base-v2 | - | - | - | - | 0.3210 | 0.3243 | - | - | - | - | - |
|  | all-roberta-large-v1 | - | - | - | - | 0.2792 | 0.2817 | - | - | - | - | - |
| cranfield | all-MiniLM-L6-v2 | - | 0.3900 | 0.3917 | 0.3913 | 0.3879 | 0.3926 | - | - | - | - | - |
|  | all-mpnet-base-v2 | - | 0.2797 | 0.2796 | 0.2794 | 0.2765 | 0.2851 | - | - | - | - | - |
|  | all-roberta-large-v1 | - | 0.2249 | 0.2253 | 0.2242 | 0.2289 | 0.2263 | - | - | - | - | - |
| csl-trec-2023 | all-MiniLM-L6-v2 | - | - | - | - | - | - | 0.3491 | 0.3481 | 0.3583 | 0.3360 | - |
|  | all-mpnet-base-v2 | - | - | - | - | - | - | 0.2557 | 0.2587 | 0.2647 | 0.2472 | - |
|  | all-roberta-large-v1 | - | - | - | - | - | - | 0.2363 | 0.2468 | 0.2539 | 0.2257 | - |
| istella22-test | all-MiniLM-L6-v2 | 0.3756 | 0.3819 | 0.3834 | 0.3835 | - | - | - | - | - | - | - |
|  | all-mpnet-base-v2 | 0.2898 | 0.2882 | 0.2931 | 0.2944 | - | - | - | - | - | - | - |
|  | all-roberta-large-v1 | 0.2672 | 0.2655 | 0.2693 | 0.2678 | - | - | - | - | - | - | - |
| lotte-lifestyle-dev-forum | all-MiniLM-L6-v2 | 0.4106 | 0.4122 | 0.4130 | 0.4125 | 0.4128 | 0.4137 | 0.4189 | - | - | - | - |
|  | all-mpnet-base-v2 | 0.3010 | 0.3018 | 0.3020 | 0.3016 | 0.3020 | 0.3012 | 0.3020 | - | - | - | - |

| Dataset | Encoder | 1 | 2 | 4 | 8 | 16 | 32 | 64 | 128 | 256 | 512 | 1024 |
|---|---|---|---|---|---|---|---|---|---|---|---|---|
| | all-roberta-large-v1 | 0.2596 | 0.2593 | 0.2598 | 0.2602 | 0.2615 | 0.2614 | 0.2613 | - | - | - | - |
| | all-MiniLM-L6-v2 | 0.4120 | 0.4120 | 0.4131 | 0.4143 | 0.4162 | 0.4230 | - | - | - | - | - |
| lotte-lifestyle-test-forum | all-mpnet-base-v2 | 0.3022 | 0.3025 | 0.3030 | 0.3032 | 0.3039 | 0.3053 | - | - | - | - | - |
| | all-roberta-large-v1 | 0.2615 | 0.2615 | 0.2618 | 0.2621 | 0.2632 | 0.2627 | - | - | - | - | - |
| | all-MiniLM-L6-v2 | 0.4150 | 0.4158 | 0.4176 | 0.4161 | - | - | - | - | - | - | - |
| lotte-lifestyle-test-search | all-mpnet-base-v2 | 0.3081 | 0.3075 | 0.3103 | 0.3061 | - | - | - | - | - | - | - |
| | all-roberta-large-v1 | 0.2598 | 0.2597 | 0.2600 | 0.2599 | - | - | - | - | - | - | - |
| | all-MiniLM-L6-v2 | 0.4109 | 0.4118 | 0.4130 | 0.4144 | 0.4161 | 0.4193 | 0.4218 | 0.4200 | 0.4250 | - | - |
| lotte-pooled-test-forum | all-mpnet-base-v2 | 0.3008 | 0.3016 | 0.3019 | 0.3022 | 0.3027 | 0.3038 | 0.3054 | 0.3051 | 0.3042 | - | - |
| | all-roberta-large-v1 | 0.2620 | 0.2622 | 0.2623 | 0.2629 | 0.2634 | 0.2646 | 0.2654 | 0.2663 | 0.2652 | - | - |
| | all-MiniLM-L6-v2 | 0.4140 | 0.4158 | 0.4169 | 0.4167 | 0.4180 | 0.4217 | - | - | - | - | - |
| lotte-pooled-test-search | all-mpnet-base-v2 | 0.3047 | 0.3057 | 0.3066 | 0.3064 | 0.3075 | 0.3117 | - | - | - | - | - |
| | all-roberta-large-v1 | 0.2623 | 0.2627 | 0.2630 | 0.2631 | 0.2647 | 0.2648 | - | - | - | - | - |
| | all-MiniLM-L6-v2 | 0.4094 | 0.4120 | 0.4122 | 0.4127 | 0.4155 | - | - | - | - | - | - |
| lotte-recreation-test-forum | all-mpnet-base-v2 | 0.3016 | 0.3022 | 0.3023 | 0.3024 | 0.3025 | - | - | - | - | - | - |
| | all-roberta-large-v1 | 0.2586 | 0.2600 | 0.2605 | 0.2604 | 0.2586 | - | - | - | - | - | - |
| | all-MiniLM-L6-v2 | 0.4174 | 0.4190 | 0.4199 | 0.4184 | - | - | - | - | - | - | - |
| lotte-recreation-test-search | all-mpnet-base-v2 | 0.3066 | 0.3074 | 0.3085 | 0.3068 | - | - | - | - | - | - | - |
| | all-roberta-large-v1 | 0.2620 | 0.2622 | 0.2633 | 0.2631 | - | - | - | - | - | - | - |
| | all-MiniLM-L6-v2 | 0.4079 | 0.4104 | 0.4124 | 0.4142 | 0.4174 | 0.4236 | 0.4283 | 0.4275 | - | - | - |
| lotte-science-test-forum | all-mpnet-base-v2 | 0.2996 | 0.3006 | 0.3017 | 0.3019 | 0.3037 | 0.3068 | 0.3106 | 0.3085 | - | - | - |
| | all-roberta-large-v1 | 0.2608 | 0.2603 | 0.2617 | 0.2617 | 0.2627 | 0.2657 | 0.2680 | 0.2680 | - | - | - |
| | all-MiniLM-L6-v2 | 0.4150 | 0.4175 | 0.4169 | 0.4173 | 0.4214 | - | - | - | - | - | - |
| lotte-science-test-search | all-mpnet-base-v2 | 0.3064 | 0.3068 | 0.3055 | 0.3068 | 0.3071 | - | - | - | - | - | - |
| | all-roberta-large-v1 | 0.2651 | 0.2650 | 0.2648 | 0.2668 | 0.2699 | - | - | - | - | - | - |
| | all-MiniLM-L6-v2 | 0.4078 | 0.4101 | 0.4108 | 0.4117 | 0.4134 | 0.4182 | 0.4192 | 0.4212 | 0.4247 | - | - |
| lotte-technology-test-forum | all-mpnet-base-v2 | 0.3002 | 0.3011 | 0.3016 | 0.3017 | 0.3021 | 0.3045 | 0.3055 | 0.3065 | 0.3051 | - | - |
| | all-roberta-large-v1 | 0.2584 | 0.2580 | 0.2587 | 0.2593 | 0.2603 | 0.2623 | 0.2621 | 0.2636 | 0.2644 | - | - |
| | all-MiniLM-L6-v2 | 0.4108 | 0.4135 | 0.4149 | 0.4157 | 0.4165 | 0.4196 | - | - | - | - | - |
| lotte-technology-test-search | all-mpnet-base-v2 | 0.3028 | 0.3043 | 0.3053 | 0.3058 | 0.3062 | 0.3088 | - | - | - | - | - |
| | all-roberta-large-v1 | 0.2619 | 0.2630 | 0.2636 | 0.2639 | 0.2656 | 0.2656 | - | - | - | - | - |
| | all-MiniLM-L6-v2 | 0.4129 | 0.4131 | 0.4150 | 0.4182 | 0.4202 | 0.4213 | 0.4328 | - | - | - | - |
| lotte-writing-test-forum | all-mpnet-base-v2 | 0.3031 | 0.3037 | 0.3047 | 0.3042 | 0.3038 | 0.3050 | 0.3047 | - | - | - | - |
| | all-roberta-large-v1 | 0.2573 | 0.2566 | 0.2582 | 0.2591 | 0.2597 | 0.2634 | 0.2654 | - | - | - | - |
| | all-MiniLM-L6-v2 | 0.4184 | 0.4197 | 0.4189 | 0.4193 | 0.4218 | - | - | - | - | - | - |
| lotte-writing-test-search | all-mpnet-base-v2 | 0.3099 | 0.3107 | 0.3108 | 0.3122 | 0.3129 | - | - | - | - | - | - |
| | all-roberta-large-v1 | 0.2581 | 0.2590 | 0.2596 | 0.2607 | 0.2609 | - | - | - | - | - | - |
| | all-MiniLM-L6-v2 | - | - | 0.3656 | 0.3721 | 0.3653 | - | - | - | - | - | - |
| miracl-ar-dev | all-mpnet-base-v2 | - | - | 0.4604 | 0.5101 | 0.4913 | - | - | - | - | - | - |
| | all-roberta-large-v1 | - | - | 0.3818 | 0.3730 | 0.3669 | - | - | - | - | - | - |
| | all-MiniLM-L6-v2 | - | - | 0.4898 | 0.4507 | - | - | - | - | - | - | - |
| miracl-bn-dev | all-mpnet-base-v2 | - | - | 0.4025 | 0.3681 | - | - | - | - | - | - | - |
| | all-roberta-large-v1 | - | - | 0.5127 | 0.3439 | - | - | - | - | - | - | - |
| | all-MiniLM-L6-v2 | - | - | - | 0.4410 | 0.4399 | - | - | - | - | - | - |
| miracl-de-dev | all-mpnet-base-v2 | - | - | - | 0.3189 | 0.3204 | - | - | - | - | - | - |
| | all-roberta-large-v1 | - | - | - | 0.2755 | 0.2782 | - | - | - | - | - | - |
| | all-MiniLM-L6-v2 | - | - | - | 0.4326 | 0.4280 | - | - | - | - | - | - |
| miracl-en-dev | all-mpnet-base-v2 | - | - | - | 0.3232 | 0.3195 | - | - | - | - | - | - |
| | all-roberta-large-v1 | - | - | - | 0.2821 | 0.2727 | - | - | - | - | - | - |
| | all-MiniLM-L6-v2 | - | 0.4678 | 0.4674 | 0.4735 | - | - | - | - | - | - | - |
| miracl-es-dev | all-mpnet-base-v2 | - | 0.3357 | 0.3389 | 0.3415 | - | - | - | - | - | - | - |
| | all-roberta-large-v1 | - | 0.2787 | 0.2798 | 0.2791 | - | - | - | - | - | - | - |
| | all-MiniLM-L6-v2 | - | - | - | 0.4673 | 0.4790 | - | - | - | - | - | - |
| miracl-fa-dev | all-mpnet-base-v2 | - | - | - | 0.4526 | 0.4862 | - | - | - | - | - | - |
| | all-roberta-large-v1 | - | - | - | 0.3584 | 0.3741 | - | - | - | - | - | - |
| | all-MiniLM-L6-v2 | 0.4145 | 0.4145 | 0.4495 | 0.4145 | 0.4169 | - | - | - | - | - | - |
| miracl-fi-dev | all-mpnet-base-v2 | 0.3141 | 0.3147 | 0.3398 | 0.3186 | 0.3292 | - | - | - | - | - | - |
| | all-roberta-large-v1 | 0.2858 | 0.2920 | 0.2907 | 0.2915 | 0.3015 | - | - | - | - | - | - |
| | all-MiniLM-L6-v2 | - | - | - | 0.4518 | - | - | - | - | - | - | - |
| miracl-fr-dev | all-mpnet-base-v2 | - | - | - | 0.3312 | - | - | - | - | - | - | - |
| | all-roberta-large-v1 | - | - | - | 0.2897 | - | - | - | - | - | - | - |
| | all-MiniLM-L6-v2 | - | - | 0.4176 | 0.4211 | - | - | - | - | - | - | - |
| miracl-hi-dev | all-mpnet-base-v2 | - | - | 0.2764 | 0.3430 | - | - | - | - | - | - | - |
| | all-roberta-large-v1 | - | - | 0.2677 | 0.3082 | - | - | - | - | - | - | - |
| | all-MiniLM-L6-v2 | - | 0.4236 | 0.4195 | 0.4254 | 0.3969 | - | - | - | - | - | - |
| miracl-id-dev | all-mpnet-base-v2 | - | 0.3271 | 0.3275 | 0.3261 | 0.2950 | - | - | - | - | - | - |
| | all-roberta-large-v1 | - | 0.3137 | 0.3294 | 0.3065 | 0.2621 | - | - | - | - | - | - |
| | all-MiniLM-L6-v2 | 0.4936 | 0.5160 | - | 0.4780 | 0.4668 | - | - | - | - | - | - |
| miracl-ja-dev | all-mpnet-base-v2 | 0.3311 | 0.3337 | - | 0.3235 | 0.3280 | - | - | - | - | - | - |
| | all-roberta-large-v1 | 0.2811 | 0.2872 | - | 0.2747 | 0.2949 | - | - | - | - | - | - |
| | all-MiniLM-L6-v2 | - | - | - | 0.4881 | 0.4953 | - | - | - | - | - | - |
| miracl-ko-dev | all-mpnet-base-v2 | - | - | - | 0.4290 | 0.4293 | - | - | - | - | - | - |
| | all-roberta-large-v1 | - | - | - | 0.5820 | 0.5710 | - | - | - | - | - | - |
| | all-MiniLM-L6-v2 | - | - | - | 0.3476 | 0.3534 | - | - | - | - | - | - |
| miracl-ru-dev | all-mpnet-base-v2 | - | - | - | 0.2780 | 0.2773 | - | - | - | - | - | - |
| | all-roberta-large-v1 | - | - | - | 0.2618 | 0.2603 | - | - | - | - | - | - |
| | all-MiniLM-L6-v2 | 0.4251 | 0.3793 | 0.4197 | 0.4028 | 0.3995 | - | - | - | - | - | - |
| miracl-sw-dev | all-mpnet-base-v2 | 0.3182 | 0.3119 | 0.3229 | 0.3123 | 0.3030 | - | - | - | - | - | - |
| | all-roberta-large-v1 | 0.2808 | 0.2611 | 0.2652 | 0.2661 | 0.2831 | - | - | - | - | - | - |
| | all-MiniLM-L6-v2 | 1.1169 | - | 0.3764 | 1.0614 | - | - | - | - | - | - | - |
| miracl-te-dev | all-mpnet-base-v2 | 1.1058 | - | 0.3410 | 1.1580 | - | - | - | - | - | - | - |
| | all-roberta-large-v1 | 0.5029 | - | 0.3052 | 0.4989 | - | - | - | - | - | - | - |
| | all-MiniLM-L6-v2 | 0.5452 | 0.6405 | 0.5715 | 0.5824 | - | - | - | - | - | - | - |
| miracl-th-dev | all-mpnet-base-v2 | 0.3995 | 0.4333 | 0.4013 | 0.3969 | - | - | - | - | - | - | - |
| | all-roberta-large-v1 | 0.3925 | 0.3454 | 0.3674 | 0.4009 | - | - | - | - | - | - | - |
| | all-MiniLM-L6-v2 | - | - | - | 0.4139 | - | - | - | - | - | - | - |
| miracl-yo-dev | all-mpnet-base-v2 | - | - | - | 0.3167 | - | - | - | - | - | - | - |
| | all-roberta-large-v1 | - | - | - | 0.2648 | - | - | - | - | - | - | - |
| | all-MiniLM-L6-v2 | - | - | - | 0.4446 | - | - | - | - | - | - | - |
| miracl-zh-dev | all-mpnet-base-v2 | - | - | - | 0.3074 | - | - | - | - | - | - | - |

| Dataset | Encoder | 1 | 2 | 4 | 8 | 16 | 32 | 64 | 128 | 256 | 512 | 1024 |
|---|---|---|---|---|---|---|---|---|---|---|---|---|
| | all-roberta-large-v1 | - | - | - | 0.3638 | - | - | - | - | - | - | - |
| | all-MiniLM-L6-v2 | 0.4296 | 0.4271 | 0.4292 | - | - | - | - | - | - | - | - |
| mmarco-de-dev | all-mpnet-base-v2 | 0.3176 | 0.3157 | 0.3162 | - | - | - | - | - | - | - | - |
| | all-roberta-large-v1 | 0.2807 | 0.2764 | 0.2768 | - | - | - | - | - | - | - | - |
| | all-MiniLM-L6-v2 | 0.4502 | 0.4445 | 0.4468 | - | - | - | - | - | - | - | - |
| mmarco-es-dev | all-mpnet-base-v2 | 0.3380 | 0.3337 | 0.3346 | - | - | - | - | - | - | - | - |
| | all-roberta-large-v1 | 0.2829 | 0.2816 | 0.2799 | - | - | - | - | - | - | - | - |
| | all-MiniLM-L6-v2 | 0.4411 | 0.4406 | 0.4470 | - | - | - | - | - | - | - | - |
| mmarco-fr-dev | all-mpnet-base-v2 | 0.3337 | 0.3322 | 0.3344 | - | - | - | - | - | - | - | - |
| | all-roberta-large-v1 | 0.2830 | 0.2810 | 0.2822 | - | - | - | - | - | - | - | - |
| | all-MiniLM-L6-v2 | 0.4028 | 0.4112 | 0.4043 | - | - | - | - | - | - | - | - |
| mmarco-id-dev | all-mpnet-base-v2 | 0.2957 | 0.2999 | 0.3017 | - | - | - | - | - | - | - | - |
| | all-roberta-large-v1 | 0.2708 | 0.2783 | 0.2825 | - | - | - | - | - | - | - | - |
| | all-MiniLM-L6-v2 | 0.4243 | 0.4208 | 0.4209 | - | - | - | - | - | - | - | - |
| mmarco-it-dev | all-mpnet-base-v2 | 0.3348 | 0.3326 | 0.3325 | - | - | - | - | - | - | - | - |
| | all-roberta-large-v1 | 0.2887 | 0.2843 | 0.2856 | - | - | - | - | - | - | - | - |
| | all-MiniLM-L6-v2 | 0.4397 | 0.4318 | 0.4391 | - | - | - | - | - | - | - | - |
| mmarco-pt-dev | all-mpnet-base-v2 | 0.3276 | 0.3211 | 0.3259 | - | - | - | - | - | - | - | - |
| | all-roberta-large-v1 | 0.2941 | 0.2896 | 0.2941 | - | - | - | - | - | - | - | - |
| | all-MiniLM-L6-v2 | 0.3689 | 0.3688 | 0.3613 | - | - | - | - | - | - | - | - |
| mmarco-ru-dev | all-mpnet-base-v2 | 0.2986 | 0.2946 | 0.2899 | - | - | - | - | - | - | - | - |
| | all-roberta-large-v1 | 0.2878 | 0.2756 | 0.2717 | - | - | - | - | - | - | - | - |
| | all-MiniLM-L6-v2 | 0.8436 | 0.7491 | - | - | - | - | - | - | - | - | - |
| mr-tydi-ar-test | all-mpnet-base-v2 | 0.4937 | 0.4963 | - | - | - | - | - | - | - | - | - |
| | all-roberta-large-v1 | 0.3642 | 0.3635 | - | - | - | - | - | - | - | - | - |
| | all-MiniLM-L6-v2 | 0.7608 | 0.7566 | - | - | - | - | - | - | - | - | - |
| mr-tydi-bn-test | all-mpnet-base-v2 | 0.2579 | 0.2520 | - | - | - | - | - | - | - | - | - |
| | all-roberta-large-v1 | 0.3160 | 0.3291 | - | - | - | - | - | - | - | - | - |
| | all-MiniLM-L6-v2 | 1.0570 | 1.0489 | - | - | - | - | - | - | - | - | - |
| mr-tydi-en-test | all-mpnet-base-v2 | 0.3235 | 0.3220 | - | - | - | - | - | - | - | - | - |
| | all-roberta-large-v1 | 0.2827 | 0.2822 | - | - | - | - | - | - | - | - | - |
| | all-MiniLM-L6-v2 | 1.0114 | 0.9412 | - | - | - | - | - | - | - | - | - |
| mr-tydi-fi-test | all-mpnet-base-v2 | 0.3230 | 0.3189 | - | - | - | - | - | - | - | - | - |
| | all-roberta-large-v1 | 0.2933 | 0.2918 | - | - | - | - | - | - | - | - | - |
| | all-MiniLM-L6-v2 | 1.0252 | 0.9791 | - | - | - | - | - | - | - | - | - |
| mr-tydi-id-test | all-mpnet-base-v2 | 0.3289 | 0.3310 | - | - | - | - | - | - | - | - | - |
| | all-roberta-large-v1 | 0.2958 | 0.3010 | - | - | - | - | - | - | - | - | - |
| | all-MiniLM-L6-v2 | 1.2202 | 1.1451 | - | - | - | - | - | - | - | - | - |
| mr-tydi-ja-test | all-mpnet-base-v2 | 0.3324 | 0.3289 | - | - | - | - | - | - | - | - | - |
| | all-roberta-large-v1 | 0.2893 | 0.2824 | - | - | - | - | - | - | - | - | - |
| | all-MiniLM-L6-v2 | 0.9719 | 0.9485 | - | - | - | - | - | - | - | - | - |
| mr-tydi-ko-test | all-mpnet-base-v2 | 0.4082 | 0.4190 | - | - | - | - | - | - | - | - | - |
| | all-roberta-large-v1 | 0.5299 | 0.5542 | - | - | - | - | - | - | - | - | - |
| | all-MiniLM-L6-v2 | 0.8008 | 0.7448 | - | - | - | - | - | - | - | - | - |
| mr-tydi-ru-test | all-mpnet-base-v2 | 0.2839 | 0.2806 | - | - | - | - | - | - | - | - | - |
| | all-roberta-large-v1 | 0.2703 | 0.2685 | - | - | - | - | - | - | - | - | - |
| | all-MiniLM-L6-v2 | 1.0325 | 0.9230 | - | - | - | - | - | - | - | - | - |
| mr-tydi-sw-test | all-mpnet-base-v2 | 0.3392 | 0.3243 | - | - | - | - | - | - | - | - | - |
| | all-roberta-large-v1 | 0.2883 | 0.2799 | - | - | - | - | - | - | - | - | - |
| | all-MiniLM-L6-v2 | 2.0689 | 2.2970 | - | - | - | - | - | - | - | - | - |
| mr-tydi-te-test | all-mpnet-base-v2 | 0.8633 | 1.0369 | - | - | - | - | - | - | - | - | - |
| | all-roberta-large-v1 | 0.4493 | 0.4743 | - | - | - | - | - | - | - | - | - |
| | all-MiniLM-L6-v2 | 1.7386 | 1.2216 | - | - | - | - | - | - | - | - | - |
| mr-tydi-th-test | all-mpnet-base-v2 | 0.4721 | 0.3958 | - | - | - | - | - | - | - | - | - |
| | all-roberta-large-v1 | 0.4313 | 0.4155 | - | - | - | - | - | - | - | - | - |
| | all-MiniLM-L6-v2 | - | - | - | - | - | - | - | 0.8831 | 0.8993 | 0.8955 | 0.8594 |
| msmarco-document-trec-dl-2019 | all-mpnet-base-v2 | - | - | - | - | - | - | - | 0.2959 | 0.2976 | 0.2961 | 0.2972 |
| | all-roberta-large-v1 | - | - | - | - | - | - | - | 0.2562 | 0.2565 | 0.2549 | 0.2576 |
| | all-MiniLM-L6-v2 | - | - | - | - | - | - | - | 0.8846 | 0.8763 | - | - |
| msmarco-document-trec-dl-2020 | all-mpnet-base-v2 | - | - | - | - | - | - | - | 0.2980 | 0.2966 | - | - |
| | all-roberta-large-v1 | - | - | - | - | - | - | - | 0.2572 | 0.2570 | - | - |
| | all-MiniLM-L6-v2 | - | 0.9460 | 0.9307 | - | - | - | - | 0.9064 | 0.9081 | - | - |
| msmarco-document-trec-dl-hard-fold1 | all-mpnet-base-v2 | - | 0.2923 | 0.3001 | - | - | - | - | 0.2988 | 0.2991 | - | - |
| | all-roberta-large-v1 | - | 0.2547 | 0.2587 | - | - | - | - | 0.2598 | 0.2588 | - | - |
| | all-MiniLM-L6-v2 | - | - | 0.9591 | - | - | - | - | 0.8840 | 0.9144 | - | - |
| msmarco-document-trec-dl-hard-fold2 | all-mpnet-base-v2 | - | - | 0.2969 | - | - | - | - | 0.2923 | 0.2930 | - | - |
| | all-roberta-large-v1 | - | - | 0.2544 | - | - | - | - | 0.2526 | 0.2554 | - | - |
| | all-MiniLM-L6-v2 | - | 0.9374 | 0.9528 | 0.9828 | - | - | - | 0.8860 | - | - | - |
| msmarco-document-trec-dl-hard-fold3 | all-mpnet-base-v2 | - | 0.2978 | 0.2973 | 0.2979 | - | - | - | 0.2976 | - | - | - |
| | all-roberta-large-v1 | - | 0.2489 | 0.2557 | 0.2536 | - | - | - | 0.2564 | - | - | - |
| | all-MiniLM-L6-v2 | - | 0.9645 | 0.9171 | - | - | - | - | 0.9169 | 0.9172 | - | - |
| msmarco-document-trec-dl-hard-fold4 | all-mpnet-base-v2 | - | 0.3006 | 0.2955 | - | - | - | - | 0.3006 | 0.2998 | - | - |
| | all-roberta-large-v1 | - | 0.2565 | 0.2574 | - | - | - | - | 0.2526 | 0.2596 | - | - |
| | all-MiniLM-L6-v2 | - | - | 0.9782 | - | - | - | - | 0.8724 | 0.8774 | 0.9023 | 0.8561 |
| msmarco-document-trec-dl-hard-fold5 | all-mpnet-base-v2 | - | - | 0.2959 | - | - | - | - | 0.2952 | 0.2975 | 0.2968 | 0.2941 |
| | all-roberta-large-v1 | - | - | 0.2548 | - | - | - | - | 0.2574 | 0.2555 | 0.2548 | 0.2557 |
| | all-MiniLM-L6-v2 | - | - | - | - | - | - | 0.9243 | 0.9177 | 0.9260 | 0.9083 | 0.8789 |
| msmarco-document-v2-trec-dl-2019 | all-mpnet-base-v2 | - | - | - | - | - | - | 0.2975 | 0.2963 | 0.2981 | 0.2969 | 0.2972 |
| | all-roberta-large-v1 | - | - | - | - | - | - | 0.2563 | 0.2535 | 0.2541 | 0.2539 | 0.2522 |
| | all-MiniLM-L6-v2 | - | - | - | - | - | - | 0.9272 | 0.9143 | 0.9046 | - | - |
| msmarco-document-v2-trec-dl-2020 | all-mpnet-base-v2 | - | - | - | - | - | - | 0.2985 | 0.2985 | 0.2984 | - | - |
| | all-roberta-large-v1 | - | - | - | - | - | - | 0.2546 | 0.2557 | 0.2552 | - | - |
| | all-MiniLM-L6-v2 | - | - | - | - | - | - | 0.9576 | 0.9492 | 0.9554 | 0.9598 | - |
| msmarco-document-v2-trec-dl-2021 | all-mpnet-base-v2 | - | - | - | - | - | - | 0.3000 | 0.2977 | 0.2977 | 0.2975 | - |
| | all-roberta-large-v1 | - | - | - | - | - | - | 0.2547 | 0.2547 | 0.2541 | 0.2555 | - |
| | all-MiniLM-L6-v2 | - | - | - | - | - | - | - | 0.9936 | 1.0010 | - | - |
| msmarco-passage-trec-dl-2020 | all-mpnet-base-v2 | - | - | - | - | - | - | - | 0.3016 | 0.3008 | - | - |
| | all-roberta-large-v1 | - | - | - | - | - | - | - | 0.2624 | 0.2627 | - | - |
| | all-MiniLM-L6-v2 | - | - | 1.0031 | 0.9983 | - | - | - | 0.9901 | 0.9867 | - | - |
| msmarco-passage-trec-dl-hard | all-mpnet-base-v2 | - | - | 0.3003 | 0.3016 | - | - | - | 0.3022 | 0.3006 | - | - |

| Dataset | Encoder | 1 | 2 | 4 | 8 | 16 | 32 | 64 | 128 | 256 | 512 | 1024 |
|---|---|---|---|---|---|---|---|---|---|---|---|---|
| | all-roberta-large-v1 | - | - | 0.2646 | 0.2634 | - | - | - | 0.2634 | 0.2620 | - | - |
| msmarco-qna-dev | all-MiniLM-L6-v2 | 0.9992 | 1.0049 | 1.0007 | 1.0018 | 1.0019 | - | - | - | - | - | - |
| | all-mpnet-base-v2 | 0.2994 | 0.3002 | 0.3005 | 0.3007 | 0.3005 | - | - | - | - | - | - |
| | all-roberta-large-v1 | 0.2602 | 0.2620 | 0.2619 | 0.2620 | 0.2618 | - | - | - | - | - | - |
| natural-questions-dev | all-MiniLM-L6-v2 | 0.9993 | 0.9960 | 0.9951 | - | - | - | - | - | - | - | - |
| | all-mpnet-base-v2 | 0.2969 | 0.2972 | 0.2971 | - | - | - | - | - | - | - | - |
| | all-roberta-large-v1 | 0.2609 | 0.2610 | 0.2607 | - | - | - | - | - | - | - | - |
| pmc-v1-trec-cds-2014 | all-MiniLM-L6-v2 | - | - | - | - | - | - | - | - | - | 0.7506 | 0.7304 |
| | all-mpnet-base-v2 | - | - | - | - | - | - | - | - | - | 0.3334 | 0.3482 |
| | all-roberta-large-v1 | - | - | - | - | - | - | - | - | - | 0.3011 | 0.3202 |
| pmc-v1-trec-cds-2015 | all-MiniLM-L6-v2 | - | - | - | - | - | - | - | - | - | 0.8911 | 0.8315 |
| | all-mpnet-base-v2 | - | - | - | - | - | - | - | - | - | 0.3396 | 0.3685 |
| | all-roberta-large-v1 | - | - | - | - | - | - | - | - | - | 0.3041 | 0.3310 |
| pmc-v2-trec-cds-2016 | all-MiniLM-L6-v2 | - | - | - | - | - | - | - | - | - | 0.8121 | 0.7413 |
| | all-mpnet-base-v2 | - | - | - | - | - | - | - | - | - | 0.4148 | 0.4006 |
| | all-roberta-large-v1 | - | - | - | - | - | - | - | - | - | 0.3735 | 0.3653 |
| trec-tot-2023-dev | all-MiniLM-L6-v2 | 1.1284 | - | - | - | - | - | - | - | - | - | - |
| | all-mpnet-base-v2 | 0.3374 | - | - | - | - | - | - | - | - | - | - |
| | all-roberta-large-v1 | 0.3029 | - | - | - | - | - | - | - | - | - | - |
| vaswani | all-MiniLM-L6-v2 | 0.9931 | 0.9366 | 0.9209 | 0.9150 | 0.9220 | 0.9055 | 0.9218 | - | - | - | - |
| | all-mpnet-base-v2 | 0.2941 | 0.2940 | 0.2907 | 0.2927 | 0.2928 | 0.2934 | 0.2934 | - | - | - | - |
| | all-roberta-large-v1 | 0.2366 | 0.2361 | 0.2336 | 0.2365 | 0.2376 | 0.2351 | 0.2366 | - | - | - | - |

*Table 6.* Median sensitivity value per log bin for $K_3$

| Dataset | Encoder | 1 | 2 | 4 | 8 | 16 | 32 | 64 | 128 | 256 | 512 | 1024 |
|---|---|---|---|---|---|---|---|---|---|---|---|---|
| **misc** | | | | | | | | | | | | |
| | all-MiniLM-L6-v2 | - | 0.4499 | 0.4637 | 0.4662 | 0.4632 | 0.4669 | 0.4561 | - | - | - | - |
| antique-test | all-mpnet-base-v2 | - | 0.3190 | 0.3289 | 0.3292 | 0.3293 | 0.3314 | 0.3252 | - | - | - | - |
| | all-roberta-large-v1 | - | 0.2810 | 0.2864 | 0.2878 | 0.2864 | 0.2875 | 0.2814 | - | - | - | - |
| | all-MiniLM-L6-v2 | 0.4151 | - | - | - | - | - | - | - | - | - | - |
| beir-arguana | all-mpnet-base-v2 | 0.3158 | - | - | - | - | - | - | - | - | - | - |
| | all-roberta-large-v1 | 0.2665 | - | - | - | - | - | - | - | - | - | - |
| | all-MiniLM-L6-v2 | 0.4259 | 0.4195 | 0.4279 | - | - | - | - | - | - | - | - |
| beir-climate-fever | all-mpnet-base-v2 | 0.3017 | 0.2991 | 0.3033 | - | - | - | - | - | - | - | - |
| | all-roberta-large-v1 | 0.2827 | 0.2782 | 0.3190 | - | - | - | - | - | - | - | - |
| | all-MiniLM-L6-v2 | 0.4330 | 0.4316 | 0.4338 | 0.4332 | 0.4345 | 0.4348 | - | - | 0.4337 | - | - |
| beir-cqadupstack-android | all-mpnet-base-v2 | 0.3221 | 0.3164 | 0.3186 | 0.3187 | 0.3214 | 0.3207 | - | - | 0.3148 | - | - |
| | all-roberta-large-v1 | 0.2782 | 0.2732 | 0.2741 | 0.2738 | 0.2756 | 0.2754 | - | - | 0.2713 | - | - |
| | all-MiniLM-L6-v2 | 0.4472 | 0.4490 | 0.4512 | 0.4545 | 0.4510 | 0.4558 | 0.4509 | - | - | - | - |
| beir-cqadupstack-english | all-mpnet-base-v2 | 0.3217 | 0.3218 | 0.3225 | 0.3257 | 0.3232 | 0.3241 | 0.3211 | - | - | - | - |
| | all-roberta-large-v1 | 0.2800 | 0.2789 | 0.2809 | 0.2822 | 0.2815 | 0.2841 | 0.2796 | - | - | - | - |
| | all-MiniLM-L6-v2 | 0.4437 | 0.4447 | 0.4488 | 0.4438 | 0.4480 | - | - | - | - | - | - |
| beir-cqadupstack-gaming | all-mpnet-base-v2 | 0.3213 | 0.3231 | 0.3243 | 0.3194 | 0.3241 | - | - | - | - | - | - |
| | all-roberta-large-v1 | 0.2789 | 0.2806 | 0.2811 | 0.2796 | 0.2811 | - | - | - | - | - | - |
| | all-MiniLM-L6-v2 | 0.4493 | 0.4483 | 0.4532 | 0.4412 | 0.4350 | - | - | - | - | - | - |
| beir-cqadupstack-gis | all-mpnet-base-v2 | 0.3192 | 0.3201 | 0.3238 | 0.3149 | 0.3050 | - | - | - | - | - | - |
| | all-roberta-large-v1 | 0.2767 | 0.2761 | 0.2816 | 0.2767 | 0.2703 | - | - | - | - | - | - |
| | all-MiniLM-L6-v2 | 0.4458 | 0.4469 | 0.4520 | 0.4514 | 0.4653 | 0.4475 | - | - | - | - | - |
| beir-cqadupstack-mathematica | all-mpnet-base-v2 | 0.3217 | 0.3230 | 0.3247 | 0.3230 | 0.3383 | 0.3206 | - | - | - | - | - |
| | all-roberta-large-v1 | 0.2767 | 0.2786 | 0.2805 | 0.2795 | 0.2834 | 0.2750 | - | - | - | - | - |
| | all-MiniLM-L6-v2 | 0.4488 | 0.4527 | 0.4500 | 0.4498 | 0.4486 | 0.4453 | 0.4723 | - | - | - | - |
| beir-cqadupstack-physics | all-mpnet-base-v2 | 0.3243 | 0.3285 | 0.3245 | 0.3239 | 0.3235 | 0.3219 | 0.3331 | - | - | - | - |
| | all-roberta-large-v1 | 0.2826 | 0.2851 | 0.2822 | 0.2808 | 0.2791 | 0.2848 | 0.2911 | - | - | - | - |
| | all-MiniLM-L6-v2 | 0.4439 | 0.4419 | 0.4410 | 0.4405 | 0.4424 | 0.4429 | - | 0.4315 | - | - | - |
| beir-cqadupstack-programmers | all-mpnet-base-v2 | 0.3190 | 0.3192 | 0.3186 | 0.3168 | 0.3168 | 0.3189 | - | 0.3118 | - | - | - |
| | all-roberta-large-v1 | 0.2778 | 0.2774 | 0.2774 | 0.2762 | 0.2772 | 0.2783 | - | 0.2727 | - | - | - |
| | all-MiniLM-L6-v2 | 0.4482 | 0.4524 | 0.4603 | 0.4459 | 0.4458 | - | - | - | - | - | - |
| beir-cqadupstack-stats | all-mpnet-base-v2 | 0.3267 | 0.3257 | 0.3310 | 0.3235 | 0.3208 | - | - | - | - | - | - |
| | all-roberta-large-v1 | 0.2820 | 0.2803 | 0.2856 | 0.2779 | 0.2762 | - | - | - | - | - | - |
| | all-MiniLM-L6-v2 | 0.4503 | 0.4494 | 0.4529 | 0.4524 | 0.4502 | 0.4677 | - | 0.4555 | - | - | - |
| beir-cqadupstack-tex | all-mpnet-base-v2 | 0.3199 | 0.3239 | 0.3247 | 0.3238 | 0.3219 | 0.3342 | - | 0.3240 | - | - | - |
| | all-roberta-large-v1 | 0.2740 | 0.2767 | 0.2764 | 0.2757 | 0.2740 | 0.2819 | - | 0.2773 | - | - | - |
| | all-MiniLM-L6-v2 | 0.4423 | 0.4424 | 0.4467 | 0.4441 | 0.4473 | - | - | - | - | - | - |
| beir-cqadupstack-unix | all-mpnet-base-v2 | 0.3236 | 0.3216 | 0.3246 | 0.3216 | 0.3257 | - | - | - | - | - | - |
| | all-roberta-large-v1 | 0.2824 | 0.2798 | 0.2816 | 0.2791 | 0.2795 | - | - | - | - | - | - |
| | all-MiniLM-L6-v2 | 0.4346 | 0.4363 | 0.4313 | 0.4358 | 0.4331 | 0.4258 | 0.4606 | 0.4344 | - | - | - |
| beir-cqadupstack-webmasters | all-mpnet-base-v2 | 0.3176 | 0.3170 | 0.3152 | 0.3174 | 0.3147 | 0.3122 | 0.3327 | 0.3153 | - | - | - |
| | all-roberta-large-v1 | 0.2748 | 0.2753 | 0.2726 | 0.2748 | 0.2715 | 0.2695 | 0.2847 | 0.2766 | - | - | - |
| | all-MiniLM-L6-v2 | 0.4470 | 0.4442 | 0.4449 | 0.4612 | - | 0.4376 | - | - | - | - | - |
| beir-cqadupstack-wordpress | all-mpnet-base-v2 | 0.3171 | 0.3176 | 0.3222 | 0.3375 | - | 0.3158 | - | - | - | - | - |
| | all-roberta-large-v1 | 0.2699 | 0.2723 | 0.2710 | 0.2902 | - | 0.2668 | - | - | - | - | - |
| | all-MiniLM-L6-v2 | - | - | - | - | 0.5090 | 0.4813 | 0.4773 | 0.4781 | 0.4827 | 0.4709 | 0.4715 |
| beir-dbpedia-entity-test | all-mpnet-base-v2 | - | - | - | - | 0.3525 | 0.3393 | 0.3365 | 0.3379 | 0.3398 | 0.3343 | 0.3321 |
| | all-roberta-large-v1 | - | - | - | - | 0.3001 | 0.2906 | 0.2878 | 0.2892 | 0.2896 | 0.2889 | 0.2868 |
| | all-MiniLM-L6-v2 | 0.4430 | 0.4485 | 0.4468 | 0.4525 | - | - | - | - | - | - | - |
| beir-fever-test | all-mpnet-base-v2 | 0.3205 | 0.3218 | 0.3220 | 0.3267 | - | - | - | - | - | - | - |
| | all-roberta-large-v1 | 0.2837 | 0.2853 | 0.2850 | 0.2876 | - | - | - | - | - | - | - |
| | all-MiniLM-L6-v2 | 0.4385 | 0.4400 | 0.4361 | 0.4351 | - | - | - | - | - | - | - |
| beir-fiqa-test | all-mpnet-base-v2 | 0.3234 | 0.3216 | 0.3202 | 0.3210 | - | - | - | - | - | - | - |
| | all-roberta-large-v1 | 0.2695 | 0.2702 | 0.2695 | 0.2696 | - | - | - | - | - | - | - |
| | all-MiniLM-L6-v2 | - | 0.4399 | - | - | - | - | - | - | - | - | - |
| beir-hotpotqa-test | all-mpnet-base-v2 | - | 0.3252 | - | - | - | - | - | - | - | - | - |
| | all-roberta-large-v1 | - | 0.2815 | - | - | - | - | - | - | - | - | - |

| Dataset | Encoder | 1 | 2 | 4 | 8 | 16 | 32 | 64 | 128 | 256 | 512 | 1024 |
|---|---|---|---|---|---|---|---|---|---|---|---|---|
| beir-msmarco-test | all-MiniLM-L6-v2 | - | - | - | - | - | - | - | 0.4600 | 0.4637 | 0.4599 | - |
| | all-mpnet-base-v2 | - | - | - | - | - | - | - | 0.3205 | 0.3229 | 0.3232 | - |
| | all-roberta-large-v1 | - | - | - | - | - | - | - | 0.2794 | 0.2816 | 0.2742 | - |
| beir-nfcorpus-test | all-MiniLM-L6-v2 | 0.4322 | 0.4386 | 0.4330 | 0.4361 | 0.4383 | 0.4358 | 0.4443 | 0.4523 | 0.4695 | - | - |
| | all-mpnet-base-v2 | 0.2973 | 0.3034 | 0.2982 | 0.3024 | 0.3033 | 0.3026 | 0.3075 | 0.3115 | 0.3200 | - | - |
| | all-roberta-large-v1 | 0.2736 | 0.2771 | 0.2741 | 0.2756 | 0.2772 | 0.2767 | 0.2777 | 0.2801 | 0.2874 | - | - |
| beir-nq | all-MiniLM-L6-v2 | 0.4337 | 0.4351 | 0.4357 | - | - | - | - | - | - | - | - |
| | all-mpnet-base-v2 | 0.3094 | 0.3093 | 0.3084 | - | - | - | - | - | - | - | - |
| | all-roberta-large-v1 | 0.2722 | 0.2715 | 0.2707 | - | - | - | - | - | - | - | - |
| beir-quora-test | all-MiniLM-L6-v2 | 0.4443 | 0.4453 | 0.4452 | 0.4462 | 0.4455 | 0.4467 | 0.4673 | - | - | - | - |
| | all-mpnet-base-v2 | 0.3178 | 0.3177 | 0.3173 | 0.3185 | 0.3178 | 0.3204 | 0.3398 | - | - | - | - |
| | all-roberta-large-v1 | 0.2779 | 0.2775 | 0.2768 | 0.2780 | 0.2776 | 0.2792 | 0.2975 | - | - | - | - |
| beir-scidocs | all-MiniLM-L6-v2 | - | - | - | - | 0.4492 | - | - | - | - | - | - |
| | all-mpnet-base-v2 | - | - | - | - | 0.3196 | - | - | - | - | - | - |
| | all-roberta-large-v1 | - | - | - | - | 0.2810 | - | - | - | - | - | - |
| beir-scifact-test | all-MiniLM-L6-v2 | 0.4227 | 0.4246 | 0.4400 | - | - | - | - | - | - | - | - |
| | all-mpnet-base-v2 | 0.3124 | 0.3138 | 0.3197 | - | - | - | - | - | - | - | - |
| | all-roberta-large-v1 | 0.2773 | 0.2779 | 0.2849 | - | - | - | - | - | - | - | - |
| beir-trec-covid | all-MiniLM-L6-v2 | - | - | - | - | - | - | - | - | - | 0.4430 | 0.4380 |
| | all-mpnet-base-v2 | - | - | - | - | - | - | - | - | - | 0.3266 | 0.3200 |
| | all-roberta-large-v1 | - | - | - | - | - | - | - | - | - | 0.2900 | 0.2850 |
| beir-webis-touche2020-v2 | all-MiniLM-L6-v2 | - | - | - | - | - | 0.4600 | - | - | - | - | - |
| | all-mpnet-base-v2 | - | - | - | - | - | 0.3287 | - | - | - | - | - |
| | all-roberta-large-v1 | - | - | - | - | - | 0.2874 | - | - | - | - | - |
| car-v1.5-test200 | all-MiniLM-L6-v2 | 0.4591 | 0.4574 | 0.4589 | 0.4565 | 0.4639 | - | - | - | - | - | - |
| | all-mpnet-base-v2 | 0.3274 | 0.3255 | 0.3256 | 0.3252 | 0.3275 | - | - | - | - | - | - |
| | all-roberta-large-v1 | 0.2876 | 0.2866 | 0.2864 | 0.2858 | 0.2883 | - | - | - | - | - | - |
| clinicaltrials-2021-trec-ct-2021 | all-MiniLM-L6-v2 | - | - | - | - | - | - | - | - | 0.2699 | 0.2708 | - |
| | all-mpnet-base-v2 | - | - | - | - | - | - | - | - | 0.3209 | 0.3255 | - |
| | all-roberta-large-v1 | - | - | - | - | - | - | - | - | 0.2956 | 0.2970 | - |
| clueweb12-touche-2022-task-2 | all-MiniLM-L6-v2 | - | - | - | - | 0.2834 | 0.2876 | - | - | - | - | - |
| | all-mpnet-base-v2 | - | - | - | - | 0.3285 | 0.3322 | - | - | - | - | - |
| | all-roberta-large-v1 | - | - | - | - | 0.2849 | 0.2888 | - | - | - | - | - |
| cranfield | all-MiniLM-L6-v2 | - | 0.4042 | 0.4056 | 0.4040 | 0.3934 | 0.3937 | - | - | - | - | - |
| | all-mpnet-base-v2 | - | 0.2884 | 0.2888 | 0.2864 | 0.2795 | 0.2838 | - | - | - | - | - |
| | all-roberta-large-v1 | - | 0.2312 | 0.2309 | 0.2297 | 0.2293 | 0.2286 | - | - | - | - | - |
| csl-trec-2023 | all-MiniLM-L6-v2 | - | - | - | - | - | - | 0.3845 | 0.3769 | 0.3831 | 0.3692 | - |
| | all-mpnet-base-v2 | - | - | - | - | - | - | 0.2768 | 0.2785 | 0.2829 | 0.2697 | - |
| | all-roberta-large-v1 | - | - | - | - | - | - | 0.2514 | 0.2579 | 0.2616 | 0.2455 | - |
| istella22-test | all-MiniLM-L6-v2 | 0.3766 | 0.3833 | 0.3837 | 0.3821 | - | - | - | - | - | - | - |
| | all-mpnet-base-v2 | 0.2906 | 0.2884 | 0.2922 | 0.2924 | - | - | - | - | - | - | - |
| | all-roberta-large-v1 | 0.2639 | 0.2627 | 0.2652 | 0.2643 | - | - | - | - | - | - | - |
| lotte-lifestyle-dev-forum | all-MiniLM-L6-v2 | 0.4467 | 0.4473 | 0.4448 | 0.4482 | 0.4492 | 0.4393 | 0.4446 | - | - | - | - |
| | all-mpnet-base-v2 | 0.3208 | 0.3217 | 0.3193 | 0.3215 | 0.3224 | 0.3152 | 0.3170 | - | - | - | - |
| | all-roberta-large-v1 | 0.2763 | 0.2755 | 0.2738 | 0.2763 | 0.2780 | 0.2722 | 0.2743 | - | - | - | - |
| lotte-lifestyle-test-forum | all-MiniLM-L6-v2 | 0.4485 | 0.4488 | 0.4469 | 0.4489 | 0.4488 | 0.4470 | - | - | - | - | - |
| | all-mpnet-base-v2 | 0.3236 | 0.3245 | 0.3230 | 0.3237 | 0.3229 | 0.3219 | - | - | - | - | - |
| | all-roberta-large-v1 | 0.2798 | 0.2800 | 0.2790 | 0.2797 | 0.2793 | 0.2776 | - | - | - | - | - |
| lotte-lifestyle-test-search | all-MiniLM-L6-v2 | 0.4426 | 0.4411 | 0.4409 | 0.4416 | - | - | - | - | - | - | - |
| | all-mpnet-base-v2 | 0.3217 | 0.3191 | 0.3209 | 0.3189 | - | - | - | - | - | - | - |
| | all-roberta-large-v1 | 0.2727 | 0.2711 | 0.2713 | 0.2714 | - | - | - | - | - | - | - |
| lotte-pooled-test-forum | all-MiniLM-L6-v2 | 0.4519 | 0.4517 | 0.4551 | 0.4579 | 0.4595 | 0.4607 | 0.4698 | 0.4648 | 0.4573 | - | - |
| | all-mpnet-base-v2 | 0.3239 | 0.3235 | 0.3252 | 0.3266 | 0.3273 | 0.3272 | 0.3337 | 0.3316 | 0.3235 | - | - |
| | all-roberta-large-v1 | 0.2819 | 0.2810 | 0.2822 | 0.2832 | 0.2839 | 0.2850 | 0.2877 | 0.2894 | 0.2814 | - | - |
| lotte-pooled-test-search | all-MiniLM-L6-v2 | 0.4475 | 0.4484 | 0.4511 | 0.4503 | 0.4507 | 0.4622 | - | - | - | - | - |
| | all-mpnet-base-v2 | 0.3188 | 0.3194 | 0.3214 | 0.3204 | 0.3207 | 0.3302 | - | - | - | - | - |
| | all-roberta-large-v1 | 0.2752 | 0.2749 | 0.2759 | 0.2753 | 0.2764 | 0.2831 | - | - | - | - | - |
| lotte-recreation-test-forum | all-MiniLM-L6-v2 | 0.4427 | 0.4453 | 0.4468 | 0.4464 | 0.4488 | - | - | - | - | - | - |
| | all-mpnet-base-v2 | 0.3200 | 0.3207 | 0.3217 | 0.3213 | 0.3207 | - | - | - | - | - | - |
| | all-roberta-large-v1 | 0.2746 | 0.2750 | 0.2762 | 0.2759 | 0.2732 | - | - | - | - | - | - |
| lotte-recreation-test-search | all-MiniLM-L6-v2 | 0.4404 | 0.4407 | 0.4422 | 0.4425 | - | - | - | - | - | - | - |
| | all-mpnet-base-v2 | 0.3183 | 0.3186 | 0.3200 | 0.3177 | - | - | - | - | - | - | - |
| | all-roberta-large-v1 | 0.2733 | 0.2724 | 0.2737 | 0.2717 | - | - | - | - | - | - | - |
| lotte-science-test-forum | all-MiniLM-L6-v2 | 0.4576 | 0.4610 | 0.4592 | 0.4612 | 0.4626 | 0.4681 | 0.4764 | 0.4750 | - | - | - |
| | all-mpnet-base-v2 | 0.3296 | 0.3306 | 0.3287 | 0.3294 | 0.3307 | 0.3346 | 0.3406 | 0.3413 | - | - | - |
| | all-roberta-large-v1 | 0.2852 | 0.2850 | 0.2841 | 0.2846 | 0.2853 | 0.2897 | 0.2911 | 0.2949 | - | - | - |
| lotte-science-test-search | all-MiniLM-L6-v2 | 0.4463 | 0.4488 | 0.4479 | 0.4485 | 0.4473 | - | - | - | - | - | - |
| | all-mpnet-base-v2 | 0.3190 | 0.3190 | 0.3177 | 0.3200 | 0.3163 | - | - | - | - | - | - |
| | all-roberta-large-v1 | 0.2758 | 0.2760 | 0.2754 | 0.2790 | 0.2765 | - | - | - | - | - | - |
| lotte-technology-test-forum | all-MiniLM-L6-v2 | 0.4449 | 0.4468 | 0.4470 | 0.4480 | 0.4492 | 0.4495 | 0.4595 | 0.4581 | 0.4518 | - | - |
| | all-mpnet-base-v2 | 0.3205 | 0.3215 | 0.3213 | 0.3215 | 0.3218 | 0.3212 | 0.3290 | 0.3281 | 0.3220 | - | - |
| | all-roberta-large-v1 | 0.2748 | 0.2749 | 0.2753 | 0.2756 | 0.2766 | 0.2777 | 0.2821 | 0.2836 | 0.2791 | - | - |
| lotte-technology-test-search | all-MiniLM-L6-v2 | 0.4409 | 0.4444 | 0.4468 | 0.4491 | 0.4512 | 0.4639 | - | - | - | - | - |
| | all-mpnet-base-v2 | 0.3173 | 0.3186 | 0.3200 | 0.3207 | 0.3210 | 0.3311 | - | - | - | - | - |
| | all-roberta-large-v1 | 0.2748 | 0.2754 | 0.2762 | 0.2763 | 0.2780 | 0.2853 | - | - | - | - | - |
| lotte-writing-test-forum | all-MiniLM-L6-v2 | 0.4550 | 0.4547 | 0.4543 | 0.4546 | 0.4574 | 0.4580 | 0.4660 | - | - | - | - |
| | all-mpnet-base-v2 | 0.3277 | 0.3274 | 0.3269 | 0.3260 | 0.3272 | 0.3268 | 0.3327 | - | - | - | - |
| | all-roberta-large-v1 | 0.2771 | 0.2766 | 0.2768 | 0.2777 | 0.2788 | 0.2792 | 0.2863 | - | - | - | - |
| lotte-writing-test-search | all-MiniLM-L6-v2 | 0.4529 | 0.4528 | 0.4528 | 0.4529 | 0.4520 | - | - | - | - | - | - |
| | all-mpnet-base-v2 | 0.3229 | 0.3232 | 0.3236 | 0.3245 | 0.3237 | - | - | - | - | - | - |
| | all-roberta-large-v1 | 0.2695 | 0.2696 | 0.2709 | 0.2720 | 0.2712 | - | - | - | - | - | - |
| miracl-ar-dev | all-MiniLM-L6-v2 | - | - | 0.2500 | 0.2619 | 0.2497 | - | - | - | - | - | - |
| | all-mpnet-base-v2 | - | - | 0.2969 | 0.3262 | 0.3251 | - | - | - | - | - | - |
| | all-roberta-large-v1 | - | - | 0.2228 | 0.2333 | 0.2311 | - | - | - | - | - | - |
| miracl-bn-dev | all-MiniLM-L6-v2 | - | - | 0.2917 | 0.2432 | - | - | - | - | - | - | - |
| | all-mpnet-base-v2 | - | - | 0.2891 | 0.2310 | - | - | - | - | - | - | - |
| | all-roberta-large-v1 | - | - | 0.2236 | 0.1671 | - | - | - | - | - | - | - |
| miracl-de-dev | all-MiniLM-L6-v2 | - | - | - | 0.4260 | 0.4318 | - | - | - | - | - | - |
| | all-mpnet-base-v2 | - | - | - | 0.3165 | 0.3227 | - | - | - | - | - | - |
| | all-roberta-large-v1 | - | - | - | 0.2765 | 0.2806 | - | - | - | - | - | - |

| Dataset | Encoder | 1 | 2 | 4 | 8 | 16 | 32 | 64 | 128 | 256 | 512 | 1024 |
|---|---|---|---|---|---|---|---|---|---|---|---|---|
| miracl-en-dev | all-MiniLM-L6-v2 | - | - | - | 0.4643 | 0.4578 | - | - | - | - | - | - |
|  | all-mpnet-base-v2 | - | - | - | 0.3353 | 0.3292 | - | - | - | - | - | - |
|  | all-roberta-large-v1 | - | - | - | 0.2923 | 0.2829 | - | - | - | - | - | - |
| miracl-es-dev | all-MiniLM-L6-v2 | - | 0.3962 | 0.4177 | 0.4179 | - | - | - | - | - | - | - |
|  | all-mpnet-base-v2 | - | 0.3225 | 0.3320 | 0.3313 | - | - | - | - | - | - | - |
|  | all-roberta-large-v1 | - | 0.2716 | 0.2775 | 0.2733 | - | - | - | - | - | - | - |
| miracl-fa-dev | all-MiniLM-L6-v2 | - | - | - | 0.2716 | 0.2718 | - | - | - | - | - | - |
|  | all-mpnet-base-v2 | - | - | - | 0.2821 | 0.2980 | - | - | - | - | - | - |
|  | all-roberta-large-v1 | - | - | - | 0.1766 | 0.1816 | - | - | - | - | - | - |
| miracl-fi-dev | all-MiniLM-L6-v2 | 0.3888 | 0.3838 | 0.4223 | 0.3807 | 0.3792 | - | - | - | - | - | - |
|  | all-mpnet-base-v2 | 0.3088 | 0.3068 | 0.3287 | 0.3061 | 0.3052 | - | - | - | - | - | - |
|  | all-roberta-large-v1 | 0.2506 | 0.2590 | 0.2650 | 0.2552 | 0.2582 | - | - | - | - | - | - |
| miracl-fr-dev | all-MiniLM-L6-v2 | - | - | - | 0.4322 | - | - | - | - | - | - | - |
|  | all-mpnet-base-v2 | - | - | - | 0.3275 | - | - | - | - | - | - | - |
|  | all-roberta-large-v1 | - | - | - | 0.2866 | - | - | - | - | - | - | - |
| miracl-hi-dev | all-MiniLM-L6-v2 | - | - | 0.2608 | 0.2704 | - | - | - | - | - | - | - |
|  | all-mpnet-base-v2 | - | - | 0.2031 | 0.2416 | - | - | - | - | - | - | - |
|  | all-roberta-large-v1 | - | - | 0.1698 | 0.1854 | - | - | - | - | - | - | - |
| miracl-id-dev | all-MiniLM-L6-v2 | - | 0.3837 | 0.3646 | 0.3805 | 0.3903 | - | - | - | - | - | - |
|  | all-mpnet-base-v2 | - | 0.3082 | 0.2988 | 0.3055 | 0.3055 | - | - | - | - | - | - |
|  | all-roberta-large-v1 | - | 0.2820 | 0.2823 | 0.2784 | 0.2659 | - | - | - | - | - | - |
| miracl-ja-dev | all-MiniLM-L6-v2 | 0.4089 | 0.4082 | - | 0.3946 | 0.4110 | - | - | - | - | - | - |
|  | all-mpnet-base-v2 | 0.2983 | 0.2982 | - | 0.2924 | 0.3044 | - | - | - | - | - | - |
|  | all-roberta-large-v1 | 0.2411 | 0.2439 | - | 0.2348 | 0.2435 | - | - | - | - | - | - |
| miracl-ko-dev | all-MiniLM-L6-v2 | - | - | - | 0.3445 | 0.3440 | - | - | - | - | - | - |
|  | all-mpnet-base-v2 | - | - | - | 0.3155 | 0.3133 | - | - | - | - | - | - |
|  | all-roberta-large-v1 | - | - | - | 0.2572 | 0.2575 | - | - | - | - | - | - |
| miracl-ru-dev | all-MiniLM-L6-v2 | - | - | - | 0.2789 | 0.2867 | - | - | - | - | - | - |
|  | all-mpnet-base-v2 | - | - | - | 0.2422 | 0.2485 | - | - | - | - | - | - |
|  | all-roberta-large-v1 | - | - | - | 0.2132 | 0.2149 | - | - | - | - | - | - |
| miracl-sw-dev | all-MiniLM-L6-v2 | 0.3755 | 0.3462 | 0.3521 | 0.3463 | 0.3512 | - | - | - | - | - | - |
|  | all-mpnet-base-v2 | 0.2949 | 0.2845 | 0.2858 | 0.2828 | 0.2795 | - | - | - | - | - | - |
|  | all-roberta-large-v1 | 0.2535 | 0.2454 | 0.2403 | 0.2437 | 0.2529 | - | - | - | - | - | - |
| miracl-te-dev | all-MiniLM-L6-v2 | 0.4637 | - | 0.3850 | 0.4565 | - | - | - | - | - | - | - |
|  | all-mpnet-base-v2 | 0.4615 | - | 0.3167 | 0.4587 | - | - | - | - | - | - | - |
|  | all-roberta-large-v1 | 0.2145 | - | 0.2247 | 0.2109 | - | - | - | - | - | - | - |
| miracl-th-dev | all-MiniLM-L6-v2 | 0.4339 | 0.4090 | 0.3842 | 0.4126 | - | - | - | - | - | - | - |
|  | all-mpnet-base-v2 | 0.3129 | 0.3198 | 0.3048 | 0.3166 | - | - | - | - | - | - | - |
|  | all-roberta-large-v1 | 0.2158 | 0.1951 | 0.1929 | 0.2166 | - | - | - | - | - | - | - |
| miracl-yo-dev | all-MiniLM-L6-v2 | - | - | - | 0.3630 | - | - | - | - | - | - | - |
|  | all-mpnet-base-v2 | - | - | - | 0.2889 | - | - | - | - | - | - | - |
|  | all-roberta-large-v1 | - | - | - | 0.2501 | - | - | - | - | - | - | - |
| miracl-zh-dev | all-MiniLM-L6-v2 | - | - | - | 0.3884 | - | - | - | - | - | - | - |
|  | all-mpnet-base-v2 | - | - | - | 0.2885 | - | - | - | - | - | - | - |
|  | all-roberta-large-v1 | - | - | - | 0.2517 | - | - | - | - | - | - | - |
| mmarco-de-dev | all-MiniLM-L6-v2 | 0.4392 | 0.4359 | 0.4358 | - | - | - | - | - | - | - | - |
|  | all-mpnet-base-v2 | 0.3255 | 0.3225 | 0.3229 | - | - | - | - | - | - | - | - |
|  | all-roberta-large-v1 | 0.2884 | 0.2837 | 0.2838 | - | - | - | - | - | - | - | - |
| mmarco-es-dev | all-MiniLM-L6-v2 | 0.4336 | 0.4229 | 0.4229 | - | - | - | - | - | - | - | - |
|  | all-mpnet-base-v2 | 0.3367 | 0.3306 | 0.3295 | - | - | - | - | - | - | - | - |
|  | all-roberta-large-v1 | 0.2854 | 0.2828 | 0.2809 | - | - | - | - | - | - | - | - |
| mmarco-fr-dev | all-MiniLM-L6-v2 | 0.4362 | 0.4371 | 0.4362 | - | - | - | - | - | - | - | - |
|  | all-mpnet-base-v2 | 0.3375 | 0.3376 | 0.3365 | - | - | - | - | - | - | - | - |
|  | all-roberta-large-v1 | 0.2877 | 0.2878 | 0.2880 | - | - | - | - | - | - | - | - |
| mmarco-id-dev | all-MiniLM-L6-v2 | 0.4029 | 0.3984 | 0.3919 | - | - | - | - | - | - | - | - |
|  | all-mpnet-base-v2 | 0.2997 | 0.2965 | 0.2967 | - | - | - | - | - | - | - | - |
|  | all-roberta-large-v1 | 0.2713 | 0.2707 | 0.2705 | - | - | - | - | - | - | - | - |
| mmarco-it-dev | all-MiniLM-L6-v2 | 0.4046 | 0.4050 | 0.4013 | - | - | - | - | - | - | - | - |
|  | all-mpnet-base-v2 | 0.3235 | 0.3235 | 0.3209 | - | - | - | - | - | - | - | - |
|  | all-roberta-large-v1 | 0.2806 | 0.2805 | 0.2789 | - | - | - | - | - | - | - | - |
| mmarco-pt-dev | all-MiniLM-L6-v2 | 0.4176 | 0.4067 | 0.4103 | - | - | - | - | - | - | - | - |
|  | all-mpnet-base-v2 | 0.3185 | 0.3118 | 0.3150 | - | - | - | - | - | - | - | - |
|  | all-roberta-large-v1 | 0.2870 | 0.2816 | 0.2847 | - | - | - | - | - | - | - | - |
| mmarco-ru-dev | all-MiniLM-L6-v2 | 0.2941 | 0.2920 | 0.2887 | - | - | - | - | - | - | - | - |
|  | all-mpnet-base-v2 | 0.2570 | 0.2505 | 0.2473 | - | - | - | - | - | - | - | - |
|  | all-roberta-large-v1 | 0.2310 | 0.2186 | 0.2167 | - | - | - | - | - | - | - | - |
| mr-tydi-ar-test | all-MiniLM-L6-v2 | 0.2561 | 0.2720 | - | - | - | - | - | - | - | - | - |
|  | all-mpnet-base-v2 | 0.3240 | 0.3228 | - | - | - | - | - | - | - | - | - |
|  | all-roberta-large-v1 | 0.2318 | 0.2313 | - | - | - | - | - | - | - | - | - |
| mr-tydi-bn-test | all-MiniLM-L6-v2 | 0.1700 | 0.1834 | - | - | - | - | - | - | - | - | - |
|  | all-mpnet-base-v2 | 0.1585 | 0.1553 | - | - | - | - | - | - | - | - | - |
|  | all-roberta-large-v1 | 0.1455 | 0.1410 | - | - | - | - | - | - | - | - | - |
| mr-tydi-en-test | all-MiniLM-L6-v2 | 0.4553 | 0.4532 | - | - | - | - | - | - | - | - | - |
|  | all-mpnet-base-v2 | 0.3313 | 0.3300 | - | - | - | - | - | - | - | - | - |
|  | all-roberta-large-v1 | 0.2890 | 0.2882 | - | - | - | - | - | - | - | - | - |
| mr-tydi-fi-test | all-MiniLM-L6-v2 | 0.3754 | 0.3699 | - | - | - | - | - | - | - | - | - |
|  | all-mpnet-base-v2 | 0.3035 | 0.3000 | - | - | - | - | - | - | - | - | - |
|  | all-roberta-large-v1 | 0.2519 | 0.2493 | - | - | - | - | - | - | - | - | - |
| mr-tydi-id-test | all-MiniLM-L6-v2 | 0.3870 | 0.3842 | - | - | - | - | - | - | - | - | - |
|  | all-mpnet-base-v2 | 0.3099 | 0.3100 | - | - | - | - | - | - | - | - | - |
|  | all-roberta-large-v1 | 0.2724 | 0.2742 | - | - | - | - | - | - | - | - | - |
| mr-tydi-ja-test | all-MiniLM-L6-v2 | 0.4039 | 0.4055 | - | - | - | - | - | - | - | - | - |
|  | all-mpnet-base-v2 | 0.2953 | 0.2943 | - | - | - | - | - | - | - | - | - |
|  | all-roberta-large-v1 | 0.2418 | 0.2376 | - | - | - | - | - | - | - | - | - |
| mr-tydi-ko-test | all-MiniLM-L6-v2 | 0.2892 | 0.3090 | - | - | - | - | - | - | - | - | - |
|  | all-mpnet-base-v2 | 0.2974 | 0.2997 | - | - | - | - | - | - | - | - | - |
|  | all-roberta-large-v1 | 0.2328 | 0.2368 | - | - | - | - | - | - | - | - | - |
| mr-tydi-ru-test | all-MiniLM-L6-v2 | 0.2827 | 0.2972 | - | - | - | - | - | - | - | - | - |
|  | all-mpnet-base-v2 | 0.2527 | 0.2506 | - | - | - | - | - | - | - | - | - |
|  | all-roberta-large-v1 | 0.2244 | 0.2218 | - | - | - | - | - | - | - | - | - |

| Dataset | Encoder | 1 | 2 | 4 | 8 | 16 | 32 | 64 | 128 | 256 | 512 | 1024 |
|---|---|---|---|---|---|---|---|---|---|---|---|---|
| | all-MiniLM-L6-v2 | 0.3607 | 0.3513 | - | - | - | - | - | - | - | - | - |
| mr-tydi-sw-test | all-mpnet-base-v2 | 0.3019 | 0.2905 | - | - | - | - | - | - | - | - | - |
| | all-roberta-large-v1 | 0.2611 | 0.2544 | - | - | - | - | - | - | - | - | - |
| | all-MiniLM-L6-v2 | 0.4393 | 0.4661 | - | - | - | - | - | - | - | - | - |
| mr-tydi-te-test | all-mpnet-base-v2 | 0.4081 | 0.4720 | - | - | - | - | - | - | - | - | - |
| | all-roberta-large-v1 | 0.2033 | 0.2103 | - | - | - | - | - | - | - | - | - |
| | all-MiniLM-L6-v2 | 0.4300 | 0.4114 | - | - | - | - | - | - | - | - | - |
| mr-tydi-th-test | all-mpnet-base-v2 | 0.3506 | 0.3215 | - | - | - | - | - | - | - | - | - |
| | all-roberta-large-v1 | 0.2259 | 0.2282 | - | - | - | - | - | - | - | - | - |
| | all-MiniLM-L6-v2 | - | - | - | - | - | - | - | 0.4486 | 0.4531 | 0.4549 | 0.4472 |
| msmarco-document-trec-dl-2019 | all-mpnet-base-v2 | - | - | - | - | - | - | - | 0.3143 | 0.3180 | 0.3184 | 0.3128 |
| | all-roberta-large-v1 | - | - | - | - | - | - | - | 0.2715 | 0.2749 | 0.2771 | 0.2700 |
| | all-MiniLM-L6-v2 | - | - | - | - | - | - | - | 0.4516 | 0.4462 | - | - |
| msmarco-document-trec-dl-2020 | all-mpnet-base-v2 | - | - | - | - | - | - | - | 0.3174 | 0.3127 | - | - |
| | all-roberta-large-v1 | - | - | - | - | - | - | - | 0.2746 | 0.2724 | - | - |
| | all-MiniLM-L6-v2 | - | 0.4492 | 0.4411 | - | - | - | - | 0.4539 | 0.4479 | - | - |
| msmarco-document-trec-dl-hard-fold1 | all-mpnet-base-v2 | - | 0.3112 | 0.3166 | - | - | - | - | 0.3174 | 0.3183 | - | - |
| | all-roberta-large-v1 | - | 0.2979 | 0.2781 | - | - | - | - | 0.2795 | 0.2748 | - | - |
| | all-MiniLM-L6-v2 | - | - | 0.4525 | - | - | - | - | 0.4525 | 0.4512 | - | - |
| msmarco-document-trec-dl-hard-fold2 | all-mpnet-base-v2 | - | - | 0.3198 | - | - | - | - | 0.3157 | 0.3135 | - | - |
| | all-roberta-large-v1 | - | - | 0.2687 | - | - | - | - | 0.2682 | 0.2679 | - | - |
| | all-MiniLM-L6-v2 | - | 0.4345 | 0.4435 | 0.4541 | - | - | - | 0.4626 | - | - | - |
| msmarco-document-trec-dl-hard-fold3 | all-mpnet-base-v2 | - | 0.3098 | 0.3126 | 0.3129 | - | - | - | 0.3217 | - | - | - |
| | all-roberta-large-v1 | - | 0.2667 | 0.2703 | 0.2695 | - | - | - | 0.2720 | - | - | - |
| | all-MiniLM-L6-v2 | - | 0.4547 | 0.4278 | - | - | - | - | 0.4480 | 0.4590 | - | - |
| msmarco-document-trec-dl-hard-fold4 | all-mpnet-base-v2 | - | 0.3143 | 0.3026 | - | - | - | - | 0.3186 | 0.3250 | - | - |
| | all-roberta-large-v1 | - | 0.2680 | 0.2637 | - | - | - | - | 0.2708 | 0.2721 | - | - |
| | all-MiniLM-L6-v2 | - | - | 0.4418 | - | - | - | - | 0.4420 | 0.4414 | 0.4423 | 0.4327 |
| msmarco-document-trec-dl-hard-fold5 | all-mpnet-base-v2 | - | - | 0.3132 | - | - | - | - | 0.3106 | 0.3148 | 0.3089 | 0.3057 |
| | all-roberta-large-v1 | - | - | 0.2714 | - | - | - | - | 0.2705 | 0.2763 | 0.2741 | 0.2656 |
| | all-MiniLM-L6-v2 | - | - | - | - | - | - | 0.4432 | 0.4508 | 0.4563 | 0.4580 | 0.4476 |
| msmarco-document-v2-trec-dl-2019 | all-mpnet-base-v2 | - | - | - | - | - | - | 0.3144 | 0.3150 | 0.3183 | 0.3188 | 0.3160 |
| | all-roberta-large-v1 | - | - | - | - | - | - | 0.2670 | 0.2688 | 0.2732 | 0.2750 | 0.2657 |
| | all-MiniLM-L6-v2 | - | - | - | - | - | - | 0.4642 | 0.4517 | 0.4559 | - | - |
| msmarco-document-v2-trec-dl-2020 | all-mpnet-base-v2 | - | - | - | - | - | - | 0.3269 | 0.3167 | 0.3161 | - | - |
| | all-roberta-large-v1 | - | - | - | - | - | - | 0.2801 | 0.2725 | 0.2727 | - | - |
| | all-MiniLM-L6-v2 | - | - | - | - | - | - | 0.4628 | 0.4557 | 0.4500 | 0.4636 | - |
| msmarco-document-v2-trec-dl-2021 | all-mpnet-base-v2 | - | - | - | - | - | - | 0.3243 | 0.3208 | 0.3185 | 0.3249 | - |
| | all-roberta-large-v1 | - | - | - | - | - | - | 0.2778 | 0.2741 | 0.2722 | 0.2807 | - |
| | all-MiniLM-L6-v2 | - | - | - | - | - | - | - | 0.4628 | 0.4633 | - | - |
| msmarco-passage-trec-dl-2020 | all-mpnet-base-v2 | - | - | - | - | - | - | - | 0.3219 | 0.3209 | - | - |
| | all-roberta-large-v1 | - | - | - | - | - | - | - | 0.2811 | 0.2794 | - | - |
| | all-MiniLM-L6-v2 | - | - | 0.4720 | 0.4518 | - | - | - | 0.4575 | 0.4523 | - | - |
| msmarco-passage-trec-dl-hard | all-mpnet-base-v2 | - | - | 0.3286 | 0.3180 | - | - | - | 0.3207 | 0.3153 | - | - |
| | all-roberta-large-v1 | - | - | 0.2885 | 0.2792 | - | - | - | 0.2803 | 0.2746 | - | - |
| | all-MiniLM-L6-v2 | 0.4471 | 0.4487 | 0.4517 | 0.4523 | 0.4531 | - | - | - | - | - | - |
| msmarco-qna-dev | all-mpnet-base-v2 | 0.3174 | 0.3189 | 0.3197 | 0.3196 | 0.3194 | - | - | - | - | - | - |
| | all-roberta-large-v1 | 0.2741 | 0.2768 | 0.2784 | 0.2783 | 0.2784 | - | - | - | - | - | - |
| | all-MiniLM-L6-v2 | 0.4309 | 0.4323 | 0.4347 | - | - | - | - | - | - | - | - |
| natural-questions-dev | all-mpnet-base-v2 | 0.3066 | 0.3070 | 0.3082 | - | - | - | - | - | - | - | - |
| | all-roberta-large-v1 | 0.2699 | 0.2704 | 0.2714 | - | - | - | - | - | - | - | - |
| | all-MiniLM-L6-v2 | - | - | - | - | - | - | - | - | - | 0.4435 | 0.4490 |
| pmc-v1-trec-cds-2014 | all-mpnet-base-v2 | - | - | - | - | - | - | - | - | - | 0.3281 | 0.3355 |
| | all-roberta-large-v1 | - | - | - | - | - | - | - | - | - | 0.2914 | 0.2986 |
| | all-MiniLM-L6-v2 | - | - | - | - | - | - | - | - | - | 0.4663 | 0.4625 |
| pmc-v1-trec-cds-2015 | all-mpnet-base-v2 | - | - | - | - | - | - | - | - | - | 0.3438 | 0.3508 |
| | all-roberta-large-v1 | - | - | - | - | - | - | - | - | - | 0.3070 | 0.3167 |
| | all-MiniLM-L6-v2 | - | - | - | - | - | - | - | - | - | 0.4520 | 0.4524 |
| pmc-v2-trec-cds-2016 | all-mpnet-base-v2 | - | - | - | - | - | - | - | - | - | 0.3548 | 0.3514 |
| | all-roberta-large-v1 | - | - | - | - | - | - | - | - | - | 0.3114 | 0.3124 |
| | all-MiniLM-L6-v2 | 0.4268 | - | - | - | - | - | - | - | - | - | - |
| trec-tot-2023-dev | all-mpnet-base-v2 | 0.3309 | - | - | - | - | - | - | - | - | - | - |
| | all-roberta-large-v1 | 0.2929 | - | - | - | - | - | - | - | - | - | - |
| | all-MiniLM-L6-v2 | 0.4386 | 0.4316 | 0.4403 | 0.4375 | 0.4414 | 0.4363 | 0.4392 | - | - | - | - |
| vaswani | all-mpnet-base-v2 | 0.3088 | 0.3083 | 0.3096 | 0.3081 | 0.3103 | 0.3082 | 0.3087 | - | - | - | - |
| | all-roberta-large-v1 | 0.2680 | 0.2700 | 0.2668 | 0.2692 | 0.2707 | 0.2672 | 0.2735 | - | - | - | - |

