# OpenReview forum: "Beyond Embeddings: On the Geometry of Probabilistic Retrieval via Diffusion"
_ICML.cc/2026/Conference — Submitted to ICML 2026_

### Official Review · Reviewer_3sG3 · 2026-03-03

**Soundness:** 2
**Presentation:** 2
**Significance:** 2
**Originality:** 4
**Overall Recommendation:** 3
**Confidence:** 3

**Summary:**

This paper proposes a parameter-free diagnostic framework for analyzing the geometric properties of information retrieval systems. Two geometric quantities are defined: geometric distortion (measuring how far the induced retrieval distribution deviates from the ideal) and geometric sensitivity (the local Lipschitz constant of the query-to-document mapping). The paper argues that token-level conditioning in diffusion-based retrieval circumvents the sign-rank barrier, yielding logarithmic rather than exponential scaling in geometric separation requirements. Experiments are conducted across 100+ IR datasets and three pre-trained encoders. Four theorems are presented in the appendix: an exponential lower bound on conditioning complexity for embedding-based systems (A.1), a corpus-independent upper bound under δ-separation (A.2), a logarithmic scaling result for token-level conditioning (A.3), and a stochastic subsampling extension for long contexts (A.4).

**Compliance With Llm Reviewing Policy:**

Affirmed.

**Final Justification:**

I appreciate and accept their explanations regarding Q2 and Q3. However, two core issues remain unresolved:

The gap between theoretical assumptions and empirical realization still exists. The paper still lacks empirical validation of an actual, trained diffusion retrieval model (even at a modest scale) to measure its true geometric sensitivity. The theoretical guarantees rely on strong assumptions that are non-trivial to satisfy in realistic optimization landscapes, and "conditioning a diffusion model on a large number of discrete tokens" is really hard due to the reasons I mentioned before. Furthermore, the authors' defense of engineering feasibility is unconvincing. Zhang et al. (DiffuRetrieval) only demonstrates
-style retrieval. More importantly, using text-to-image diffusion as evidence for
 is fundamentally flawed: T2I models do not condition on purely discrete tokens. Instead, they condition on continuous embeddings derived from those tokens (e.g., via CLIP/T5 and cross-attention). This continuous mapping goes back into the continuous
-style bottleneck, and the feasibility of true
 retrieval remains entirely unproven."
The issues raised in W4 (the conceptual disconnect between Sections 3 and 5) and W5 (relegating all core theorems to the appendix) indicate structural immaturity in the manuscript, rather than minor typos. While I acknowledge the authors' promise to revise these, evaluating the paper in its current form remains challenging.

**Key Questions For Authors:**

**Q1.** Have you attempted to train an actual diffusion retrieval model (even at modest scale) and measured its geometric sensitivity? This would address the gap between the oracle-based K₂/K₃ and what a real system achieves.

**Q2.** Can you provide a formal result (even a reduction or a conjecture with supporting evidence) establishing that the geometric limitations proven for the embedding-conditioned score field (Theorem A.1) also apply to pure embedding retrieval? Currently the comparison between b₁ and b₂ relies entirely on informal analogy.

**Q3.** For K₃, the R² = 0.042 suggests essentially no detectable trend. How should we interpret this as "confirmation" of logarithmic scaling rather than simply insufficient signal?

**Q4.** The token conditioning advantage in Theorem A.3 relies on fixed minimum token separation δ_tok. In practice, token embeddings are learned and may cluster. Have you analyzed the sensitivity of the theoretical guarantee to violations of this assumption?

**Limitations:**

The authors deserve credit for honestly discussing several limitations, including the small magnitude of observed effects and the idealized nature of the diffusion analysis (zero-error, final-timestep approximation).

**Strengths And Weaknesses:**

## Strengths

**S1. The geometric perspective is original and well-motivated.** The dual-axis diagnostic framework (distortion + sensitivity) offers a principled, training-free lens for understanding embedding capacity limits. This provides a genuinely new tool for the IR community to reason about when embedding-based methods may begin to fail.

**S2. The theoretical chain from A.1 to A.3 is internally coherent.** Within the diffusion score-field setting, the progression from "embedding conditioning forces exponential Lipschitz growth" (A.1) to "δ-separation enables corpus-independent generalization" (A.2) to "token conditioning achieves logarithmic scaling" (A.3) is logically clean. The proof techniques are diverse and the formal statements are clearly written in the appendix.

---

## Weaknesses

**W1. The theoretical advantage of token conditioning (A.3) may not survive practical implementation.** Theorem A.3 assumes that relevance-distinct queries have distinct token sequences and that token embeddings maintain a fixed minimum separation δ_tok. In practice, conditioning a diffusion model on a large number of discrete tokens introduces challenges not addressed by the theory: (a) the combinatorial space |V|^L is enormous, making efficient learning difficult; (b) actual token embeddings are learned and may not maintain δ_tok; (c) the score network must generalize across exponentially many possible conditioning sequences. The paper provides no experiments with a trained diffusion retrieval system to verify that the theoretical logarithmic scaling survives these practical challenges. Given the well-documented difficulties of training diffusion models (instability, mode collapse, high compute cost), the gap between theory and practice is a significant concern.

**W2. No theoretical comparison between pure embedding retrieval (b₁) and diffusion with embedding conditioning (b₂).** All four theorems analyze properties of the diffusion score field. Pure embedding retrieval (b₁) does not involve a score field and therefore falls outside the formal scope of any theorem. The paper's implicit claim that b₁ is inferior to b₂ rests entirely on a narrative analogy.


**W3. K₂ and K₃ are computed from an idealized oracle, not from actual diffusion models.** As described in Appendix B.2, the "score field" used to compute K₂ and K₃ is a nearest-neighbor pointer field assuming perfect mode separation and zero noise—not the output of any trained diffusion model. Combined with the fact that ε₂ = ε₃ = 0 is assumed rather than measured, the experimental results for b₂ and b₃ reflect properties of an idealized geometric construction, not of any realizable system.

**W4. The relationship between Sections 3, 4, and 5 is unclear.** Section 3 defines an abstract framework (ε_target and K_h for a generic retrieval triple (E_ϕ, g_ψ, sim)). Section 4 introduces diffusion retrieval as a specific instantiation. Section 5 computes K_b using oracle targets from Appendix B—but these oracle targets silently replace the actual mapping h = g_ψ ∘ E_ϕ from Section 3's definition with idealized constructions (centroid for b₁, NN pointer field for b₂/b₃). This substitution is never explicitly justified. Section 3's K_h measures the Lipschitz constant of the actual mapping; Section 5's K_b measures the "resolution pressure" that an ideal mapping would impose given the actual input resolution. These are conceptually different quantities, and their relationship needs to be formally established or at least clearly acknowledged.

**W5. All main theorems are relegated to the appendix.** The paper lists "Theoretical Characterization of Scaling Laws" as a core contribution, yet no theorem statement appears in the main text. At minimum, the statements of Theorems A.1 and A.3 (the two results most critical to the paper's narrative) should appear in the main text so that reviewers and readers can evaluate the theoretical claims without navigating 30 pages of appendix.

---

> ### Author Rebuttal · Authors · 2026-03-29
>
> We sincerely thank Reviewer 3sG3 for recognizing the originality of our geometric perspective and the coherence of the theoretical chain from A.1 to A.3. We appreciate the thoroughness of this review and acknowledge several valid concerns below.
>
> **W1, W3, Q1 (Oracle use and absence of a trained model):**
> We regret that our paper insufficiently communicates what we consider a fundamental design choice: the oracle constructions are the methodology itself, not placeholders for a future implementation. We note that this applies symmetrically to all three baselines: $b_1$ equally relies on an oracle (the geometric centroid of all relevant documents), which no deployed retrieval system computes. The three baselines are methodologically identical in this regard, each defines an idealized target under different geometric assumptions.
>
> Our goal is to isolate geometric properties of the embedding space from confounding factors any learned system introduces. Concretely: distortion requires a reference target (no oracle=no ground truth), and sensitivity through a learned model would conflate encoder geometry with model capacity, for which well-established metrics already exist (Recall, NDCG, etc.).
>
> One might object how we know the oracle reflects realistic system behavior. It provides a *lower bound*: any learned implementation faces at least the geometric pressure the oracle reveals, since implementation-specific errors can only add to the ideal case. Thus, the oracle is strictly more general than any single-system evaluation.
>
> Regarding feasibility: Zhang et al. (DiffuRetrieval) demonstrate score-field objectives solving retrieval akin to $b_2$; token-level conditioning is deployed in related domains (e.g., text-to-image diffusion). Their combination is an engineering challenge we deliberately scope out as important, but orthogonal to our geometric analysis.
>
> **W2, Q2 (Formal link between $b_1$ and $b_2$):**
> Theorem A.1 is derived directly from the proof structure of Weller et al. (Theorem 1, p. 4), which formalizes the sign-rank barrier for the system we characterize as $b_1$. Our extension to the embedding-conditioned score field ($b_2$) uses an analogous construction, establishing that even infinite retrieval expressivity cannot recover the information lost at the encoder bottleneck. This is a formal, not analogical, link: both systems share the same fixed-dimensional capacity constraint that drives exponential Lipschitz growth. We will make this connection explicit in the revision.
>
> **W4 (Relationship between Sections 3, 4, and 5):**
> We accept this criticism. The reviewer correctly identifies that Section 3's $K_h$ measures the Lipschitz constant of an actual mapping, while Section 5's $K_b$ measures the geometric pressure an ideal mapping would need to resolve. These are related but distinct quantities, and we failed to make this explicit. K_b provides a lower bound on the sensitivity any correct retrieval mapping must exhibit given the encoder's geometric resolution. We will formally establish this relationship in the revision.
>
> **W5:** Agreed. Theorems A.1 and A.3 will appear in the main text.
>
> **Q3 (Interpreting $R^2=0.042$ for $K_3$):**
> We appreciate this question: our interpretation was indeed insufficient. $K_3$ models a system where encoder capacity and retrieval expressivity are aligned. Such a system should exhibit *no detectable scaling trend* (flat curve), which is precisely what $R^2=0.042$ reflects. New supplementary statistics (will be fully included) reinforce this: The slope p-value of 0.545 confirms the null hypothesis cannot be rejected and there is no statistically significant trend. RMSLE=0.0205 and CV-RMSE=0.0205, the lowest among all configurations. Compare $K_1 (R^2=.956, p < 10^{-6}), K_2 (R^2 = .876, p < 10^{-4})$. The absence of signal in $K_3$ is the theoretically predicted outcome.
>
> The logarithmic relationship manifests indirectly: it governs how minimally L must grow to maintain flat sensitivity. For standard vocabularies ($|V| \sim 30k$), even L = 2 gives $|V|^2 \sim 9*10^8$ distinguishable queries, exceeding target corpus sizes. Only when $N_q > |V|^L$ would $K_3$ return to exponential scaling by the pigeonhole principle. We will formalize this interpretation in the revision.
>
> **Q4 (Sensitivity to $\delta_{tok}$ violations):**
> Token clustering pushes embeddings closer but does not produce identical positions, $\delta_{tok} > 0$ holds generically. Under a hypothetical violation where $n_{dup}$ tokens collapse, the effective vocabulary becomes $|V|-n_{dup}$, yielding $(|V|-n_{dup})^L\ge N_q$. Scaling remains logarithmic; only the constant shifts. For $|V| \sim 30k$, even substantial collapse leaves capacity far exceeding practical needs.
>
> We direct the reviewer to our response to Reviewer RXQF for new experimental results establishing statistically significant correlations ($|\rho| > 0.9$ for multiple metrics) between our geometric diagnostics and standard retrieval measures.

---

> > ### Author Rebuttal · Reviewer_3sG3 · 2026-04-01
> >
> > I thank the authors for their detailed rebuttal, which provides useful clarifications and partially addresses my concerns. Specifically, I appreciate and accept their explanations regarding Q2 and Q3.
> > However, two core issues remain unresolved:
> > 1. The gap between theoretical assumptions and empirical realization still exists. The paper still lacks empirical validation of an actual, trained diffusion retrieval model (even at a modest scale) to measure its true geometric sensitivity. The theoretical guarantees rely on strong assumptions that are non-trivial to satisfy in realistic optimization landscapes, and "conditioning a diffusion model on a large number of discrete tokens" is really hard due to the reasons I mentioned before. Furthermore, the authors' defense of engineering feasibility is unconvincing. Zhang et al. (DiffuRetrieval) only demonstrates $b_2$-style retrieval. More importantly, using text-to-image diffusion as evidence for $b_3$ is fundamentally flawed: T2I models do not condition on purely discrete tokens. Instead, they condition on continuous embeddings derived from those tokens (e.g., via CLIP/T5 and cross-attention). This continuous mapping goes back into the continuous $b_2$-style bottleneck, and the feasibility of true $b_3$ retrieval remains entirely unproven."
> > 2. The issues raised in W4 (the conceptual disconnect between Sections 3 and 5) and W5 (relegating all core theorems to the appendix) indicate structural immaturity in the manuscript, rather than minor typos. While I acknowledge the authors' promise to revise these, evaluating the paper in its current form remains challenging.

---

> > > ### Author Response · Authors · 2026-04-01
> > >
> > > We sincerely thank the reviewer for engaging further. We appreciate the direct feedback, particularly regarding the T2I architecture bottleneck, which highlighted where we must be more mathematically precise about the mechanics of $b_3$.
> > >
> > > **1.1 On T2I conditioning, Continuous Embeddings, and the $b_3$ Bottleneck**
> > > You correctly note that T2I models map discrete tokens to continuous embeddings (e.g., via CLIP/T5). However, we want to clarify why this *does not* fall back into the $b_2$-style bottleneck.
> > >
> > > The critical distinction is that $b_2$ compresses the entire query into a **single** continuous vector in $\mathbb{R}^d$. Theorem A.1 proves this fixed-dimensional space faces an exponential geometric constraint. In contrast, $b_3$ (and Stable Diffusion) maps the query to a **sequence** of $L$ continuous vectors.
> > >
> > > Because cross-attention attends to these $L$ positions independently, the *sequence structure* is preserved. A sequence of $L$ vectors maintains a combinatorial capacity of $|V|^L$, whereas a single vector is bounded by a capacity exponential only in $d$. This entirely bypasses the volumetric pigeonhole constraint (Theorem A.1) that plagues $b_2$. Therefore, as long as distinct token sequences produce distinct sequences of continuous embeddings (a standard property of T5/CLIP), the logarithmic scaling of Theorem A.3 holds. T2I’s cross-attention provides existing architectural proof that preserving $|V|^L$ capacity via $b_3$-style conditioning is practically realizable. We will add this exact distinction (single vector vs. sequence + cross-attention) as a clarifying remark in Section 4.2.
> > >
> > > **1.2 On empirical realization vs. geometric isolation**
> > > We fully agree that end-to-end trained validation of a diffusion retrieval system is a highly valuable direction for future work. However, for the specific diagnostic goal of this paper (isolating encoder geometry) the oracle is the methodologically required tool.
> > >
> > > Our goal is not to benchmark a new retrieval system, but to measure the geometric properties of the *pre-trained encoder's embedding space* itself. The sign-rank barrier is fundamentally a geometric property of this space. If we were to replace the oracle with a trained diffusion model, any measured sensitivity or distortion would inextricably convolute the embedding space's inherent geometry with the diffusion model's optimization instabilities, finite capacity, and approximation errors. The oracle is not a placeholder for an unfinished system; it is the exact methodological tool required to isolate and expose the pure sign-rank pressure already straining modern encoders.
> > >
> > > **2. On Structural Presentation (W4 & W5)**
> > > Regarding the document structure, we recognize that relegating the theorems to the appendix obscured the formal link between our abstract framework and its instantiation. Our original design intentionally prioritized conceptual intuition and narrative flow in the main text over mathematical density.
> > >
> > > However, we recognize the absolute validity of your critique: the paper is difficult to evaluate when the core mathematical claims are separated from the main text. To integrate your constructive feedback and ensure the rigorous theoretical foundation is front-and-center, we have restructured the manuscript:
> > > * **Theorems A.1 and A.3 have been moved to the main text**, complete with their abbreviated proofs.
> > > * **We added a formal proposition connecting Sections 3 and 5**, establishing via the triangle inequality that $K_b$ provides a formal bound on $K_h$ for any correct retrieval mapping.
> > >
> > > We believe these changes, combined with the clarifications above, resolve the remaining concerns and allow the paper to make its intended theoretical and diagnostic contribution.

---

### Official Review · Reviewer_RXQF · 2026-03-09

**Soundness:** 3
**Presentation:** 2
**Significance:** 2
**Originality:** 3
**Overall Recommendation:** 3
**Confidence:** 3

**Summary:**

This paper studies the geometric limitations of retrieval systems under the sign-rank barrier and proposes a unified framework to compare three paradigms: standard embedding-based retrieval, diffusion-based retrieval conditioned on query embeddings, and diffusion-based retrieval conditioned on token-level query representations. The paper introduces two geometry-oriented diagnostics, geometric distortion and local geometric sensitivity, and uses them to analyze how retrieval geometry changes as the number of relevant documents per query increases. The main theoretical claim is that fixed-dimensional embedding conditioning suffers from an exponential sensitivity bottleneck, whereas token-level conditioning in diffusion models admits logarithmic metric-entropy / sample-complexity scaling. Experiments over more than 100 IR datasets show that embedding-based methods exhibit increasing geometric saturation, while token-level conditioning remains much more stable. The paper aims to provide a principled explanation for why embeddings still work well in practice despite known theoretical limitations, while also arguing that token-level probabilistic retrieval may become advantageous in more compositional large-scale retrieval settings.

**Compliance With Llm Reviewing Policy:**

Affirmed.

**Final Justification:**

While this paper provides interesting theoretical insights into generative retrieval, its practical significance requires more evidence and better justification. So I decide to maintain my original rating.

**Key Questions For Authors:**

1. How much of the theoretical gain comes from diffusion itself versus simply increasing conditioning capacity via token sequences?

2. If embedding degradation is small at current scales, in what realistic settings would embeddings reach the predicted saturation regime?

3. What advantage does diffusion-based retrieval provide over existing token-level methods such as late interaction?

**Limitations:**

yes

**Strengths And Weaknesses:**

Strengths
- The paper addresses a relevant and timely question: how the sign-rank barrier manifests in practical retrieval settings, and why embedding-based retrieval still works well despite its theoretical limitations.
- A major strength is the attempt to place embedding retrieval, diffusion with embedding conditioning, and diffusion with token-level conditioning into a unified theoretical framework. This makes the paper conceptually coherent and gives a clearer geometric perspective on the potential advantage of token-level conditioning.
- The theoretical analysis is carefully developed and uses standard, meaningful tools such as geometric distortion, local Lipschitz sensitivity, covering-number arguments, and sign-rank-inspired complexity proxies.
- The empirical section is reasonably broad in scope, covering over one hundred IR datasets and multiple pretrained encoders, which makes the observed trends more convincing than results on only a few benchmarks.


Weaknesses
- The practical significance is limited because the paper largely abstracts away the central capacity bottleneck of generative retrieval systems: the model must effectively memorize or represent a large number of query-document associations in its parameters. In practice, model size, optimization difficulty, and memorization capacity are major obstacles for generative retrieval, but these issues are not really modeled here. The paper itself acknowledges that the analysis ignores training instability, memory cost, stochasticity, and parameter scaling considerations.
- The theoretical advantage of token-level conditioning is somewhat less surprising than the paper suggests. The logarithmic result in Theorem A.3 relies crucially on allowing the conditioning sequence length to grow so that the discrete conditioning capacity
$|V|^L$ is large enough to distinguish the relevant queries. In that sense, the comparison is close to contrasting fixed-capacity conditioning with growing-capacity conditioning, which weakens the conceptual novelty of the result.
- More broadly, the “exponential vs. logarithmic” contrast is not entirely apples-to-apples. On the embedding side, the theory emphasizes exponential growth in local Lipschitz sensitivity under fixed-dimensional continuous conditioning; on the token-conditioning side, the argument uses a covering-number / metric-entropy bound to obtain logarithmic sample complexity. These are related but not identical quantities, so the comparison can feel somewhat rhetorically stronger than what is actually proved.
- Although the paper repeatedly emphasizes scaling behavior, the empirical section mainly reports very small fitted exponents and near-flat trends, especially for token-level conditioning. This makes the observed practical effect rather modest. The paper itself concludes that the degradation of embedding methods remains small in current regimes, which somewhat reduces the practical force of the headline claim.
- The empirical analysis is entirely geometry-based and does not establish a concrete link to end-to-end retrieval quality. The paper explicitly avoids downstream metrics such as Recall or NDCG and acknowledges that it cannot conclude these geometric effects directly cause retrieval failures. As a result, the work is more diagnostic than demonstrative from a systems perspective.

---

> ### Author Rebuttal · Authors · 2026-03-29
>
> We thank Reviewer RXQF for their nuanced assessment. The characterization *more diagnostic than demonstrative* aligns with our intent, and we regret this positioning was insufficiently clear.
>
> **W1 (Practical significance):** We agree that memorization, optimization, and parameter scaling are critical challenges our framework deliberately abstracts. Our goal is to characterize geometric requirements any correct retrieval system must satisfy, independent of implementation. Analogous to channel capacity, one need not build the capacity-achieving code to establish the bound. Our oracles provide lower bounds that any learned system must respect.
>
> **W2 (Token conditioning advantage *less surprising*):** Fair. The intuition that growing-capacity conditioning scales better than fixed-capacity conditioning is natural. We see our contribution not in the direction of the result, but in making it precise: proving the growth rate is logarithmic in $N_q$, establishing that geometric sensitivity remains bounded under this regime, and showing via newly added experiments ($K_4$, see Q1) that capacity alone is insufficient without expressivity. Without formalization, the intuition cannot distinguish between *token conditioning helps* and *token conditioning suffices*, nor where it fails. We will revise to frame this as quantifying an intuitive advantage rather than revealing a surprising one.
>
> **W3 (Different mathematical quantities):** This is undoubtedly a valid observation we should have disclosed explicitly. Local Lipschitz sensitivity and metric-entropy bounds are distinct quantities. However, both answer the same question of how system complexity must scale to maintain retrieval correctness as $N_q$ grows. Theorem A.1 shows mapping complexity grows exponentially under fixed conditioning; Theorem A.3 shows conditioning resolution grows logarithmically under token-level input. These are different scaling strategies for identical information-theoretic requirements, analogous to comparing space vs. time complexity for the same problem.
>
> We anticipate the objection that this weakens our headline framing. We agree and will revise to: *exponential growth in mapping complexity under fixed conditioning vs. logarithmic growth in conditioning resolution under token-level input*: complementary quantities informing the same design decision.
>
> **W4 (Small effects):** We view this as a finding: our paper asks why embeddings work well despite theoretical limitations, and answers that the exponential trend is real ($K_1: R^2=0.956, p<10^{-6}$) but at tiny magnitudes ($b=-0.0003$). This resolves the theory-practice gap rather than undermining our claims.
>
> **W5 (Correlation with retrieval metrics):** We analyzed correlation between geometric metrics and retrieval measures across all 101 datasets. Recall, Precision, F1 normalized by theoretical maximum; NDCG, MRR, Success unnormalized. Averaged across @5/10/20/50/100 for $b_1$ as the retrieval metrics originate from this paradigm. Pearson (P) and Spearman (S), p < 0.05:
>
> Sensitivity: Recall ($P: 0.749, p=.008; S: 0.955, p<10^{-5}$), Precision ($P: -0.635, p=.036; S: -0.936, p<10^{-5}$), F1 ($P: 0.689, p=.019$), NDCG ($S: 0.745, p=.008$), Success ($S: -0.900, p<10^{-3}$).
>
> Distortion: Recall ($P: -0.954, p<10^{-5}; S: -0.952, p<10^{-5}$), Precision ($P: 0.949 p<10^{-4}; S: 0.879 p<10^{-3}$), NDCG ($P: -0.838, p=.002; S: -0.842, p=.002$), Success ($P: 0.798, p=.006$).
>
> $K_3$ shows no significant correlation with any metric, consistent with a well-calibrated oracle. We do not claim causation. However, $|\rho| > 0.9$ for multiple pairs validates geometric diagnostics as meaningful proxies for retrieval quality. MRR shows no correlation for any K so far, potentially explained by the single top-ranked document being more likely to appear earlier for larger number of relevant docs per query. Full results and methodology will appear in the revised version.
>
> **Q1 (Diffusion vs. token-level conditioning):** Token conditioning provides capacity, diffusion expressivity. In new post-submission experiments, $K_4$ isolates this: token-level conditioning with centroid retrieval mapping (high capacity, low expressivity) yields exponential scaling ($b=-0.0005, R^2=0.963, p<10^{-6}$). $K_2$ (expressivity without capacity) yields $R^2=0.876$. Only $K_3$ (both) yields flat sensitivity ($R^2 = 0.042, p = 0.545$). Both components contribute independently.
>
> **Q2:** Embeddings are already in the exponential regime ($K_1: p < 10^{-6}$). When magnitudes become catastrophic is speculative, but our framework enables monitoring this for arbitrary encoder-corpus pairs.
>
> **Q3 (Late interaction):** $K_4$ experiments (see Q1) directly address this: token-level conditioning with centroid retrieval (high capacity, low expressivity) yields exponential scaling identical to embeddings: capacity alone is insufficient without expressivity. Late interaction may face analogous expressivity constraints and is promising future work.

---

> > ### Author Rebuttal · Reviewer_RXQF · 2026-04-02
> >
> > Thank you for the careful and thoughtful rebuttal. I appreciate the clarifications and the effort to better position the paper as a diagnostic and theoretically motivated study. The additional analyses, especially regarding correlations with retrieval metrics, are helpful.
> >
> > However, my main concern regarding the practical significance remains. While the geometric perspective is well-developed and the theoretical insights are sound, the empirical effects appear modest, and the connection to end-to-end retrieval improvements is still limited.
> >
> > Therefore, although I find the work interesting and technically solid, I maintain my original rating

---

> > > ### Author Response · Authors · 2026-04-02
> > >
> > > We sincerely thank Reviewer RXQF for their time, constructive engagement, and for recognizing the theoretical soundness and technical solidity of our work. We completely respect your assessment regarding the practical significance, and we view your rationale for maintaining the score as entirely fair!
> > >
> > > As we explicitly acknowledge in our discussion, the current empirical magnitudes of these geometric limits are indeed modest at present scales, and our contribution is fundamentally a diagnostic baseline rather than an immediate end-to-end system improvement. We recognize that the balance between theoretical/diagnostic foundations and immediate empirical gains is evaluated differently across the community, and we accept your evaluation of this trade-off as completely valid.
> > > Your feedback, particularly in prompting the correlation analysis, has genuinely strengthened the manuscript and helped us clarify our boundaries.
> > >
> > > Thank you again for your time, expertise, and fair evaluation!

---

### Official Review · Reviewer_GYEk · 2026-03-11

**Soundness:** 3
**Presentation:** 3
**Significance:** 2
**Originality:** 2
**Overall Recommendation:** 3
**Confidence:** 4

**Summary:**

This paper studies the geometric limitations of fixed-dimensional embedding-based retrieval under the sign-rank barrier as corpus size increases. A parameter- and training-free diagnostic framework based on the local Lipschitz constant is proposed to analyze geometric saturation and to compare embedding-based retrieval with diffusion-based probabilistic retrieval. The scaling behavior and separability properties of the two paradigms are analyzed through theoretical arguments and experiments on more than 100 information retrieval datasets.

**Compliance With Llm Reviewing Policy:**

Affirmed.

**Key Questions For Authors:**

The experimental evaluation in Section 5 analyzes geometric sensitivity using three sentence embedding models but does not include comparisons with state-of-the-art retrieval systems (e.g., ColBERT, DSI) on standard IR benchmarks. Could the authors provide head-to-head empirical comparisons between the diffusion-based retrieval framework proposed in Section 4 and strong embedding-based baselines using standard retrieval metrics (e.g., NDCG, Recall@k) across different corpus sizes and query complexities?

Theorem A.1 assumes document embeddings are “pairwise separated by a large constant” relative to the diffusion noise level σ. In practice, dense embeddings (e.g., from contrastive learning) are known to crowd on the hypersphere, making such separation unrealistic. How does this assumption affect the real‑world relevance of your exponential lower bound? If the bound collapses under realistic embedding distributions, what remains of the theoretical motivation for diffusion?

Section 4.1 explicitly states that the diffusion analysis relies on a “zero‑error final‑timestep approximation” and that no diffusion model is actually trained or evaluated (as admitted in Section 5.4). Given this, on what empirical basis do the authors claim that diffusion “circumvents the sign‑rank barrier” (Section 1) and exhibits “favorable theoretical scaling” (Section 5.3)? How would the conclusions change if practical issues such as score‑field instability or inference stochasticity were taken into account?

The geometric sensitivity metric K_b is defined as a local Lipschitz constant (Section B.5) and is used as the primary experimental indicator in Section 5.2. However, the paper does not demonstrate that this metric correlates with actual retrieval failures or ranking degradation on the analyzed datasets. Could the authors provide evidence that higher geometric sensitivity corresponds to measurable declines in retrieval metrics such as Recall or NDCG? Establishing such a link would strengthen the practical interpretation of the experimental results.

Appendix B.2 approximates the diffusion score field using only the nearest neighbor among relevant documents, completely ignoring contributions from all other relevant documents and all non-relevant documents. Real score fields are mixtures of all Gaussians. This approximation appears quite strong. Could the authors clarify how this simplification affects the validity of the diffusion analysis?

**Limitations:**

Yes.

**Strengths And Weaknesses:**

Soundness

Strengths

The paper provides a technically detailed analysis of the geometric constraints of embedding-based retrieval. Theoretical analysis formalizes sign-rank barrier constraints on fixed-dimensional embedding retrieval, proving logarithmic scaling for token-level conditioned diffusion vs. exponential scaling for embeddings. Experiments cover 100+ IR datasets and three embedding models (all-MiniLM-L6-v2, all-mpnet-base-v2, all-roberta-large-v1), with logarithmic n_q binning isolating combinatorial stress.

Weaknesses
	Diffusion analysis (Sec. 4) is based on an idealized formulation that assumes an exact score field and a zero-error final timestep, ignoring optimization dynamics, approximation errors in score estimation, and training instability in diffusion models.
	Empirical validation is fundamentally disconnected from retrieval performance. Section 5.2 measures only geometric proxies (Lipschitz constants) but provides no comparison against established retrieval baselines (e.g., ColBERT, DPR). In Section 5.4, the authors explicitly acknowledge that the proposed geometric metrics are not necessarily correlated with standard retrieval metrics such as Recall or NDCG. As a result, the empirical analysis does not convincingly demonstrate that the observed geometric sensitivity leads to practical retrieval failures. This significantly weakens the causal link between the theoretical analysis and real retrieval performance.
	The theoretical conclusions rely on assumptions that significantly simplify real-world retrieval systems. In Section 3 and Appendix A, the analysis of the sign-rank barrier assumes a fixed-dimensional single-vector representation and a simplified similarity formulation. However, many modern retrieval systems rely on multi-vector representations or late-interaction architectures. Because these architectures are not considered in the theoretical model, the generality of the claimed geometric limitation is unclear.

Presentation

Strengths
Well-structured (problem-theory-method-experiment-discussion). Clear positioning against prior work; key concepts visualized; technical details (proofs, metrics) in appendices.

Weaknesses

Section 4's transition is abrupt: the conceptual transition from the geometric analysis to the diffusion-based formulation lacks an intuitive explanation of why token conditioning avoids the sign-rank barrier.

The experimental section lacks sufficient methodological clarity for reproducibility. In Section 5.1, the process used to construct query bins based on n_q is only briefly described, and details regarding dataset preprocessing, query grouping, and evaluation procedures are limited.

Significance

Does the paper address an important or relevant problem? Does it advance understanding, capabilities, or practice in ML? Could it influence future research or applications? Even modest or domain-specific improvements can be significant if they unlock new directions or provide practical utility.

Strengths
The work describes an important question regarding the representational limits of fixed-dimensional embeddings in large-scale retrieval. Understanding the geometric behavior of embedding spaces as retrieval complexity increases is potentially valuable for future research on retrieval architectures and representation learning.

Weaknesses

Although diffusion-based retrieval is proposed as an alternative paradigm in Section 4, the paper does not provide empirical evidence showing that diffusion models outperform embedding-based retrieval in realistic retrieval benchmarks. Without empirical comparison to existing retrieval systems or actionable design implications, the contribution remains largely conceptual rather than providing clear guidance for improving practical retrieval models.

Originality

Strengths
The paper proposes a geometric perspective on information retrieval by connecting the sign-rank barrier with empirical scaling behavior in embedding-based retrieval. The introduction of geometric sensitivity as a diagnostic indicator provides an interesting analytical perspective for studying retrieval representations.

Weaknesses

Section 4 mainly interprets retrieval as conditional sampling within a standard diffusion modeling framework. However, the paper does not propose a new architecture, objective function, or training procedure specifically designed for retrieval, so the contribution functions more as a conceptual perspective than a methodological innovation.

The experiments (Sec. 5) measure geometric sensitivity across datasets and encoders, but they do not introduce new benchmarks, evaluation protocols, or findings that substantially change the current understanding of retrieval system behavior.

---

> ### Author Rebuttal · Authors · 2026-03-29
>
> We thank Reviewer GYEk for their thorough and structured review. We recognize several concerns stem from a misalignment between our intent and what the paper communicates, and take responsibility for this.
>
> **The role of diffusion:** We regret failing to clearly establish diffusion's role. Diffusion is used as a mathematical tool that is (i) provably capable of distortion-free retrieval under idealized conditions and (ii) geometrically intuitive. Beyond this, its selection is fairly arbitrary. Our contribution is a diagnostic framework using idealized oracles to isolate geometric properties of pre-trained embedding spaces, not a diffusion retrieval system.
>
> Replacing the oracle with a learned model would undermine the analysis: distortion requires a reference target (no oracle=no ground truth), and sensitivity through a learned model conflates encoder geometry with model capacity.
>
> **Q1 (Head-to-head comparison with ColBERT, DSI):** We understand why this is desired, but it answers a different question. Standard retrieval metrics measure end-to-end system quality, a convolution of encoder geometry, retrieval mapping, and training dynamics. Our framework deliberately decomposes these to study geometry in isolation.
>
> That said, we validate our metrics are not disconnected from practice. In our response to Reviewer RXQF, we present new experiments showing statistically significant correlations between geometric diagnostics and standard retrieval metrics across all 101 datasets.
>
> Key results: geometric sensitivity $K_1$ correlates (Spearman) with Recall ($\rho=0.955, p<10^{-5}$), Precision ($\rho=-0.936, p<10^{-5}$), and NDCG ($\rho=0.745, p<0.009$). Geometric distortion shows even stronger associations (Recall: $\rho=-0.952, p<10^{-5}$; Precision: $\rho=0.879, p<10^{-3}$; NDCG: $\rho=-0.842, p=0.002$). While qualitative analysis is required to claim causation, these results validate that our diagnostics correlate with downstream performance.
>
> **Q2 (Contrastive learning and the separation assumption):** Contrastive learning allocates embedding space more efficiently but does not create additional capacity, the same fixed-dimensional crowding persists. Regarding Theorem A.1's pairwise separation assumption: it is intentionally set in a regime favorable to embedding retrieval. If embeddings crowd on the hypersphere (as the reviewer correctly notes), the exponential lower bound becomes *harder* to satisfy, not easier, so our result is conservative. Token-level conditioning offers a new scaling axis orthogonal to contrastive optimization: the former scales conditioning capacity, the latter optimizes encoder representations. They are complementary, not substitutes.
>
> **Q3: (Zero-error assumption):** The zero-error oracle isolates encoder geometry, not a claim about practical models. At T=0, the score field is maximally expressive, providing the sharpest geometric bounds, any other timestep yields a smoothed representation with less pronounced sensitivity, making T = 0 the principled worst-case per timestep.
>
> We anticipate the concern that idealized analysis may not transfer: (1) the oracle provides a *lower bound* on geometric pressure any correct system faces; (2) the correlation results above confirm oracle-derived metrics track real performance.
>
> **Q4 ($K_b$ correlation with retrieval failures):** Addressed above and detailed in our response to Reviewer RXQF.
>
> **Q5 (Nearest-neighbor approximation):** At T=0 the score field is maximally sharp: each point is directed toward its nearest relevant document. The nearest-neighbor limit is the appropriate choice at this timestep, providing the most demanding geometric scenario. Introducing Gaussian mixtures would require tuning bandwidth parameters that are highly sensitive in high dimensions and encoder-dependent, weakening cross-encoder comparability.
>
> **Contribution scope:** We respectfully maintain that geometric distortion and sensitivity as parameter-free diagnostics constitute a methodological contribution, the first empirical estimate of sign-rank barrier effects in pre-trained retrieval, to our knowledge. We want to clarify our intended direction for future work: it is not *build diffusion for IR* but *design retrieval systems informed by their geometric properties*. Diffusion is one analytically convenient example; any architecture achieving sufficient expressivity and capacity alignment would benefit equally. We believe diagnosing geometric bottlenecks is prerequisite to systematically addressing them, regardless of the specific technology chosen.
>
> **Reproducibility:** All formulas are in the text or appendix, and full code is included. We are happy to expand specific steps the reviewer finds unclear.
>
> **Presentation:** We will add an intuitive explanation: token conditioning shifts from a fixed-dimensional bottleneck (exponential demands) to variable-length conditioning (capacity |V|^L, requiring only logarithmic growth in L).

---

> > ### Author Rebuttal · Reviewer_GYEk · 2026-04-01
> >
> > The rebuttal provides useful clarifications and partially addresses my concerns. In particular, the newly added correlation analysis between the proposed geometric metrics and standard retrieval metrics (Recall, Precision, NDCG) strengthens the empirical connection to practical performance.
> >
> > However, several key issues remain unresolved:
> >
> > 1.The paper still lacks head-to-head comparisons with established retrieval systems (e.g., ColBERT, DPR) using standard evaluation protocols. The provided correlation analysis does not substitute for end-to-end retrieval evaluation, and thus the practical advantage of the proposed framework remains unclear.
> >
> > 2.The diffusion-based analysis continues to rely on idealized assumptions (e.g., zero-error oracle, exact score field) without empirical validation using trained diffusion models. As a result, the claim that diffusion circumvents the sign-rank barrier remains largely theoretical.
> >
> > 3.Some theoretical assumptions (e.g., pairwise separation) are acknowledged but not empirically examined under realistic embedding distributions, leaving questions about real-world applicability.

---

> > > ### Author Response · Authors · 2026-04-01
> > >
> > > We sincerely thank the reviewer for their continued engagement and for recognizing that our new correlation analysis strengthens the empirical connection to practical performance. Your follow-up highlights a fundamental flaw in how we framed our contributions, and we deeply appreciate the opportunity to clarify.
> > >
> > > **1. On head-to-head comparisons (ColBERT, DPR)**
> > > You are completely right: if we were proposing a new, deployable retrieval system, head-to-head comparisons against ColBERT and DPR would be absolutely mandatory. We take full responsibility for this confusion: our text overly emphasized "diffusion-based retrieval" as a competing paradigm, rather than what it actually is in this paper: an idealized, mathematical bounding tool (an oracle).
> > >
> > > Our contribution is strictly an **analytical and diagnostic framework** to measure the geometric strain on pre-trained encoders (which systems like DPR or ColBERT must build upon). Because downstream retrieval metrics (NDCG) entangle encoder geometry, mapping capacity, and training dynamics, we use our oracle-based framework to isolate the *encoder geometry* alone. We will explicitly reframe the introduction and discussion to state clearly that we are proposing a diagnostic tool for researchers, not a competitor to DPR/ColBERT, and we will explicitly mention evaluating these end-to-end systems as the immediate next step for future work.
> > >
> > > **2. On idealized diffusion assumptions without trained validation**
> > > This ties directly into the point above. You are entirely correct that our diffusion analysis relies on idealized, zero-error assumptions. We intentionally use this idealized formulation because we *need* a perfect "ground truth" oracle to measure geometric distortion against.
> > >
> > > If we trained an actual diffusion model, optimization instabilities and approximation errors would be introduced. We would then have no way to isolate whether a drop in performance was due to the underlying *geometric sign-rank barrier of the embeddings* or simply *training instability*. By using the idealized zero-error limit, we establish a strict **lower bound**: we show the geometric pressure that *even a mathematically perfect* retrieval mapping faces. We will add a dedicated paragraph clarifying that this idealized diffusion is a "theoretical probe" to measure embeddings, not a claim of current real-world diffusion superiority.
> > >
> > > **3. On pairwise separation under realistic embedding distributions**
> > > Your intuition here is absolutely on point, and this is actually the exact phenomenon we aimed to measure in Section 5! You are completely correct that real dense embeddings (trained via contrastive learning) crowd on the hypersphere, heavily violating the ideal pairwise separation assumed in Theorem A.1.
> > >
> > > Theorem A.1 predicts that *if* separation is poor, the Lipschitz constant (sensitivity) will scale exponentially. Because we knew real embeddings violate this separation, we ran the experiments across the 101 datasets to see what happens in reality. Our empirical finding (Section 5) proves your exact intuition: the exponential scaling *is* undeniably present in practice ($K_1$: $R^2 = 0.956$, $p < 10^{-6}$), meaning the theoretical sign-rank barrier actively manifests due to this crowding. However, the exponent is remarkably small ($b = -0.0003$).
> > >
> > > Thus, our empirical answer to your question is: *When the separation assumption is violated by realistic contrastive embeddings, the theoretical exponential blow-up occurs, but modern encoders map the space efficiently enough that the practical magnitude of this failure remains small at current scales.* We realize we did not connect this empirical finding back to the theoretical assumption clearly enough, and we will explicitly highlight this connection in the revised text. We will add an explicit paragraph in Section 5.3 that directly connects the crowding observed in contrastive embeddings to the exponential scaling predicted by Theorem A.1 and a small empirical exponent we report.
> > >
> > > We hope this clarifies that we are in violent agreement with your empirical intuitions, and we are committed to fixing the framing of the paper to reflect this.

---

### Official Review · Reviewer_99Pm · 2026-03-15

**Soundness:** 3
**Presentation:** 4
**Significance:** 3
**Originality:** 3
**Overall Recommendation:** 4
**Confidence:** 3

**Summary:**

This paper investigates the expressive capacity bottleneck of fixed-dimensional embedding models in information retrieval (IR), namely the exponential sign-rank barrier. The authors propose a parameter-free and training-free diagnostic framework that quantifies the geometric saturation effects of retrieval models through two core metrics: the local Lipschitz constant and the divergence from the target distribution.
The main contributions of the paper are as follows: (1) It formally proves that token-level conditioning in diffusion models can circumvent the sign-rank barrier, reducing the dimensionality or capacity requirements for maintaining retrieval separability from exponential to logarithmic scaling. (2) It validates the divergent scaling trajectories of embedding models and diffusion models across over 100 IR datasets. (3) It explains why embedding models, despite their theoretical sub-optimality, still perform remarkably well under current task complexities.

**Compliance With Llm Reviewing Policy:**

Affirmed.

**Final Justification:**

The authors have provided a comprehensive response to the review comments. Hence, I maintain my score.

**Key Questions For Authors:**

(1) Token-level conditioning requires a cross-attention mechanism to process long sequences. For corpora at the million-scale or larger, will this geometric advantage be offset by enormous inference overhead?
(2) If the database distribution changes, is it necessary to retrain the diffusion model, or can adjustments be made only during the inference phase?
(3) The framework intentionally abstracts away retrieval metrics such as NDCG. Do the authors have any preliminary correlation analysis to illustrate the relationship between geometric metrics and the degradation of actual retrieval performance?

**Limitations:**

Yes

**Strengths And Weaknesses:**

Strengths:
(1) The paper features rigorous theory with detailed proofs, revealing how token-level conditioning leverages the combinatorial capacity of discrete vocabularies to avoid geometric crowding.
(2) Tests conducted across over 100 diverse IR datasets enhance the generality and persuasiveness of the conclusions.
(3) The overall writing of the article is relatively clear, and the structure is reasonably arranged. The formula derivation process is relatively comprehensive, and the illustrations effectively help clarify the geometric differences between diffusion processes and embedding-based retrieval.

Weaknesses:
(1) In real-world retrieval scenarios, queries may contain natural noise (such as spelling errors and ambiguous expressions), and these perturbations may affect the stability of retrieval systems. The paper fails to evaluate whether its diagnostic framework can accurately capture changes in geometric distortion and sensitivity under perturbed scenarios.
(2) The paper does not conduct sufficient analysis on computational cost, scalability, and inference efficiency.

---

> ### Author Rebuttal · Authors · 2026-03-29
>
> We sincerely thank Reviewer 99Pm for the positive assessment and recognition of both the theoretical rigor and the breadth of our empirical evaluation. We address the remaining concerns and questions below.
>
> **Q1 (Cross-attention overhead at million-scale corpora):** This is an important practical consideration. We want to clarify the scaling arithmetic: for a standard vocabulary of $|V| \sim 30,000$ (strictly less than all encoders analyzed), a sequence length of L=2 already yields $|V|^2 \sim 9 * 10^8$ distinguishable conditioning inputs; sufficient for corpora far beyond current benchmarks, yet involving no meaningful cross-attention overhead. Of course, L=2 carries no semantic content. But even modest thresholds of L=25 or L=100 produce astronomically large capacity, while remaining well within the sequence lengths that current language models and diffusion language models routinely process (thousands to hundreds of thousands of tokens).
>
> The key distinction our framework highlights is a shift from a *geometric* bottleneck (exponential sensitivity growth under fixed-dimensional conditioning, which cannot be engineered away) to a *capacity* bottleneck (cross-attention cost scaling with L, which is amenable to standard engineering solutions: efficient attention, quantization, hardware scaling). We see this as a favorable trade-off: the community has decades of experience scaling capacity constraints, while geometric bottlenecks are structural.
>
> **Q2 (Distribution shift and retraining):** We assume a pre-trained embedding model that is semantically well-defined globally (trained on diverse document corpora). The diffusion model would learn to navigate this embedding landscape by discovering relationships between token sequences and document positions in the space.
>
> If the training scope is narrow and the distribution shift drastic, the diffusion model will be unable to traverse unfamiliar regions of embedding space: retraining of the retrieval mapping would be necessary, regardless of how smooth the underlying geometry is. However, for broad-scope training (analogous to the diversity seen in large-scale language model pre-training), near-global understanding of the embedding landscape is a reasonable expectation. Distribution shifts within the support of training data could plausibly be handled at inference time without retraining.
>
> **Q3 (Correlation between geometric metrics and retrieval performance):** We appreciate this question, as it was raised by multiple reviewers. In our response to Reviewer RXQF, we present comprehensive new experiments analyzing the correlation between our geometric diagnostics and standard retrieval metrics (Recall, Precision, F1, NDCG, MRR, Success) across all 101 datasets.
>
> Key findings: geometric sensitivity ($K_1$) shows strong rank correlation (Spearman) with Recall ($\rho = 0.955, p < 10^{-5}$), Precision ($\rho = -0.936, p < 10^{-5}$), and NDCG ($\rho = 0.745, p < 0.009$). Geometric distortion shows even stronger associations, with $|\rho| > 0.84$ for Recall, Precision, NDCG, and Success (all $p < 0.01$). $K_3$ (well-calibrated oracle) shows no significant correlation with any metric, consistent with theoretical predictions. MRR is the only metric showing no significant correlation, which can be attributed to its inherent bias towards larger number of relevant documents (i.e., if a query has 1000 relevant documents, one of them is more likely to appear at a higher rank).
>
> We remain cautious about claiming causation, as retrieval metrics are dataset-dependent and sensitive to corpus composition. Nonetheless, these results establish that our geometric diagnostics meaningfully track downstream retrieval quality, validating the framework's practical relevance. Full statistical details are provided in our response to Reviewer RXQF.
>
> **W1 (Robustness under query noise):** We acknowledge this was insufficiently covered. We offer the following observations: (1) datasets were not hand-selected and likely contain natural noise from their creation, which is implicitly reflected in our results; (2) geometric sensitivity is computed over millions of query-document pairs across 101 datasets, averaging out random noise by construction, systematic bias would be required to distort the signal. A controlled perturbation study (injecting synthetic noise at varying levels) would make for valuable future work and is compatible with our framework.
>
> **W2 (Computational cost):** We will add exact hardware specifications and runtimes in the appendix. The primary experiments were conducted on a Mac Studio (96 GB RAM) and MacBook Air (48 GB RAM), with each full run across all three encoders and 101 datasets taking approximately 2–3 days per encoder. The framework is parameter-free and requires no training, making computational costs modest relative to model training.

---

> > ### Author Rebuttal · Reviewer_99Pm · 2026-04-04
> >
> > The authors have provided a comprehensive response to the review comments. Hence, I maintain my score.

---

### Decision · Program_Chairs · 2026-04-30

**Decision:**

Reject

**Comment:**

- The detailed rebuttal and additional analyses helped clarify the intended diagnostic scope of the work. While the following are generally agreed: the theoretical framework is internally consistent, key concerns remain regarding the reliance on idealized oracle constructions and the lack of empirical validation using trained retrieval systems.
- The evaluation remains largely geometry‑based and does not include end‑to‑end comparisons against established retrieval baselines, limiting evidence that the observed geometric effects translate to measurable downstream improvements in practice.
- As a result, the practical impact of the work is currently limited, and the connection between the theoretical analysis and real‑world retrieval performance remains unclear. I therefore recommend rejection at this time.